# Links of Climate Variability among Arctic sea ice, Eurasia teleconnection pattern and summer surface ozone pollution in North China

Zhicong Yin [12*], Huijun Wang[12], Yuyan Li[1], Xiaohui Ma[3], Xinyu Zhang[1]

[1]Key Laboratory of Meteorological Disaster, Ministry of Education / Joint International Research Laboratory of Climate and Environment Change (ILCEC) / Collaborative Innovation Center on Forecast and Evaluation of Meteorological Disasters (CIC-FEMD), Nanjing University of Information Science & Technology, Nanjing 210044, China
[2]Nansen-Zhu International Research Center, Institute of Atmospheric Physics, Chinese Academy of Sciences, Beijing, China
[3]Institute of Urban Meteorology, CMA / Environmental Meteorology Forecast Center of Beijing-Tianjin-Hebei

*Corresponding author:* Zhicong Yin(yinzhc@163.com)

**Abstract.** Summer surface $O_3$ pollution has rapidly intensified in China in the recent decade, damaging human and ecosystem health. In 2017, the summer mean maximum daily average 8 h concentration of ozone was greater than 150 μg/m³ in North China. Based on the close relationships between the $O_3$ concentration and the meteorological conditions, a daily surface $O_3$ weather index was constructed, which extends the study period to the historical period before 2007 and the projected future. Here, we show that in addition to anthropogenic emissions, the Eurasia teleconnection pattern (EU), a major globally atmospheric teleconnection pattern, influences surface $O_3$ pollution in North China on a time scale of climate. The local meteorological conditions associated with the EU positive phase supported intense and efficient photochemical reactions to produce more surface $O_3$. The associated southerlies over North China transported surrounding $O_3$ precursors to superpose local emissions. Increased solar radiation and high temperature during the positive EU phase dramatically enhanced $O_3$ production. Furthermore, due to the close connection between the preceding May Arctic sea ice and summer EU pattern, approximately 60% of the interannual variability of $O_3$-related weather conditions was attributed to Arctic sea ice to the north of Eurasia. This finding will aid in understanding the interannual variation of $O_3$ pollution, specifically the related meteorological conditions.

**Keywords:** ozone pollution, photochemical reaction, Eurasia teleconnection pattern, climate change, Arctic sea ice

## 1 Introduction

Over the past several decades, due to fast economic development, air pollution has been increasing in China (Chen, 2013; Watts et al., 2018). The major air pollution types in China are haze pollution (i.e., high-level fine particulate matter) in winter (Yin et al., 2015; Wang, 2018) and surface ozone ($O_3$) pollution in summer (Ma et al., 2016; Tang et al., 2018). Due to drastic air pollution control in China since 2013, haze pollution is being controlled in recent years (The environmental statistics unit of stat-centre in Peking University, 2018), appearing as a sharp decrease in fine particulate matter. However, surface $O_3$ pollution, which always occurs on clear and sunny days (Wang et al., 2017), has not improved (Li et al., 2018). The negative effects of

surface $O_3$ pollution, such as corroding human's lungs and destroying agricultural crops and forest vegetation, were not weaker than those of haze (Liu et al., 2018), but the impacts of climate variability on surface $O_3$ pollution in China (Yang et al, 2014) have not been sufficiently studied. In the major urban agglomerations in China, such as Beijing-Tianjin-Hebei, Yangtze River delta and the Pearl River delta, the surface $O_3$ concentrations exceeded the ambient air quality standard of China (100 μg/m³) by 100–200 % (Wang et al., 2017). In the Yangtze River delta, the inter-annual variations of NO and $O_3$ levels generally presented decreasing and increasing trends respectively, from 2012 to 2015 at both urban and suburban sites (Tong et al., 2017). Furthermore, the concentration of $O_3$ and its precursors, e.g. nitrogen oxides ($NO_x$) and volatile organic compounds (VOCs), in Beijing-Tianjin-Hebei was significantly larger than that in other regions of China (Wang et al., 2006; Shi et al., 2015). Revealed by the datasets from Shangdianzi Station, the long-term trend of $O_3$ concentrations in North China indicated that the $O_3$ pollution has undergone a significant increase in the period of 2005–2015, with an average rate of 1.13 ±0.01 ppb year$^{-1}$ (Ma et al., 2016).

Surface $O_3$ is a secondary pollutant. The precursors of $O_3$ photochemically react with sunlight to generate $O_3$ under suitable weather conditions, i.e., hot-day and sunny environments (An et al., 2009). Surface deposition, dynamic transport and dispersion of $O_3$ are also closely related to atmospheric circulations. For example, the prevailing positive phase of the North Atlantic Oscillation contributed to the increasing ozone concentration in western and northern Europe through the anomalous atmospheric circulations that influence regional photochemical processes (Christoudias et al., 2012; Pausata et al., 2012). The summer surface $O_3$ variability in North America is significantly modulated by the position of the jet stream (Lin et al., 2014). Barnes and Fiore (2013) pointed out jet position may dynamically modulate surface ozone variability in eastern North America and other northern mid-latitude regions. A strong positive correlation between the East Asian summer monsoon and summer mean ozone were found by model simulations (Yang et al., 2014), illustrating that the changes in meteorological parameters, associated with East Asian summer monsoon, lead to 2–5% interannual variations of surface $O_3$ concentrations over central eastern China. Focusing on the dataset in 2014, a significantly strong west Pacific subtropical high resulted in higher relative humidity, more clouds, more rainfall, less ultraviolet radiation and lower air temperatures, which were unfavourable for the formation of $O_3$ (Zhao and Wang, 2017). The photochemical reaction was the main local sources of $O_3$ (Sun et al., 2019). The hot and dry environments and the intense solar radiation could accelerate the chemical conversion from the precursor to $O_3$ (An et al., 2009; Tong et al., 2017). In 2013, a severe heat wave, with highest temperature 41.1 ℃, contributed to the high $O_3$ concentration in the Yangtze River Delta (Pu et al., 2017). The frequency of large-scale, extreme heat events is closely related to atmospheric patterns, such as the Eurasia teleconnection pattern (EU; Pu et al., 2017; Li and Sun, 2018) and aerosol effective radiative forcing (Liu and Liao, 2017). The winds from a polluted area also transport $O_3$ and its precursors downwind (Doherty et al., 2013). Due to the close relationship between surface $O_3$ and meteorological conditions, the impacts of climate change on $O_3$ have been projected by various numerical models (Doherty et al., 2013; Melkonyan and Wagner, 2013; Zhu and Liao, 2016;

Gaudel et al., 2018). Over eastern China, the surface ozone concentration and possibility of severe ozone pollution may both increase in the future (Wang et al., 2013).

However, previous studies of $O_3$ pollution in China mainly focused on observational analyses of several synoptic processes (e.g., Zhao and Wang, 2017), rather than long-term climate diagnostics, because of the lack of long-term surface $O_3$ observations. The goal of this study is to examine the large-scale atmospheric circulations associated with the interannual variation of summer surface $O_3$ pollution in North China based on long-term meteorological observations. The role of May Arctic sea ice (ASI), as a preceding and effective driver, is also analysed. The outcomes of our research, in terms of climate variability, may provide a basis for understanding the interannual variation of $O_3$ pollution, specifically the related meteorological conditions.

**2 Data and Method**

The hourly $O_3$ concentration data from 2014 to 2017 in China were provided by the Ministry of Environmental Protection of China. As one of the three regional background air-monitoring stations in China, the hourly $O_3$ concentration data at the Shangdianzi station (SDZ: located at 40°39'N, 117°07'E and 293.3 m amsl) was continuously observed from 2006 to 2017, and were controlled by the National Meteorological Information Center, China Meteorological Administration. According to the Technical Regulation on Ambient Air Quality Index of China (the Ministry of Environmental Protection of China, 2012), the maximum daily average 8 h concentration of ozone (MDA8) was used to represent the daily $O_3$ conditions. The MDA8 was calculated as the maximum of the running 8 h mean $O_3$ concentrations during 24 hours in the day. However, the systematic observation duration of the surface $O_3$ concentration was much shorter than the meteorological measurements and could not support the climate analysis.

The monthly sea ice concentrations (1°×1°) were downloaded from the Met Office Hadley Centre (Rayner et al. 2003), which are widely used in sea ice-related analysis. The sea ice fields are made more homogeneous by compensating satellite microwave-based sea ice concentrations for the impact of surface melt effects on retrievals in the Arctic, and by making the historical in situ concentrations consistent with the satellite data. The gridded sea ice data was available from 1870 to date, and those during 1979 to 2018 were extracted here.

The 1°×1° ERA-Interim data used here included the geopotential height (Z), zonal and meridional wind, relative humidity, vertical velocity, air temperature from 1000 hPa to 100hPa, boundary layer height (BLH), surface air temperature (SAT) and wind, downward UV radiation, downward solar radiation, low and medium cloud cover and precipitation (Dee et al. 2011). The daily mean and monthly mean ERA-Interim data from 1979 to present were directly downloaded from the ERA-Interim website in this study. Furthermore, the daily mean and monthly reanalysis datasets supported by the National Oceanic and

Atmospheric Administration were also employed and denoted as NCEP/NCAR (National Center for Environmental Prediction and the National Center for Atmospheric Research) data. The 2.5°×2.5° geopotential height (Z), zonal and meridional wind, relative humidity, vertical velocity, air temperature at from 1000 hPa to 100hPa, SAT and wind, downward UV radiation, downward solar radiation, low and medium cloud cover were downloaded, which was available from 1948 to present (Kalnay et al. 1996). The BLH dataset was only available from 1979 to 2014 in the website of the NOAA-CIRES 20th Century Reanalysis version 2c (Giese et al., 2016). The daily precipitation data was from the CPC global analysis of the daily precipitation dataset (Chen et al., 2008).

The EU pattern is a major teleconnection pattern in the Northern Hemisphere and appears in all seasons. Wang and Zhang (2015) used the method defined by Wallace and Gutzler (1981) to calculate the EU pattern index in winter and pointed out that the positive EU phase is associated with a cold-dry climate in East China, and *vice versa*. Meanwhile, Wang and He (2015) regarded the summer EU pattern as the main reason for the severe summer drought in North China in 2014. Considering the seasonal change of the EU pattern's location, the calculation procedure for the summertime EU index was consistent with that in Wang and He (2015), i.e., Eqn.1.

$$EU\ index = [-1 \times \overline{H500}_{(70-80°N, 60-90°E)} + 2 \times \overline{H500}_{(45-55°N, 90-110°E)}$$

$$-1 \times \overline{H500}_{(35-45°N, 120-140°E)}]/4 \qquad (1)$$

where H500 represents the geopotential height at 500 hPa, and overbars denote the area average.

The generalized additive model, a data-driven method, is particularly effective at handling the complex nonlinear and non-monotonous relationships between the dependent variable and the independent variables (Hastieand Tibshirani, 1990). This approach used a smoothing function, determined by the independent variables themselves, to transform the expressions, and addressed the dependent variable with different probability distributions by the link function. To verify the connection between the Arctic sea ice and the $O_3$ pollution, the Community Atmosphere Model version 5.3 (CAM5, Meehl and Washington 2013) was employed to design numerical experiments. The spatial resolution employed was 0.9°×1.25°, with 30 vertical hybrid sigma-pressure levels. CAM5.3 uses vertical hybrid δ-pressure coordinates including 26 layers with the top located at about 3.5 hPa. The climatological mean sea surface temperature and sea ice taken from the Hadley Centre were used to force the control run.

**3 Summer ozone pollution and associated weather conditions**

Due to increased surface $O_3$ pollution in China, the number of $O_3$ measurement stations has dramatically increased since 2014 (Figure 1 a, c, e, g). During 2006–2014, $O_3$ concentrations were only observed in the most developed regions in China. Since

2015, $O_3$ concentrations have been measured in most areas in eastern China. $O_3$ concentrations in the high-mid latitudes were higher than those in the lower latitudes, which appeared to be separated by the Yangtze River. The $O_3$ concentrations in North China were already high in 2014; the summer mean MDA8 in North China was higher than 120 $\mu g/m^3$. Observations with maximum MDA8 higher than 265 $\mu g/m^3$ (i.e., the threshold of the severe surface $O_3$ pollution in China) existed in the south of Hebei Province and the north of Shandong Province (Figure 1a). Since that time, the $O_3$ polluted region has expanded. In 2017, the areas with summer mean MDA8 > 120 $\mu g/m^3$ were visibly enlarged. In North China, the summer mean MDA8 observations were larger than 150 $\mu g/m^3$, and the maximum MDA8 was nearly 265 $\mu g/m^3$. South of the Yangtze River, the $O_3$ concentrations were distinctly lower and decreased progressively towards the Pearl River Delta.

The time span of $O_3$ observations (i.e., 2015–2017 for most of the sites) limited the possibility of determining the role of climate variability in the interannual $O_3$ variations in North China. Thus, we examined the representativeness of the $O_3$ measurements at SDZ (one of the three regional background air-monitoring stations in China, with observations from 2006–2017). The correlation coefficients between SDZ MDA8 and the observed MDA8 at the other sites were calculated and are shown in Figure 1 (b, d, f, h). The distribution of correlation coefficients is similar to the MDA8 on Figure 1 (a, c, e, g). The SDZ MDA8 significantly covaried with the MDA8 in North China in summer. Along with the increasing of the surface $O_3$ pollution, the covariation of SDZ MDA8 and MDA8 in North China strengthens the representativeness of SDZ for North China. However, the correlation coefficients between SDZ MDA8 and MDA8 in the south of China were negative, indicating opposite variation (Zhao and Wang, 2017). The variation of summer SDZ MDA8 is presented in Figure S1. According to the Technical Regulation on Ambient Air Quality Index in China (The Ministry of Environmental Protection of China, 2012), we defined non-$O_3$ polluted (NOP) level at surface as the $O_3$ concentration < 100 $\mu g/m^3$ and moderate-$O_3$ polluted (MOP) level with $O_3$ concentration > 215 $\mu g/m^3$, respectively. The upper and lower quartile of SDZ MDA8 was 188 $\mu g/m^3$ and 114 $\mu g/m^3$, indicating that more than 75% of summer days exceeded the NOP threshold even at the regional background air-monitoring station. During the years 2007–2017, there were 126 NOP days and 155 MOP days in summer at SDZ station. The maximum number of MOP days was 26 days in 2015, and the mean number of MOP days was 14 days (Table S1). Both of the interannual variation in MOP and NOP days was significant at the 95% confidence level, without an obvious trend.

Due to the significant covariation between the SDZ MDA8 to the MDA8 in North China, the meteorological conditions were composited for the MOP and NOP days in SDZ (Figure 2) and the results were also appropriate for those in North China. The local and surrounding weather conditions were significantly different (t-test). The anomalous southerlies (Figure 2a), higher BLH (Figure 2c), less rainfall (Figure 2e), warmer surface air temperature, and cooler temperature in the high troposphere (Figure 2g) favored surface $O_3$ pollution. Near surface, for the polluted conditions, the winds are northward in North China, due to cyclonic anomalies to the west and anticyclonic flow to the east (Figure S2a). Anomalous southerlies from the Yangtze River transported $O_3$ precursors (that were emitted in the economically developed Yangtze River Delta) and

superposed them with the local high emissions in North China (Figure 2a). When the anomalous winds reversed, i.e., northerlies, the $O_3$ precursors in North China were dispersed, and the surface $O_3$ concentration in North China was reduced (Figure 2b). On the upper level, significant anticyclonic anomalies (Figure S2c) resulted in sunny days in summer. A day without rain represents efficient solar radiation, in favor of the occurrence of surface $O_3$ pollution (Figure 2e). In contrast, the cloudy skies and precipitation weakened the photochemistry by influencing exposure to ultraviolet rays. In addition, precipitation was also an important indicator of the wet removal efficiency (Figure 2f). High SAT enhanced the photochemical reactions and resulted in higher surface $O_3$ concentrations (Figure 2 g). Differently from the SAT, the temperature at 200 hPa above North China was significantly negative (Figure 2 g), dynamically associated with the upper-level anticyclone. Furthermore, due to the strengthening of solar radiation, the near-surface turbulence was enhanced, and the boundary layer was lifted (Figure 2c). The entrainment of atmospheric ozone from the upper air into the boundary layer enhanced the surface $O_3$ concentration (An et al., 2009). To confirm the robustness of the link between meteorological conditions and the MOP and NOP days over North China, the above composite analysis was repeated with NCEP/NCAR reanalysis data, and identical results were obtained (Figures S3, S4).

To assess the interannual variation of surface $O_3$ pollution and its relationship with climate variability (Cai et al., 2017), we fitted an $O_3$ weather index (OWI) based on long-term meteorological observations. Firstly, the regional average meteorological elements were calculated as meteorological indexes (I), and here the selected regions determined on the most significantly different areas in the composites of MOP and NOP events in Figure 2. Then, we defined the OWI as Eqn. 2.

OWI=normalized V10mI+normalized BI–normalized PI+normalized DTI     (2)

Where the V10mI is area-averaged meridional wind at 10 m (35–50 °N, 110–122.5°E, black box in Figure 2a), and its correlation coefficient with the SDZ $O_3$ concentration was 0.39. BI indicated area-averaged BLH (37.5–47.5 °N, 112.5–120 °E, black box in Figure 2c), and correlation coefficient with the SDZ $O_3$ was 0.40. The PI defined as area-averaged precipitation (37.5–42.5 °N, 112–127.5 °E, black box in Figure 2e), whose correlation coefficient with the SDZ $O_3$ concentration was –0.35 (above the 99% confidence level). DTI represents the area-averaged difference in the temperature at the surface and 200 hPa (SAT minus temperature at 200 hPa, 37.5–47.5 °N, 110–122.5 °E, black box in Figure 2g), and the correlation coefficient with SDZ $O_3$ concentration was 0.49.

For comparison, the multiple regression equation was built between the MDA8 and associated weather indices (Figure 3). Our analysis indicated that the observed MDA8 was well fit by the multiple regression equation (Figure 3). The correlation coefficient was 0.61 between the fit and daily measured MDA8 during 2007–2017 (i.e., 92 days × 11 years). The correlation coefficient between the observed MDA8 and daily OWI was also 0.61 for the 11 year period. Thus, the OWI was easily constructed by accumulating the normalized weather index and was selected to represent the variation in surface $O_3$ pollution.

A total of 90.3% of the MOP events were in the range of OWI > 0, and correspondingly, 90.5% of the NOP events were linked with OWI < 0 (Figure 4). The correlation coefficients between the OWI and observed MDA8 at the other sites were calculated (Figure 5). The significantly positive correlations were distributed in North China (Figure 5 b-d). Thus, it is reasonable to analyse the variation in surface $O_3$-related atmospheric circulations in North China using the OWI, which also extends the study period to the historical period before 2007 and the projected future.

**4 Impacts of EU pattern on the interannual variation of surface ozone**

After the assimilation of satellite data, possible in 1979, the quality of reanalysis data improved. Here, the daily OWI was calculated with both ERA-Interim and NCEP/NCAR reanalysis data from 1979. According to the above analysis, the daily OWI could largely represent the variation in MDA8 in North China. The monthly OWI was computed as the monthly mean of the daily OWI. During 2007–2017, the constructed JJA (June-July-August) mean OWI varied similarly with the observed

MDA8 and captured the extremes (Figure 6). Although the range of the SDZ MDA8 was 2006–2017, only the data from 2007 to 2017 were used in the above OWI construction processes. Thus, the datasets in 2006 were independent samples (i.e., test set), and could verify the performance of the OWI. The JJA mean OWI in 2006 successfully reflected the variation in observed MDA8, confirming the robustness of the OWI. Derived from two different reanalysis datasets, the OWI-ERA and OWI-NCEP varied consistently. The above independent verifications proved that the performance of the summer OWI did not depend on

the specific reanalysis data. In the following study, the monthly OWI from ERA-interim data and associated physical mechanisms were analysed. During mid-1980s to the mid-1990s, the OWI was below zero, with a slightly decreasing trend and insignificant interannual variation. Since then, the OWI has increased; furthermore, the intensity of interannual variation has strengthened. The emissions of $O_3$ precursors increased persistently and linearly due to the steady economic development after 1978 in China (Wang 2017). The strong interannual variation in the OWI after mid-1990s, representing the impacts of

meteorological conditions on $O_3$ concentrations, contributed to the interannual fluctuations of the surface $O_3$ pollution. Thus, the impacts of the large-scale atmospheric circulations on the summer $O_3$ pollution, specially the related OWI, were analyzed.

The atmospheric circulations associated with summer mean OWI, indicated by the correlation coefficients, are displayed in Figure 7. In the mid-upper troposphere, cyclonic and anticyclonic anomalies were alternately distributed over the north-central Siberian Plateau (–), North China and Mongolia (+), and the Yellow Sea and Japan Sea (–) (Figure 7a). These three

atmospheric centres, propagated from the polar region to the mid-latitudes, appeared to be the positive phase of EU pattern (Wang and He 2015). This Rossby wave-like train, i.e., the EU pattern, could also be recognized in the surface air temperature. The correlation coefficient between the EU pattern index and OWI was 0.44 (after detrending and above the 99% confidence level), indicating that the strengthening of the EU positive phase contributed to the severe surface $O_3$ pollution in North China. More precisely, the positive phase of EU pattern could modulate the local meteorological conditions to enhance the

photochemical reactions. The EU pattern is considered to be the main reason for the variability of the severe drought in North China, i.e., resulting in hot and dry climate extremes (Wang and He, 2015). To a certain extent, the severe drought environment promoted the formation of surface ozone. After 2007, the EU index and the observational SDZ MDA8 showed good agreement (Figure 8). More than 80% of the SDZ MDA8 anomalies showed the same mathematical sign as the anomalous EU pattern index. Furthermore, the large EU pattern anomalies (i.e., the |EU pattern index| > 0.8 ×its standard deviation) always induced

in-phase surface ozone pollution.

Under barotropic anticyclonic circulation over North China, i.e., one of the active centres of the positive EU pattern, the significant descending air flows indicated efficient adiabatic heating (resulting in high temperatures near the surface) and dry air (i.e., less cloud cover) below 300 hPa (Figure 7c). Furthermore, over North China, the air temperature (relative humidity) anomalies were negative (positive) at 200 hPa but positive (negative) below 300 hPa (Figure 7c). The barotropic anticyclonic

circulation associated with surface ozone pollution (Figure 7b) was similar to the positive EU pattern (Figure 7c) and led to sunny days, i.e., hot temperatures (Figure 7a), strong downwards solar radiation and UV radiation (Figure 9c–d), less low and medium cloud cover (Figure 9d), and dry conditions (Figure 9b–c). Without the cover of low and medium clouds, the short wave solar radiation, especially the UV radiation, penetrated straight to the land surface. The photochemical reaction of the $O_3$ precursor was enhanced, generating more $O_3$ near the surface. The dry atmosphere near the surface, i.e., less precipitation and

lower relative humidity, accelerated the photochemical reaction but restricted the wet clearing of the stocked $O_3$ in the atmosphere. A higher BLH (Figure 9b), resulting from the strengthening of solar radiation, likely facilitated the downward transportation of $O_3$ from aloft. Near the surface, the western part of these anticyclonic anomalies manifested as significant southerlies (Figure 9a), which transported the $O_3$ precursors from the economically developed Yangtze River Delta. The extraneous $O_3$ precursor, superposed with local emissions, supported efficient photochemical production of $O_3$. To confirm the

robustness of the atmospheric circulations and associated physical mechanisms, the above analysis was repeated with the NCEP/NCAR data and identical results were obtained (Figure S5–S6). The correspondence between large-scale EU teleconnection and anti-cyclonic circulations were clear. Local meteorological conditions, such as hot land surface (Figure S5), violet solar radiation (Figure S6c–d), clear sky (Figure S6d), less precipitation (Figure S6c) and lower relative humidity (Figure S6b) were also clearly recognized. Thus, the impacts of the atmospheric circulations were confirmed by both the

ERA-Interim and NCEP/NCAR data, i.e., the analyses and conclusions were independent of data sets.

**5 Roles of the Arctic sea ice**

The positive EU pattern enhanced the local anticyclonic circulation over North China and facilitated the photochemical processes leading to the formation of surface ozone. The EU pattern originated from the Arctic region. The preceding sea ice anomalies could stimulate atmospheric responses like the EU pattern in summer (Wang and He, 2015) Thus, the role of Arctic

sea ice on the OWI was also explored in this study. The correlation between the sea ice and JJA OWI was evaluated each month (Figure omitted), and we found the interannual variation of OWI was significantly correlated with May sea ice conditions to the north of Eurasia, especially near the Gakkel Ridge, the Canada Basin and the Beaufort Sea (Figure 10a). The averaged (green boxes in Figure 10a) sea ice (SI) area in May was calculated as the SI index, whose linear correlation coefficient with JJA OWI was 0.67 (after detrending) from 1979 to 2017. During 2007–2017, 73% of the May SI anomalies are followed by observational SDZ MDA8 anomalies with the same mathematical sign (Figure 10b). Furthermore, the linear and nonlinear relationships were both introduced using the generalized additive model (Figure 11), and the contribution of May sea ice to the interannual variability of OWI was approximately 60%.

These positive sea ice anomalies could induce EU pattern responses in the subsequent summer (Figure 10c). The excited atmospheric and thermal centres were located over the Central Siberian Plateau, North China and Mongolia, and the Yellow Sea. Similarly, the local meteorological responses, such as anomalous southerlies and less precipitation (Figure 10d), less cloud and strong solar radiation (Figure 10e) were also closely connected with the positive sea ice anomalies in May. Thus, the preceding May sea ice positively modulated the EU pattern, and then, this Rossby wave train transported the impacts from the polar region and strengthened the anti-cyclonic anomalies over North China. Finally, suitable meteorological conditions, including hot-dry air, anomalous southerlies and intense sunshine, were induced to intensify the photochemical production of surface ozone pollution. To confirm the roles of Arctic sea ice and associated physical mechanisms, the above analysis was repeated with the NCEP/NCAR data, and identical results were obtained (Figure S7).

The causality, i.e., the preceding May sea ice anomalies contributing to the subsequent JJA OWI in North China, was also confirmed by CAM5. During the control experiment (CTRL), the CAM5 model was first integrated for 20 years with climate mean initial and boundary conditions. Next, the data in 1st September of the last 5 years (i.e., 16–20 years) were designated as five slightly different initial conditions. With each initial condition, the CAM5 model integrated for 10 years. The JJA mean results of the last 6 years (i.e., 6 years $\times$ 5 groups = 30 ensembles) were employed as the output of the CTRL. On the basis of CTRL, the May sea ice concentration in the two boxes of Figure 10a was separately reduced by 10% (denoted as LowASI experiments), i.e., a total of 30 sensitivity runs. Similarly, the JJA mean results of the 30 sensitive runs were employed as the output of the LowASI. The differences (LowASI minus CTRL) represent the responses of atmospheric circulations and meteorological conditions to the declining May sea ice.

It was evident that an EU Rossby wave train was induced on the mid-troposphere (Figure 12a), which propagated from the Taymyr Peninsula (–), Northeast China (+), to east of China and the west Pacific (+). Under such large-scale atmospheric anomalies, the anomalies of relative humidity were significantly positive and resulted in denser low and cloud cover in North China (Figure 12d). Furthermore, the cover of cloud efficiently prevented the solar radiation from reaching the land surface,

meanwhile, cooled the air in the boundary layer (Figure 12c). Without hot-dry air and intense sunshine, the photochemical

production was significantly decelerated and the generation of surface $O_3$ was rather weak. Additionally, sufficient moisture

and clouds caused more rainfall (Figure 12c). The wet deposition effect might be enhanced. Thus, corresponding to less

Arctic sea ice in May, the photochemical process to generate $O_3$ was weakened, and the wet deposition effect to decrease $O_3$

was enhanced. That is, the positive relationship and associated physical mechanisms (i.e., climate links among ASI, EU

pattern and summer surface ozone pollution in North China) were causally verified.

## 6 Conclusions and discussions

Recently, the summer surface $O_3$ concentrations and the number of $O_3$ observation stations have steadily increased in China. In

general, the $O_3$ concentrations in North China were substantially higher than those in South China. To reveal the climatic driver

of summer surface $O_3$ pollution in North China, a daily OWI (i.e., surface $O_3$ weather index) was constructed based on

meteorological and ozone observations. The robustness of this index (i.e., OWI) was verified by the ERA-Interim and

NCEP/NCAR reanalysis datasets and surface $O_3$ measurements. May Arctic sea ice was found to be a preceding and efficient

climatic driver, which may help for seasonal forecasting. In the historical period, variation in Arctic sea ice can explain

approximately 60% of the interannual variability of the summer OWI in North China, which was closely associated with the

surface $O_3$ pollution. Currently, the Arctic region has been warming approximately twice as much as the global average

(Huang et al., 2017; Zhou, 2017), indicating accelerated change in the sea ice. Thus, understanding the role of Arctic sea ice

may contribute to the understanding of seasonal variability of $O_3$ pollution.

The EU pattern acted as an atmospheric bridge to link May Arctic sea ice and the summer surface $O_3$ pollution in North China.

The accumulated sea ice in May could induce the positive EU phase. The anticyclonic circulation over North China, i.e., one of

the active centres of the EU pattern, was connected with high surface temperature, strong downward solar radiation, less low-

and medium-altitude cloud cover, and drought over North China. Under such local meteorological conditions, the

photochemical reactions to produce surface $O_3$ were supported. Generally, these anticyclonic anomalies over North China

were barotropic and could persist for a long time; thus, the processes that produce surface $O_3$ were continuous to achieve a high

concentration. The connections revealed in this study were based on long-term meteorological measurements and was causally

verified by well-designed numerical experiments.

In order to extend the time range of this study, the OWI was constructed in North China. Although the feasibility of the

construction approach was strictly examined, the OWI was still a substitution focusing on the impacts of the weather

conditions. When discussing the impacts of atmospheric circulations, the linear trend was removed to weaken the signal of

anthropogenic emissions. Thus, the results in this study concentrated on and emphasized the meteorological and climate

factors. However, there is no doubt that the polluted emissions are the fundamental inducement of the surface $O_3$ pollution.

The joint effects of the climate anomalies and the historical emissions should be studied using the numerical models in the future. The EU pattern was a well-known continental Rossby wave train and could link the mid-high latitude climate with the change of the Arctic. Although the connection between the Arctic sea ice and the ozone pollution was revealed, the separate roles of the sea ice near the Gakkel Ridge, and the Canada Basin and Beaufort Sea should be intensively studied in the future.


**Author contribution**

Yin Z. C. and Wang H. J. designed the research. Yin Z. C., Li Y. Y. and Ma X. H. performed research. Yin Z. C. and Zhang X. Y. analysed data. Yin Z. C. prepared the manuscript with contributions from all co-authors.

The authors declare no conflict of interest.

**Acknowledgments**

This research was supported by the National Key Research and Development Plan (2016YFA0600703), the National Natural Science Foundation of China (91744311 and 41705058), the Jiangsu innovation & entrepreneurship team, and the Priority Academic Program Development (PAPD) of Jiangsu Higher Education Institutions..

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

**Figure 1.** The distribution of the JJA mean MDA8 (a, c, e, g) and the correlation coefficients (b, d, f, h) between the daily MDA8 and SDZ MDA8 from 2014 to 2017. The black cross in panels a, c, e, and g indicate that the maximum daily MDA8 was larger than 265 µg/m³. The black cross in panels b, d, f, and h indicates that the CC was above the 95% confidence level. The green triangle in panel (a) illustrates the location of the Shangdianzi site.

**Figure 2.** Composite of the meteorological conditions associated with different $O_3$ events during 2007–2017. Results for MOP

(a, c, e, g) and NOP (b, d, f, h) events included (a–b) surface wind (m/s, arrow) and v-wind (m/s, shading), (c–d) BLH (m), (e–f) precipitation (mm), (g–h) SAT (°C, shading), and temperature at 200 hPa (°C, contour). The black dots denote the composite results passed the 95% confidence level. The boxes represent the area used to calculate OWI. These composites were calculated using the ERA-Interim dataset. The green triangle in panel (a-b) illustrates the location of the Shangdianzi site. The composite results were calculated as the differences between MOP or NOP events with the rest events (i.e., all events

but excluded MOP and NOP events).

**Figure 3.** The variation in the daily (a) observational SDZ MDA8 (black), (b) fitting SDZ MDA8 (red), and (c) OWI (blue) from June to August during 2007–2017. The numbers are the correlation coefficients between the observational SDZ MDA8 and fitting SDZ MDA8 (red) and OWI (blue).

**Figure 4.** The OWI for MOP (red) and NOP (blue) events during 2007–2017.

**Figure 5**. The correlation coefficients between the daily MDA8 and OWI from 2014 to 2017. The black crosses indicate that the CC was above the 95% confidence level.

**Figure 6.** The variation in the JJA mean observed SDZ MDA8 (green) from 2006 to 2017, OWI calculated from ERA-interim datasets during 1979–2017 (blue) and OWI calculated from NOAA datasets (red) during 1979–2014.

**Figure 7.** The associated atmospheric circulation. (a) The correlation coefficients between the JJA mean OWI and surface air

temperature (shading), wind (arrow) at 200 hPa and geopotential height at 500 hPa (contour) from 1979 to 2017. The black dots indicate that the CC with surface air temperature was above the 95% confidence level. The cross-section (110°–125°E mean) correlation coefficients between JJA mean OWI (a), EU pattern index (b) and relative humidity (shading), temperature (contour), wind (arrow, vertical speed multiplied by 100) from 1979 to 2017. The black dots indicate that the CC with relative humidity exceeded the 95% confidence level (t test). The data used here are ERA-Interim datasets.

Figure 8. The variation in the JJA mean observational SDZ MDA8 (µg/m³, blue) and EU index (gpm, red) from 2007 to 2017.

**Figure 9.** The associated meteorological conditions. (a) The correlation coefficients between the JJA mean OWI and v wind at 10 m (shading), surface wind (arrow), (b) relative humidity near the surface (shading), boundary layer height (contour), (c) precipitation (shading), downward UV radiation at the surface (contour), (d) downward solar radiation at the surface

(shading), sum of low and medium cloud cover (contour) from 1979 to 2017. The black dots indicate that the CC with temperature was above the 95% confidence level. The contours plotted in panel (b–d) exceeded the 95% confidence level. The data used here are ERA-Interim datasets.

**Figure 10.** The role of the Arctic sea ice. (a) The correlation coefficients between the JJA mean OWI and May sea ice, (b) The variation of the May SI index (red bar, area-averaged sea ice of the green boxes in panel a), JJA mean EU pattern index

(blue bar) and JJA mean observational SDZ MDA8 (black bar) from 2007 to 2017. (c) The correlation coefficients between the May SI index and surface air temperature (shading), geopotential height at 500 hPa (contour) from 1979 to 2017. The black dots indicate that the CC with surface air temperature was above the 95% confidence level. (d) The correlation coefficients between the May SI index and precipitation (shading), surface wind (arrow), (e) downward UV radiation at the surface (shading) and sum of low and medium cloud cover (contour) from 1979 to 2017. The black dots indicate that the

shading CC with precipitation (d) and downward UV radiation (e) was above the 95% confidence level. The data used here

are ERA-Interim datasets.

**Figure 11.** The variation in the observational OWI (black) and the fitted OWI by the generalized additive model (red) from 1979 to 201

**Figure 12**. Composite results of the LowASI experiments (LowASI minus Ctrl) by the CAM5 model: (a) geopotential height at 500 hPa, (b) preciptation, (c) net radiative flux at the top of the atmosphere (shading) and temperature at 925 hPa (contour), and (d) sum of low and medium cloud fraction (shading) and relative humidity at 925 hPa (contour). The black hatching denotes the differences with shading were above the 95% confidence level (t-test).






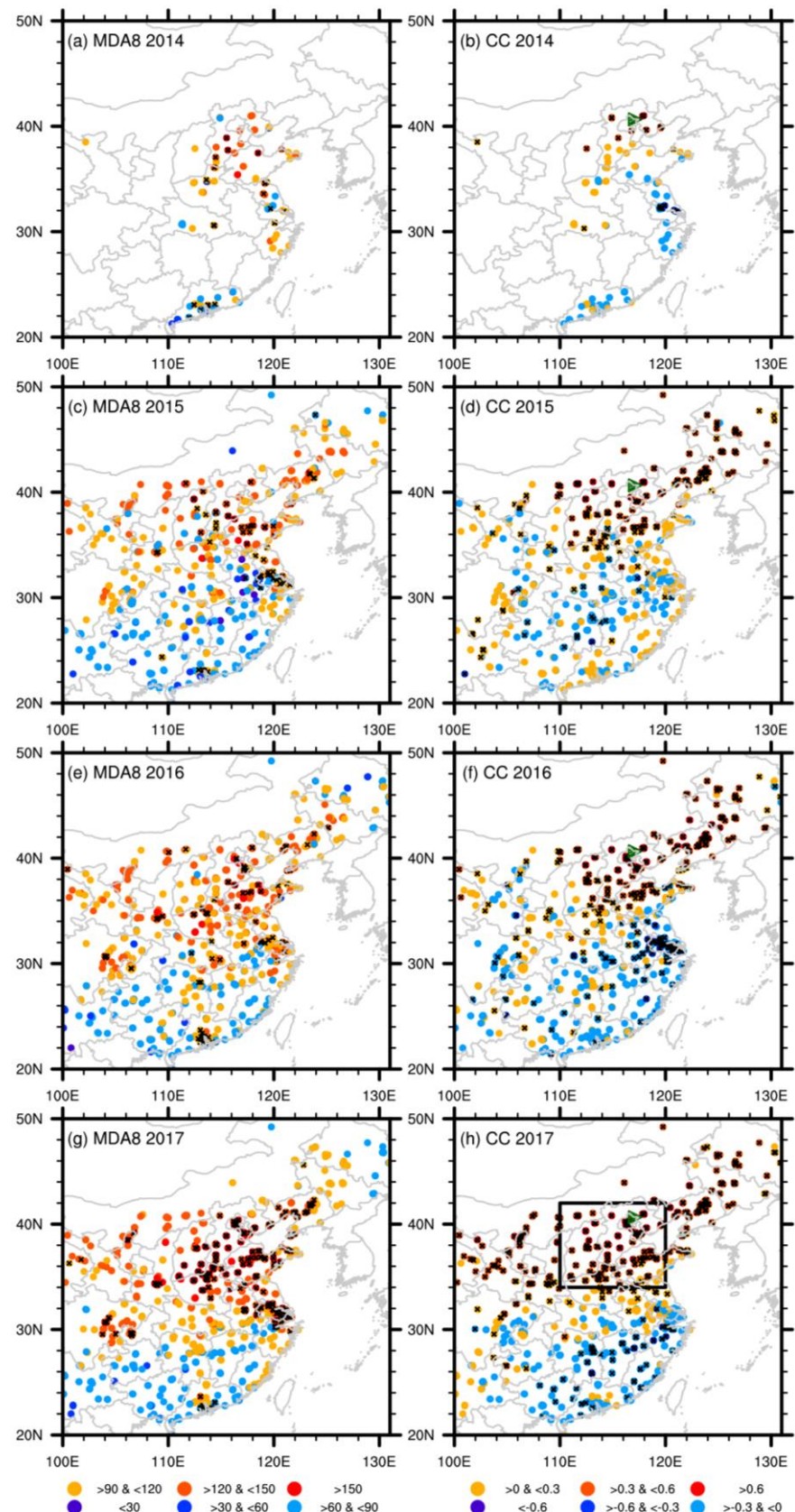

Figure 1. The distribution of the JJA mean MDA8 (a, c, e, g) and the correlation coefficients (b, d, f, h) between the daily MDA8 and SDZ MDA8 from 2014 to 2017. The black cross in panels a, c, e, and g indicate that the maximum daily MDA8 was larger than 265 μg/m³. The black cross in panels b, d, f, and h indicate that the CC was above the 95% confidence level. The green triangle in panels b, d, f, and h illustrate the location of the SDZ station. The black box in panel h is the range of North China.

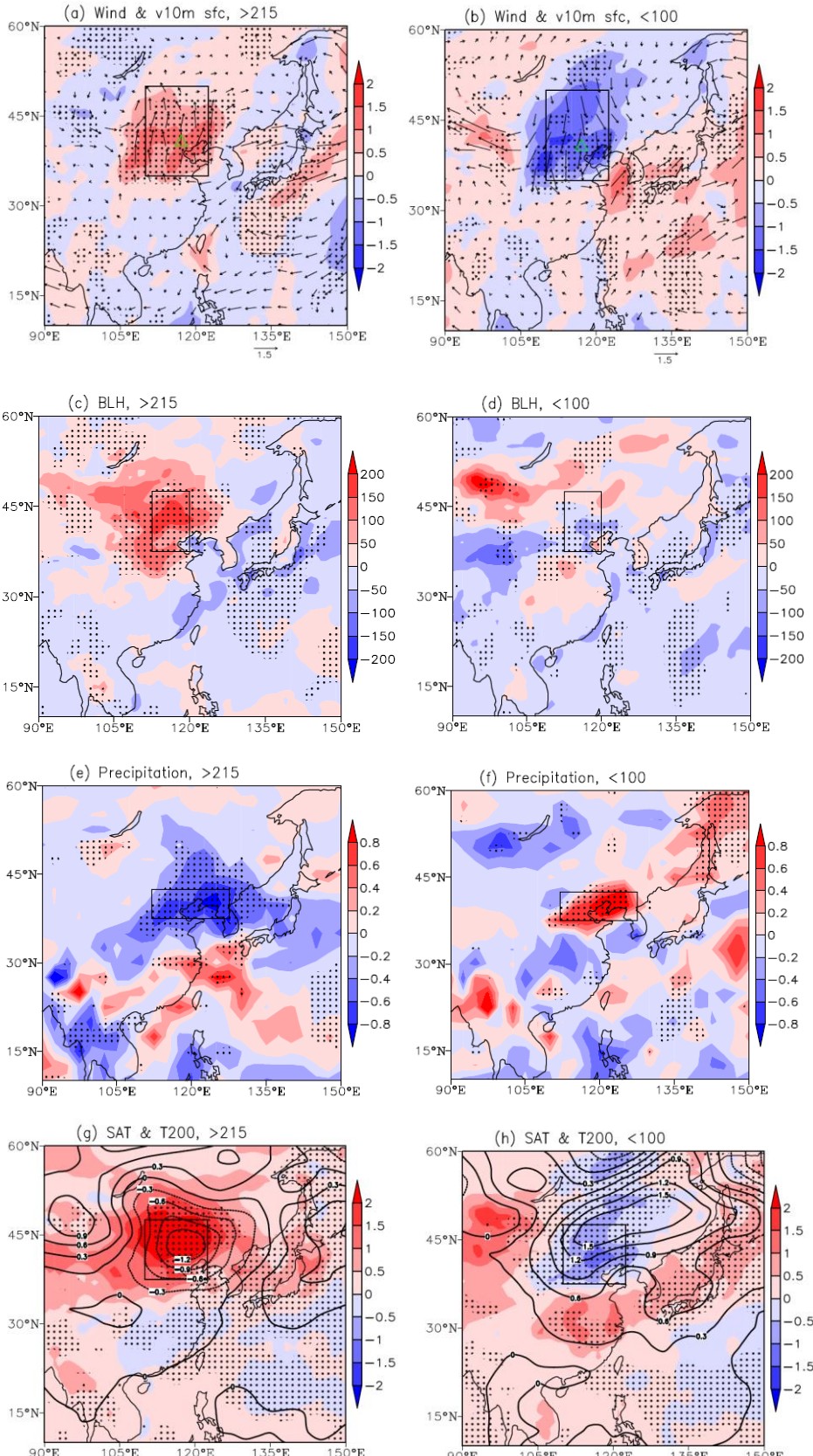

Figure 2. Composite of the meteorological conditions associated with different O₃ events during 2007–2017. Results for MOP (a, c, e, g) and NOP (b, d, f, h) events included (a–b) surface wind (m/s, arrow) and v-wind (m/s, shading), (c–d) BLH (m), (e–f) precipitation (mm), (g–h) SAT (℃, shading), and temperature at 200 hPa (℃, contour). The black dots denote the composite results passed the 95% confidence level. The boxes represent the area used to calculate OWI. These composites were calculated using the ERA-Interim dataset. The green triangle in panel (a-b) illustrates the location of the Shangdianzi site. The composite results were calculated as the differences between MOP or NOP events with the rest events (i.e., all events but excluded MOP and NOP events).

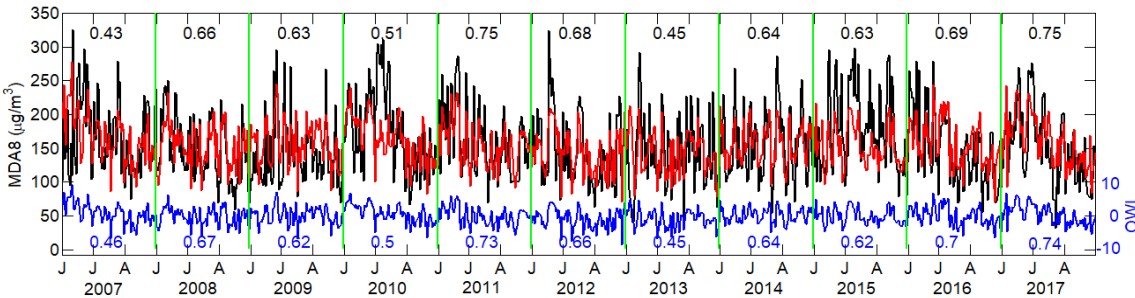

Figure 3. The variation in the daily (a) observational SDZ MDA8 (black), (b) fitting SDZ MDA8 (red), and (c) OWI (blue) from June to August during 2007–2017. The numbers are the correlation coefficients between the observational SDZ MDA8 and fitting SDZ MDA8 (red) and OWI (blue).

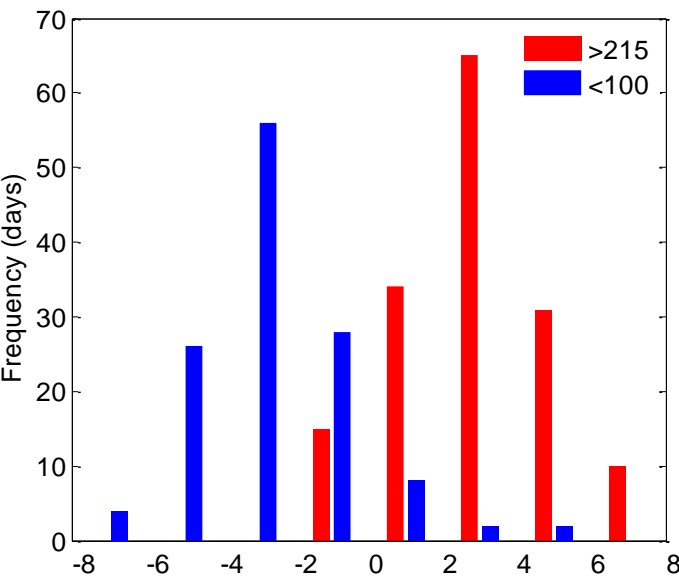

Figure 4. The OWI for MOP (red) and NOP (blue) events during 2007–2017.

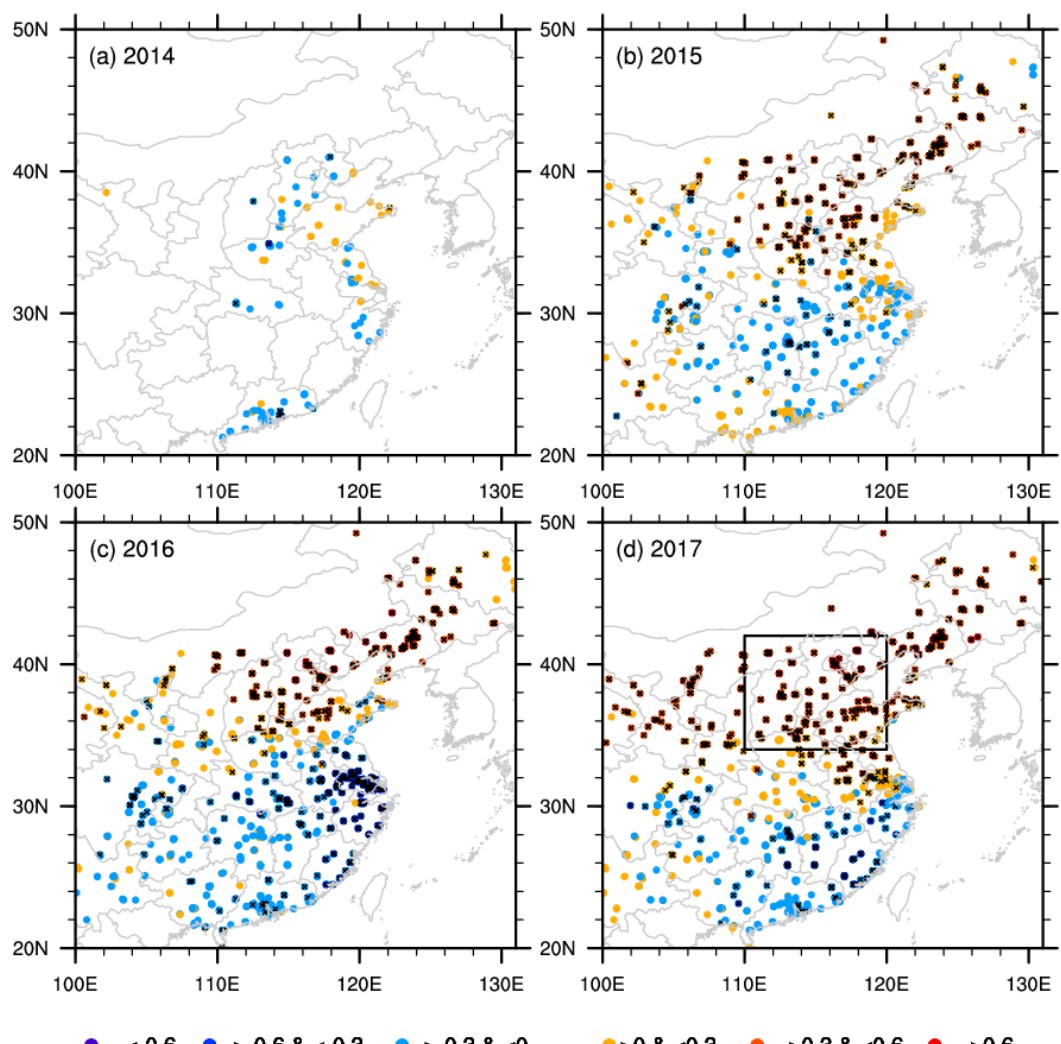

Figure 5. The correlation coefficients between the daily MDA8 and OWI from 2014 to 2017. The black crosses indicate that the CC was above the 95% confidence level. The black box in panel d is the range of North China.

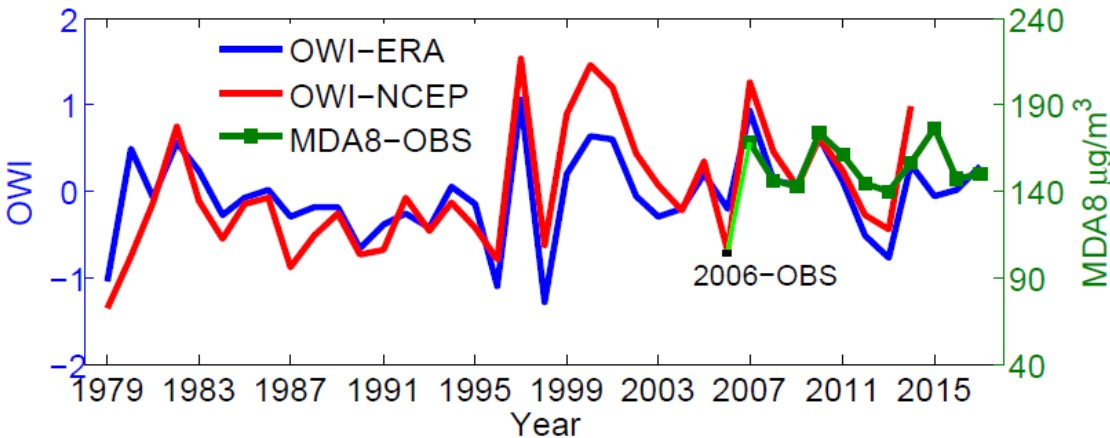

Figure 6. The variation in the JJA mean observed SDZ MDA8 (green) from 2006 to 2017, OWI calculated from ERA-interim datasets during 1979–2017 (blue) and OWI calculated from NCEP/NCAR datasets (red) during 1979–2014.

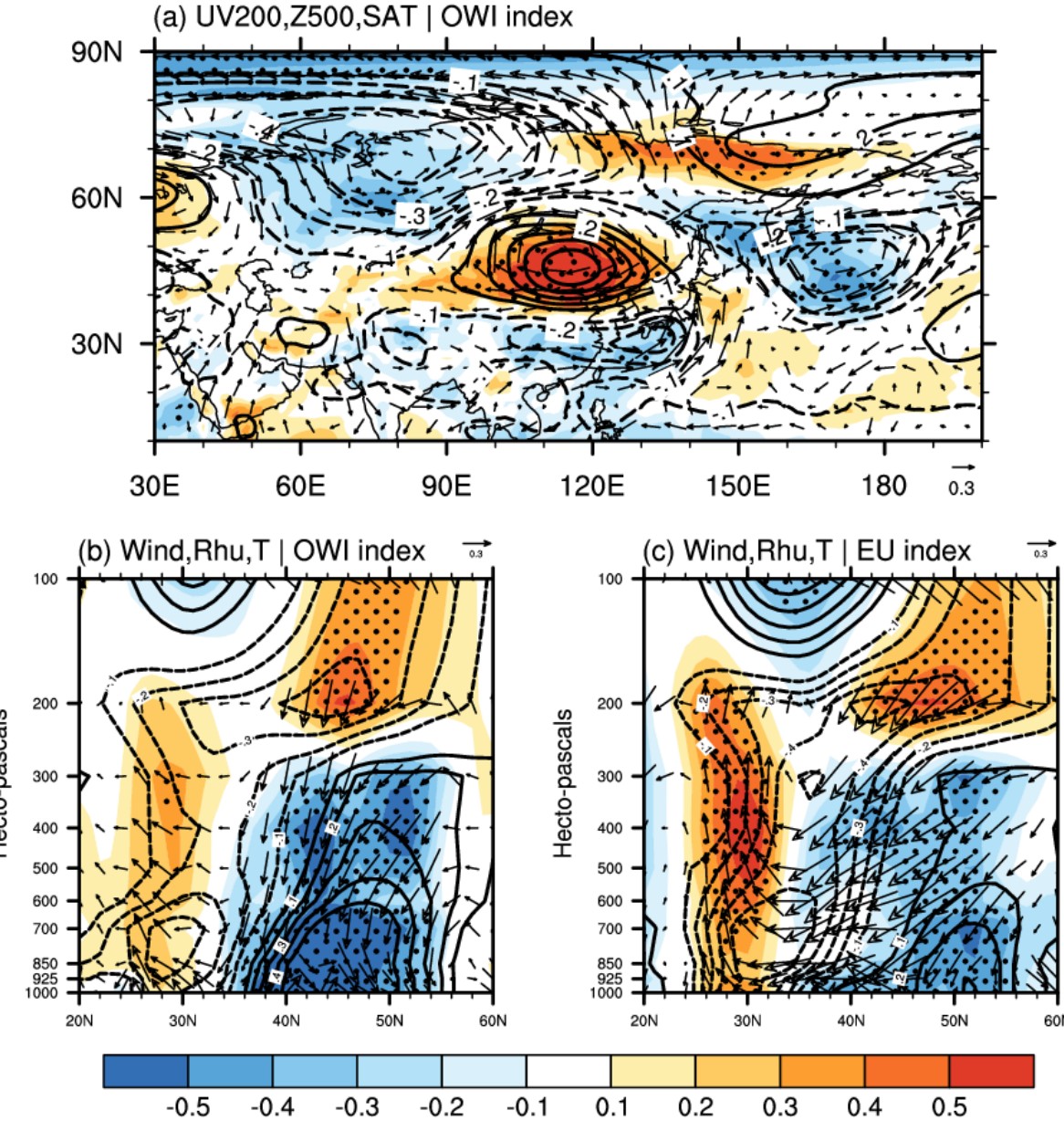

Figure 7. The associated atmospheric circulation. (a) The correlation coefficients between the JJA mean OWI and surface air temperature (shading), wind (arrow) at 200 hPa and geopotential height at 500 hPa (contour) from 1979 to 2017. The black dots indicate that the CC with surface air temperature was above the 95% confidence level. The cross-section (110 °–125 °E mean) correlation coefficients between JJA mean OWI (a), EU pattern index (b) and relative humidity (shading), temperature (contour), wind (arrow, vertical speed multiplied by 100) from 1979 to 2017. The black dots indicate that the CC with relative humidity exceeded the 95% confidence level (t test). The data used here are ERA-Interim datasets.

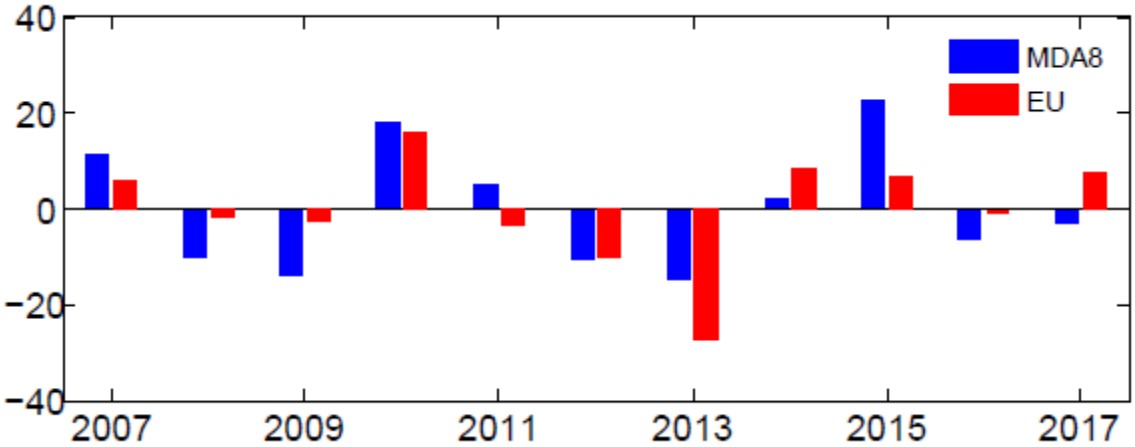

Figure 8. The variation in the JJA mean observational SDZ MDA8 (µg/m³, blue) and EU index (gpm, red) from 2007 to 2017.

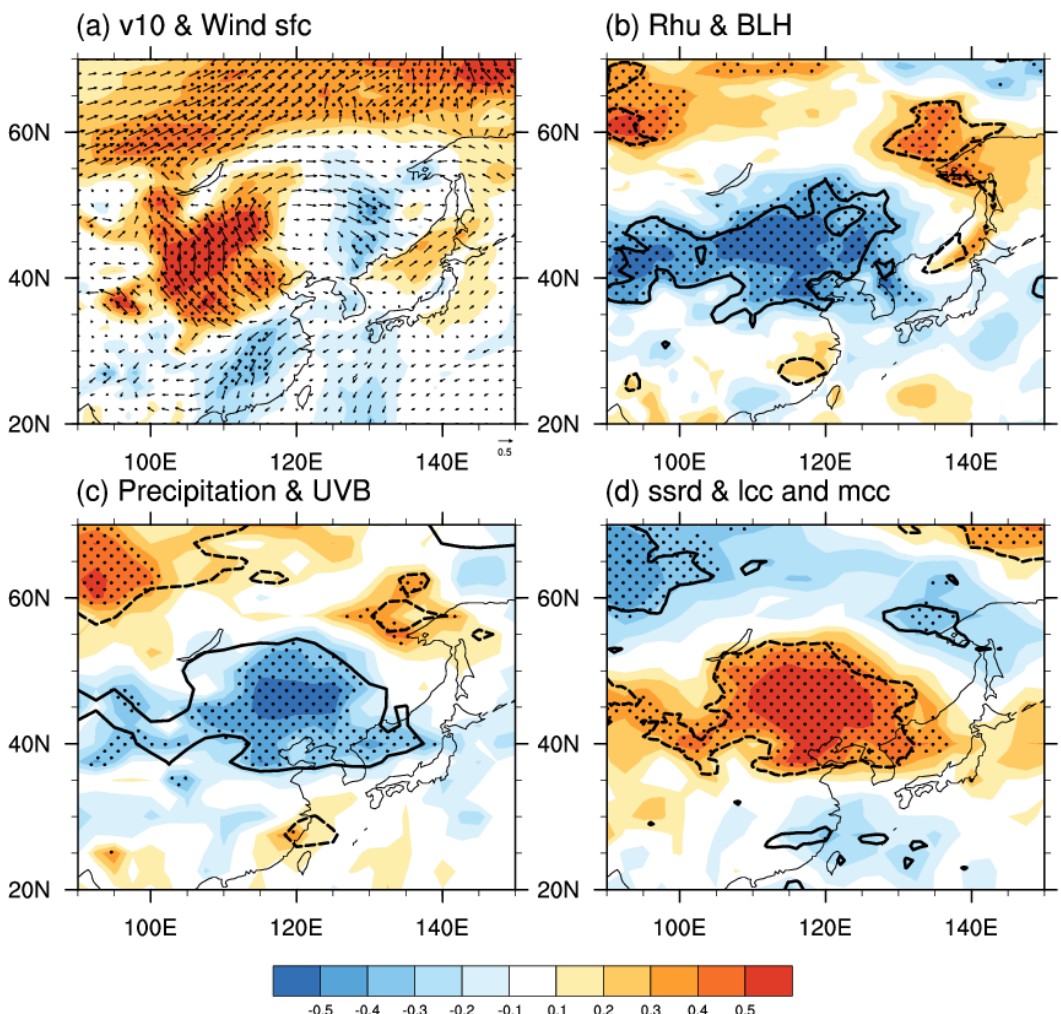

Figure 9. The associated meteorological conditions. (a) The correlation coefficients between the JJA mean OWI and v wind at 10 m (shading), surface wind (arrow), (b) relative humidity near the surface (shading), boundary layer height (contour), (c) precipitation (shading), downward UV radiation at the surface (contour), (d) downward solar radiation at the surface (shading), sum of low and medium cloud cover (contour) from 1979 to 2017. The black dots indicate that the CC with temperature was above the 95% confidence level. The contours plotted in panel (b–d) exceeded the 95% confidence level. The data used here are ERA-Interim datasets.

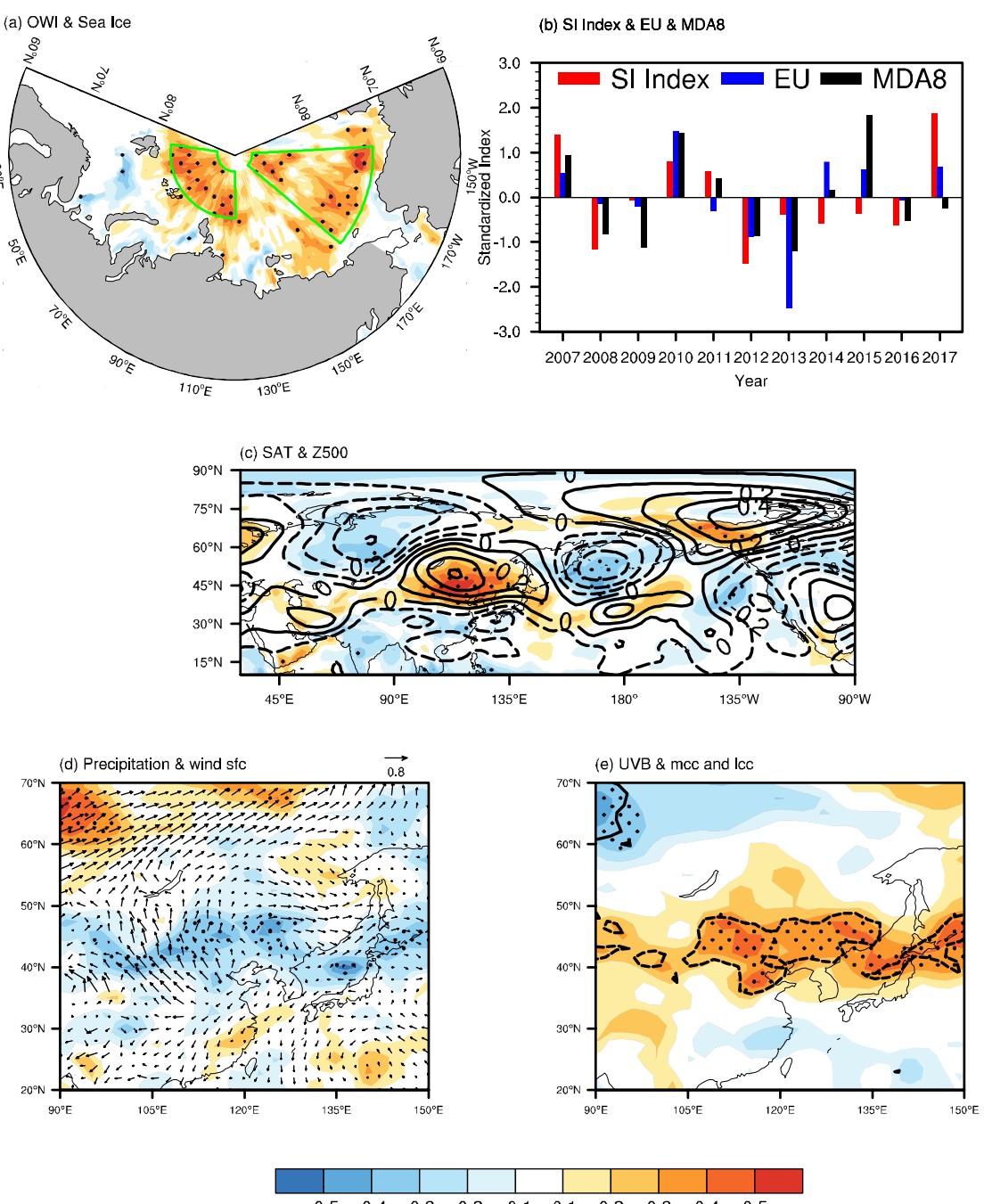

Figure 10. The role of the Arctic sea ice. (a) The correlation coefficients between the JJA mean OWI and May sea ice, (b) The variation of the May SI index (red bar, area-averaged sea ice of the green boxes in panel a), JJA mean EU pattern index (blue bar) and JJA mean observational SDZ MDA8 (black bar) from 2007 to 2017. (c) The correlation coefficients between the May SI index and surface air temperature (shading), geopotential height at 500 hPa (contour) from 1979 to 2017. The black dots indicate that the CC with surface air temperature was above the 95% confidence level. (d) The correlation coefficients between the May SI index and precipitation (shading), surface wind (arrow), (e) downward UV radiation at the surface (shading) and sum of low and medium cloud cover (contour) from 1979 to 2017. The black dots indicate that the shading CC with precipitation (d) and downward UV radiation (e) was above the 95% confidence level. The data used here are ERA-Interim datasets.

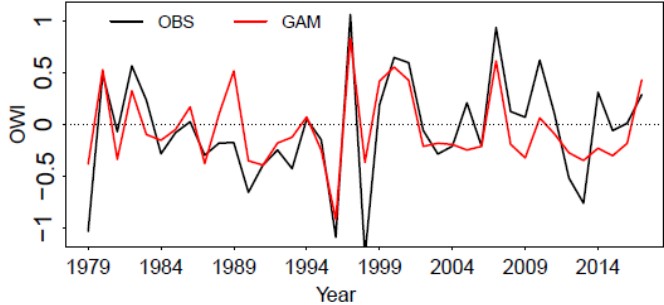

565    Figure 11. The variation in the observational OWI (black) and the fitted OWI by the generalized additive model (red) from
1979 to 2017.

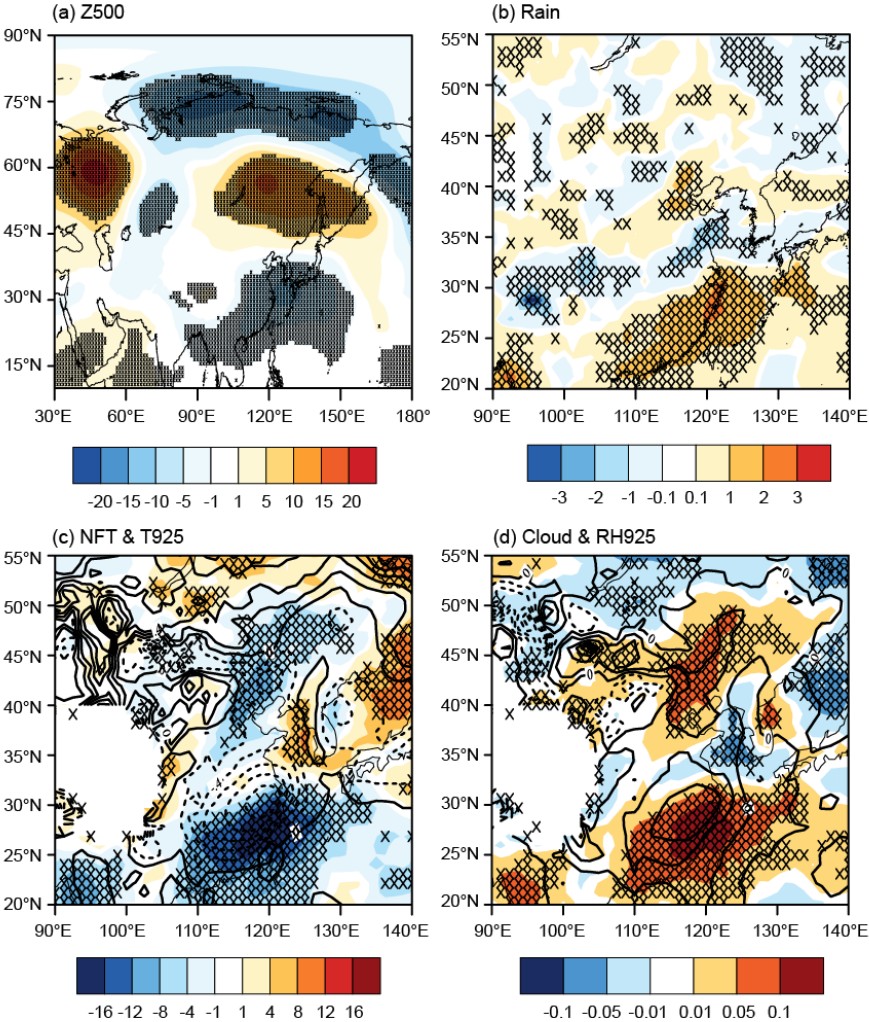

Figure 12. Composite results of the LowASI experiments (LowASI minus Ctrl) by the CAM5 model: (a) geopotential height at 500 hPa, (b)

570    preciptation, (c) net radiative flux at the top of the atmosphere (shading) and temperature at 925 hPa (contour), and (d) sum of low and

medium cloud fraction (shading) and relative humidity at 925 hPa (contour). The black hatching denotes the differences with shading were

above the 95% confidence level (t-test).