# Peer review of "Links of Climate Variability among Arctic sea ice, Eurasia teleconnection pattern and summer surface ozone pollution in North China"

_Atmospheric Chemistry and Physics, 2018_

## Referee Comment (RC1) · Anonymous Referee #1 · 27 Nov 2018

ACPD Review
Arctic sea ice, Eurasia teleconnection pattern and summer surface ozone pollution in North China

Summary:

This paper uses a combination of observation and reanalysis data to investigate the possible impact of large-scale meteorological conditions on surface air quality (specifically ozone) in North China.  Arctic sea ice concentrations in the spring are identified as a driver of the Eurasian teleconnection pattern which during the negative phase leads to meteorological conditions which are favorable to the photochemical production of ozone over North China.

While I find this may be a novel result, I find this study lacks substance that demonstrates to me that the authors fully understand how they have come to these conclusions.  I recommend the manuscript undergoes major revisions to include more details.

Major Comments:

The paper is very short compared to the number of figures included (6 Figures plus 12 in the supplement).  It reads to me like a "Letters" type of paper (i.e., Geophysical Research Letters) where one has a new time-sensitive idea or maybe a Nature or Science paper, where the article itself is short but the detailed description of data/methods/etc are in a supplemental at the end.   However, this manuscript is short and lacks the detailed description of the data and methods and discussing the results in the greater context of current literature as I would expect from an ACP article.   I found the Introduction section haphazard without a clear focus.  Take the time to clearly outline and describe each idea.  It jumps from ozone in China (Line 27) to European clean air laws (Line 28; could talk about US Clean Air act too) back to China (Line 30), to finally talking about how ozone is formed (Line 32), to how the NAO impacts European ozone (Line 34) to North American ozone and the jet stream position (Line 35-36), back to Asia (Line 36) and briefly mentions the Eurasia teleconnection pattern (line 41).  Each one of these ideas could be and should be expanded on.

The Wang and He (2015) EU calculation was adapted from the EU pattern of Wallace and Gutzler (1981).  Does the reason they adapted the equation apply for this project?  Which calculation was used in the citations in the Introduction (line 42) versus in the Methods section (Line 73)?  A full description in the introduction regarding the EU original teleconnection pattern and the characteristics of its positive and negative phases are not described or illustrated and this would be beneficial for the Section 4 and 5.  Perhaps a useful reference would be Wang, N. & Zhang, Y. Clim Dyn (2015) 44: 1017. https://doi.org/10.1007/s00382-014-2171-z

The first portion of the Results section refer to figures in the supplemental and are actually referred to before Figure 1 (Lines 79-100).  Notes to the authors from ACP states "The supplement **shall contain only complementary information** but no scientific interpretations or

findings/messages that would go beyond the contents of the manuscript." Consider including Figure S1 at least in the main body of the text. Figures S6, S7 and S8 are also referenced in the results sections as more than complementary analysis to the main findings and should be considered for the main text.

Of the figures in the main text, some improvements should be made in order for the reader to follow along with the results in Sections 4 and 5. In Figure 1g,h, which temperature is color and which is contoured? It is hard to see the Wind arrows in some of the plots (Figure 1a,b and Figure 4), consider rescaling or decluttering? Figures 4 and 6c are difficult to read. In the figure captions with the contours, it is not stated what are the contour intervals. In some of the supplemental figures the contours are labelled. Either label or define (e.g., is dashed for negative in Figure 4a?). Some of the figures are too small or the shading is too dark (saturated) or the presence of wind arrows makes it difficult to see the dotted significant areas (Figs 4, 5, 6c,d). As for the supplemental, I do not understand the blue lines in Figure S2, and the labelling of the x-axis (does it start June 2006 or June 2007, the tick makes no sense with the figure caption). I also do not know how to read figure S3 (maybe a table would be better?). It looks to me like the histogram has been cut off and values well above 90 should be shown.

Throughout the manuscript, both ERA-Interim and NCEP/NCAR reanalysis data (referred to as NOAA data in the paper, but more commonly referred to as NCEP/NCAR reanalysis) are used, with ERA-Interim being used as the main result and NCEP/NCAR reanalysis shown in the supplemental. In a similar vein, it is said that Shangdianzi station (SDZ) is one of three regional background stations in China (Line 56); Is it possible to use the other two stations to test the OWI methods?

In Section 3, the boxed regions that are used for the OWI calculations are shown in Figure 1. Make a better connection between these boxed regions in the text to the respective figure and also state that SDZ is located within these boxes. This is why you are doing the correlation coefficient between the SDZ ozone concentrations to the meteorological fields within these boxes, right? The boxes look to fit the maximum correlation for the shaded composite fields and therefore are different sizes. Can the authors discuss more as to the methods which lead to these boxed regions?

At the start of Section 4, the authors state "After 1979, the quality of the reanalysis data was improved to support studies of climate variability and change." This isn't quite true, the quality of reanalysis data improved for the period in the datasets after the assimilation of satellite data, which was made possible starting in 1979. The NCEP/NCAR reanalysis covers the period prior to the satellite era, therefore studies of climate variability and change must take into consideration the introduction of satellite data as well as subsequent changes in the observation system (introduction of new satellites and when satellites are no longer in operation). This needs to be properly addressed in the paper.

In Section 5, only the month of May sea ice is discussed. Did the authors investigated other lag periods? Is there literature that describes the interaction between sea ice concentrations and large-scale atmospheric circulation that can be referenced in this manuscript?

Minor Comments:

Line 26: smog is visible to humans and ozone is a key ingredient to smog.
Line 27-28: Can you provide any references which have looked at ozone pollution in China linked to climate variability
Line 28: 'benefitted' with two t's is the British spelling.
Line 28: Provide a reference and further details on the European 'rigorous air protection act' and what you mean by 'maintained good air quality' and in the same sentence 'ozone levels are increasing'.
Line 30: is the ambient air quality standard set by China or the World Health Organization? Please define and reference.
Line 31: Can you define North China, or indicate it on a map? It is confusing as on Line 93 it is written 'in the north of China, especially in North China".
Line 32: What do you mean by discharge?
Line 33: Describe the suitable weather conditions here, versus later in Lines 38-39 describing in detail the unfavorable weather conditions for ozone formation.
Line 41: Is it more common in the literature to refer to this teleconnection pattern simply as EU? In some of the figures EU is used. Commit to either using EU throughout or EUTP throughout the manuscript and figures.
Line 45: Can you list any more recent studies?
Line 45: Why talk about eastern China when the paper is interested in North China. Are there references that look at future ozone in North China?
Line 47: Which "previous studies" are you referring to?
Line 54-58: Is this ozone data publicly available?
Line 58: What is special about the MDA8 calculation of the Technical Regulation on Ambient Air Quality Index that it required referencing it?
Line 62: What is the native resolution of ERA-Interim? Did you download the data to this resolution or regrid it? Did you download it originally at the 6-hour resolution and then created daily and monthly datasets?
Line 69-70: As stated above, this reanalysis is more commonly referred to as NCEP/NCAR reanalysis.
Line 73: Correct Wang et al to Wang and He.
Line 80: Is it possible to label on a map these three regions?
Line 82: What is meant by "which appeared to be bordered by the Yangtze River".
Line 82: 'rather high' is subjective. Change to be more qualitative.
Line 86: Who's threshold?
Line 89: State that SDZ is labelled on Figure S1a (though really should be in S1b logically since right hand panel compares to SDZ).
Line 92: There are a few instances where the degree symbol is not superscript in the manuscript.
Line 94: How can there be a diurnal difference in a maximum daily average? This sentence makes no sense to me.
Line 97: Can you switch the order, introducing NOP before MOP.
Line 99: The mean number of MOP days is not explicitly shown in Figure S3.
Line 100: significant at what level? If this is from a figure in the supplement, consider moving that figure out of the supplement.

Line 103: use SAT instead of spelling it out and introduce why cooler temperature in the high troposphere ($T_{200}$) is favorable for surface ozone pollution (mentioned later in Line 117).

Line 121: what is downwash

Line 126: Why is Figure S5 included in the supplement

Line 130:  Any time MDA8 is used, is the reader expecting it to be the SDZ MDA8 unless otherwise stated?  Make that clear earlier on in the text.

Line 137: Reference Figure S7b-d.  'in North China (Figure S7b-d)'

Line 143-144: remind the reader that SDZ data began in 2006.  This sentence is not clear to me.

Line 145: What do you mean by a staged minimum?

Line 148: Significant to what test and level?

Line 150: provide a reference for the sentence ending '….due to the steady economic development in China"

---

## Referee Comment (RC2) · Anonymous Referee #2 · 18 Dec 2018

Review of "Arctic sea ice, Eurasia teleconnection pattern and summer surface ozone pollution in North China" by Yin et al.

Summary and General Comments:

This study attempts to link surface O3 from a site in North China to May Arctic sea ice extent and the Eurasia Teleconnection Pattern (EUTP), which amounts to a total of three degrees of separation (May Sea Ice -> Eurasia Teleconnection Pattern -> Ozone Weather Index (OWI) -> MDA8 O3 at the SDZ site). The main results of the paper seem a bit overstated because of this leap. The authors show some nice analyses linking each of these factors to one another, but the point of this paper is to show skill in predicting summer average MDA8 O3 from EUTP and May Arctic sea ice. I am not convinced the authors were so successful in that regard. For example, on

[Figure]

Figure 6b, the association with the Sea Ice Index and MDA8 is weak at best (see 2009, 2014, 2015, and 2017), and only ∼11 years of JJA O3 data from a single site are used to make these claims. The authors state in the abstract that May sea ice extent explains 60% of the interannual variability in summer surface O3, but that is actually the relationship that they found between sea ice and the OWI (Lines 188-190), which has its own separate, imperfect, relationship with MDA8 O3 levels at one specific site. In general, I found the statistical analysis and OWI formulation in the first half of the paper to be more enlightening than the second half. Perhaps a more careful discussion on EUTP and the sea ice effect will lead to a more convincing paper - Sections 4 and 5 are barely two pages of double-spaced text.

There is a wealth of information buried in the Supplemental Figures, which is often frustrating to a lot of readers. Figures S1 and S6 in particular receive a lot of attention in the text, and therefore should be included as regular figures. Figure S11 is of great importance to the result stated in the second last sentence of the abstract (though I have separate issues with it as I mention above). Several of the Supplemental Figures should be moved to the main paper.

What is the motivation behind analyzing NCEP/NCAR Reanalyses in addition to ERA-Interim? This adds unnecessary supplementary figures and text. You can simply state in the text that a sensitivity test was performed with NCEP/NCAR reanalysis, which yielded very similar results (this should be expected).

Recommendation:

In its current form, this paper is not yet suitable for publication in ACP. I am suggesting major revisions that include reorganization of Figures, and substantial edits to the text, mainly in the Introduction and Sections 4 and 5. I think that the paper discussion should remain focused on the meteorological relationships found linking the teleconnection patterns and surface O3 (which are interesting and useful to quantify!), and less on claims of seasonal predictability that may not be warranted by the current study.

Specific/Minor Comments:

Line 10: Surface ozone pollution is only increasing in certain parts of the world. Please be specific about China here.

Introduction: I find the discussion on "haze pollution" to be very confusing, especially because particulate pollution is not the focus in this paper. It would make the introduction much clearer to eliminate the use of "haze" and discuss only the O3 pollution issues in China.

Lines 26-27: Surface O3 is a major component of photochemical smog, so it is actually very frequently visible to humans when found in high amounts.

Lines 27-28: I disagree with this sentence. Surface O3 pollution in China is now a heavily studied topic. Rewrite this sentence to mainly highlight the novel aspect of your research into this topic: "The impacts of climate variability on surface O3 pollution in China have not been sufficiently studied."

Lines 28-29: This sentence on European pollution controls is out of place. Find a local example of pollution controls in China to discuss or remove this sentence entirely.

Line 30: Please quote the current Chinese air quality standards for surface O3.

Lines 35-40: These examples of large-scale atmospheric circulation patterns affecting surface O3 are relevant, but need to be presented more clearly to allow the reader to understand why they are worth discussing.

Lines 40-47: Similar comment as the one above. There are several studies discussed in one sentence each, but the organization makes this cumbersome to read. In general, the Introduction would benefit from a careful rewrite.

Lines 47-52: This paragraph is a good example of how the rest of the Introduction should be written to motivate carrying out this study.

Line 79: The first figure introduced should not be a supplemental figure. Figure S1 is

discussed extensively in the text, so it would be useful to just make this Figure 1.

Figure S1 Caption: Green "triangle", not "triple"

Line 98: Are these statistics only for June to August? Please be clear.

Line 101: Clarify that you are still referring to the O3 levels at SDZ.

Figure 1, panels g and h: Clarify in the figure caption which temperature is contoured and which is shaded in color. Does geopotential height at 500 or 250 hPa tell you more about surface O3 variability than 200 hPa temperature?

Lines 105-106: Are the northerly winds associated with higher surface O3 to the south of SDZ?

Line 108: Here and a couple of other locations, please fix the degree symbols.

Lines 108-112: Please rewrite these sentences to more simply state that clouds and precipitation are unfavorable for surface O3 production, leading to the meteorological composite in Figure 1f.

Line 116-117: The temperature of the upper troposphere is much more dynamically-than radiatively-influenced at synoptic timescales (i.e. through tropopause height variations). This sentence should be removed.

Figure S6: Much like Figure S1, this is a lot of text to dedicate to a figure that is buried in the Supplemental Information.

Line 136: Were there any sites that had a larger correlation between MDA8 and OWI than the SDZ site?

Lines 140-152 and Figure 3: I think analysis of OWI before 2006/2007 is not necessary, and introduces the uncertainty of a changing observing system (i.e. the satellite era beginning in 1979, ATOVS in 1998, etc.). These discussions are certainly outside of the scope of this paper, so I recommend not extending the OWI to periods when there

are no O3 observations to support your analyses. Limit your discussion to 2007 to the present.

Figure 3: All of the other figures that show SDZ data begin in 2007. Why does this include 2006, and why is the line a different color green here? Why does the NCEP/NCAR data end in 2014? Please clarify.

Figures 4 and 5: I thought the OWI was constructed by using some of these same meteorological variables. What is the value in calculating correlations between these variables and OWI?

Lines 170-177: This discussion is essentially a repeat of Lines 110-126. What is new here?

Lines 185-186: What sea ice variable is used here? Is this sea ice extent (generally referred to as ocean areas with at least 15% ice coverage)?

Line 188: Was the MDA8 anomaly also detrended? Is the MDA8 anomaly an anomaly from the mean of all JJA 2007-2017 MDA8 values?

Lines 205-207: As I understand your analysis, this statement is not correct (same with the abstract). The May sea ice extent contributes to about 60% of the interannual variability of OWI, not surface O3 pollution. There is a separate relationship between OWI and MDA8 at SDZ to be considered. The May Arctic sea ice anomaly does not even have the same sign as the MDA8 anomaly about 30% of the time (Line 187).

Technical Corrections:

Line 15-16: Clarify by rewriting as "Increased solar radiation and high temperatures during the EUTP positive phase dramatically enhanced O3 production."

Lines 24-25: See comments on the use of "haze" in my Specific Comments. This sentence should be removed or at least rewritten.

Line 25: Peking University?

Line 32: Rewrite this sentence "Surface O3 is a secondary pollutant."

Line 33: Change "Sedimentation" to "Surface deposition" and "attenuation" to "dispersion"

Line 50: Change "were" to "was"

Line 64: Do you mean the NCEP/NCAR Reanalysis? Please refer to these products as the NCEP/NCAR Reanalysis, instead of NOAA.

Line 82: "appeared to be delineated by the Yangtze River."

Line 84: Delete "approximately yearly."

Line 86: Replace "almost higher than" with "nearly." Also, what is the threshold of severe surface O3 pollution? Is this a definition set by the Chinese government? Are these O3 data publicly available?

Line 92: Rewrite this sentence, suggestion: "The distribution of correlation coefficients is similar to the MDA8 on Figure S1 panels a, c, e, and g."

Line 93: There is an extra period after China.

Line 94: Change "diurnal" to "daily"

Line 96 and in the references: Should be "Ministry of Environmental Protection"

Line 97: This information on surface O3 pollution definitions should be moved earlier in the paper.

Line 97: What is "nonsurface?" Please rename this to something like "non-polluted surface O3 levels"

Line 114: Delete "sufficient"

Lines 115-116: Rewrite: "In contrast, higher SAT enhances the photochemical reactions and resulted in higher surface O3 concentrations (Figure 1g)."

Line 121: Is this referring to the entrainment of O3 into the boundary layer? Please clarify and eliminate the word "downwash"

Line 126: Use NCEP/NCAR reanalysis.

Line 132: Delete "In contrast"

Figure S5: Either add a second panel with the ERA-Interim data for comparison or remove this figure. Consider also my comments about the inclusion of NCEP/NCAR Reanalysis in this paper in general.

Line 141: NCEP/NCAR reanalysis.

Figure S11 Caption: "1979 to 2017", correct?

---

## Author Comment (AC1) · 15 Jan 2019

The reply letter and the revised manuscript with track were attached as Supplement.

Please also note the supplement to this comment:
https://www.atmos-chem-phys-discuss.net/acp-2018-1127/acp-2018-1127-AC1-supplement.zip

---

## Author Response (AR1)

**Response to Reviewer #1**

Summary:

This paper uses a combination of observation and reanalysis data to investigate the possible impact of large-scale meteorological conditions on surface air quality (specifically ozone) in North China. Arctic sea ice concentrations in the spring are identified as a driver of the Eurasian teleconnection pattern which during the negative phase leads to meteorological conditions which are favorable to the photochemical production of ozone over North China.

While I find this may be a novel result, I find this study lacks substance that demonstrates to me that the authors fully understand how they have come to these conclusions. I recommend the manuscript undergoes major revisions to include more details.

**Major Comments:**

1. **The paper is very short compared to the number of figures included (6 Figures plus 12 in the supplement). It reads to me like a "Letters" type of paper (i.e., Geophysical Research Letters) where one has a new time-sensitive idea or maybe a Nature or Science paper, where the article itself is short but the detailed description of data/methods/etc are in a supplemental at the end. However, this manuscript is short and lacks the detailed description of the data and methods and discussing the results in the greater context of current literature as I would expect from an ACP article.**

*Reply:*

The manuscript has been revised according **to the ACP format**.

(1) In the revised version, the main texts of this article lengthened about **35%, i.e., from 220 lines to 290 lines.**

(2) The number of the Figures were **12** in the revised version, instead of 6 in the old version, in the main body.

(3) Both of the description of the data and methods and the discussion of the results were **rewritten and were recognized.**

*Revision:*

The revised manuscript without and with tracks were both uploaded for review.

2. **I found the Introduction section haphazard without a clear focus. Take the time to clearly outline and describe each idea. It jumps from ozone in China (Line 27) to European clean air laws (Line 28; could talk about US Clean Air act too) back to China (Line 30), to finally talking about how ozone is formed**

**(Line 32), to how the NAO impacts European ozone(Line 34) to North American ozone and the jet stream position (Line 35-36), back to Asia (Line 36) and briefly mentions the Eurasia teleconnection pattern (line 41). Each one of these ideas could be and should be expanded on.**

*Reply:*

According to the reviewer's suggestion, the introductions were **entirely revised and rewritten** now.

(1) the introduction of European clean air laws was confusing, and now was deleted. Thus, the introduction of the ozone polluted features f**ocused on those in China.**

(2) In the second paragraph of the introduction, the ideas how the climate anomalies (e.g., NAO, jet stream, west Pacific subtropical high and East Asia summer monsoon) were **expanded on**. The details can be found in the following revisions attached.

(3) Due to insufficient studies, related to how the climate anomalies impacted the ozone pollutions in China, some closely findings in North American were still introduced. Indeed, the findings, such as NAO-ozone in Europe and jet stream-ozone on North American, provided meaningful and substantial clues to our studies.

*Revision:*

…For example, the prevailing positive phase of the North Atlantic Oscillation contributed to the increasing ozone concentration in western and northern Europe, through the anomalous atmospheric circulations to influence regional photochemical processes (Christoudias et al., 2012; Pausata et al., 2012)…

…The summer surface $O_3$ variability in North America is significantly modulated by the position of the jet stream (Lin et al., 2014). Barnes and Fiore (2013) pointed out jet position may dynamically modulate surface ozone variability in eastern North America and other northern mid-latitude regions…

…A strong positive correlation between the East Asian summer monsoon and summer mean ozone existed. The model simulation by Yang et al. (2014) illustrated that the changes in meteorological parameters, associated with East Asian summer monsoon, lead to 2–5% interannual variations of surface $O_3$ concentrations over central eastern China. Focusing on the dataset in 2014, a significantly strong west Pacific subtropical high resulted in higher relative humidity, more clouds, more rainfall, less ultraviolet radiation and lower air temperatures, which were unfavourable for the formation of $O_3$ (Zhao and Wang, 2017)…

*Revision with tracks were also posted:*

There is rarely a direct discharge of s̶Surface O₃. is a secondary pollutant. The precursors of O₃ (e.g. NOx and VOC) photochemically react to generate O₃ under suitable weather conditions, i.e., hot-day and sunny environments (An et al., 2009). S̶e̶d̶i̶m̶e̶n̶t̶a̶t̶i̶o̶n̶Surface deposition, dynamic transport and a̶t̶t̶e̶n̶u̶a̶t̶i̶o̶n̶ dispersion of O₃ are also closely related to atmospheric circulations. For example, T̶h̶e̶ ̶N̶o̶r̶t̶h̶ ̶A̶t̶l̶a̶n̶t̶i̶c̶ ̶O̶s̶c̶i̶l̶l̶a̶t̶i̶o̶n̶ ̶i̶n̶t̶e̶r̶c̶o̶n̶t̶i̶n̶e̶n̶t̶a̶l̶l̶y̶ ̶a̶f̶f̶e̶c̶t̶s̶ ̶s̶u̶r̶f̶a̶c̶e̶ ̶O̶₃̶ ̶c̶o̶n̶c̶e̶n̶t̶r̶a̶t̶i̶o̶n̶s̶ ̶o̶v̶e̶r̶ ̶E̶u̶r̶o̶p̶e̶ ̶(̶C̶h̶r̶i̶s̶t̶o̶u̶d̶i̶a̶s̶ ̶e̶t̶ ̶a̶l̶.̶,̶ ̶2̶0̶1̶2̶;̶ ̶P̶a̶u̶s̶a̶t̶a̶ ̶e̶t̶ ̶a̶l̶.̶,̶ ̶2̶0̶1̶2̶)̶the prevailing positive phase of the North Atlantic Oscillation contributed to the increasing ozone concentration in western and northern Europe, through the anomalous atmospheric circulations to influence regional photochemical processes (Christoudias et al., 2012; Pausata et al., 2012). ̶-The summer surface O₃ variability in North America is significantly modulated by the position of the jet stream (B̶a̶r̶n̶e̶s̶ ̶a̶n̶d̶ ̶F̶i̶o̶r̶e̶,̶ ̶2̶0̶1̶3̶;̶-Lin et al., 2̶0̶1̶5̶2014). Barnes and Fiore (2013) pointed out jet position may dynamically modulate surface ozone variability in eastern North America and other northern mid-latitude regions. A strong positive correlation between the East Asian summer monsoon and summer mean ozone existed d̶u̶r̶i̶n̶g̶ ̶1̶9̶8̶6̶-̶2̶0̶0̶6̶,̶ ̶b̶a̶s̶e̶d̶ ̶o̶n̶ ̶n̶u̶m̶e̶r̶i̶c̶a̶l̶ ̶m̶o̶d̶e̶l̶ ̶r̶e̶s̶u̶l̶t̶s̶ ̶(̶Y̶a̶n̶g̶ ̶e̶t̶ ̶a̶l̶.̶,̶ ̶2̶0̶1̶4̶)̶. The model simulation by Yang et al. (2014) illustrated that the changes in meteorological parameters, associated with East Asian summer monsoon, lead to 2–5% interannual variations of surface O₃ concentrations over central eastern China. Focusing on the dataset in 2014, A̶a significantly strong west Pacific subtropical high resulted in higher relative humidity, more clouds, more rainfall, less ultraviolet radiation and lower air temperatures, which were u̶n̶f̶a̶v̶o̶r̶a̶b̶l̶e̶unfavourable for the formation of O₃ (Zhao and Wang, 2017). The photochemical reaction was the main local sources of O₃. The hot and dry environments and the violent solar radiation could accelerate the chemical conversion from the precursor to O₃ (An et al., 2009; Tong et al., 2017). In 2013, a severe heat wave-, with highest temperature 41.1 °C, contributed to the high O₃ concentration in the Yangtze River Delta (Pu et

3. **The Wang and He (2015) EU calculation was adapted from the EU pattern of Wallace and Gutzler (1981). Does the reason they adapted the equation apply for this project? Which calculation was used in the citations in the Introduction (line 42) versus in the Methods section (Line 73)? A full description in the introduction regarding the EU original teleconnection pattern and the characteristics of its positive and negative phases are not described or illustrated and this would be beneficial for the Section 4 and 5. Perhaps a useful reference would be Wang, N. & Zhang, Y. Clim Dyn (2015) 44: 1017.** **https://doi.org/10.1007/s00382-014-2171-z**

*Reply:*

(1) The reference, fully described the EU pattern in winter, substantially helped us to understand the impacts of EU in the Asian climate and was **detailedly introduced**.

(2) However, the season Wang and Zhang (2015) and Wallace and Gutzler (1981) concerned is winter.

(3) Wang and He (2015) regarded the **summer EU pattern** as the main reason for the severe summer drought in North China in 2014. Considering the **seasonal change** of the EU pattern's location, the calculation procedure for the EU index here was adapted from Wang and He (2015).

*Revision:*

…The EU pattern is a major teleconnection pattern in the Northern Hemisphere and appears in all seasons. **Wang and Zhang (2015)** used the method defined by Wallace and Gutzler (1981) to calculate the EU pattern index in winter and pointed out that the positive EU phase is associated with a cold-dry climate in East China, vice versa. Meanwhile, Wang and He (2015) regarded the summer EU pattern as the main reason for the severe summer drought in North China in 2014. Considering the seasonal change of the EU pattern's location, the calculation procedure for the EU index here was consistent with that in Wang and He (2015)…

Wang, N., Zhang, Y.: Evolution of Eurasian teleconnection pattern and its relationship to climate anomalies in China. Climate Dynamics, 44(3-4):1017-1028. 2015

4. **The first portion of the Results section refer to figures in the supplemental and are actually referred to before Figure 1 (Lines 79-100). Notes to the authors from ACP states "The supplement shall contain only complementary information but no scientific interpretations or findings/messages that would go beyond the contents of the manuscript." Consider including Figure S1 at least in the main body of the text. Figures S6, S7 and S8 are also referenced in the results sections as more than complementary analysis to the main findings and should be considered for the main text.**

*Reply:*

The Figure S1, S6, S7, S8 and S11 were **moved to the main text** in the revised version. Now, there were 12 Figures in the main text and 6 Figures in the supplemental information.

*Revision:*

The mentioned Figures were Figure 1, 3, 5, 8 and 11 in the revised manuscript.

5. **Of the figures in the main text, some improvements should be made in order for the reader to follow along with the results in Sections 4 and 5.**

**5.1 In Figure 1g, h, which temperature is color and which is contoured?**

*Reply:*

The negligence was revived. The SAT is with shading and the temperature at 200 hPa is contoured.

*Revision:*

Figure 2…(g–h) SAT (shading), and temperature at 200 hPa (contour)…

**5.2 It is hard to see the Wind arrows in some of the plots (Figure 1a,b and Figure 4), consider rescaling or decluttering?**

*Reply:*

The wind arrows were **enlarged**, i.e., rescaling and decluttering. The corresponding Figures in the supplementary information were also revised.

*Revision:*

These composites were calculated using the ERA-Interim datasets…

[Figure]

…(a–b) surface wind (arrow) and v-wind (shading)…

…The green triangle in panel (a-b) illustrates the location of the Shangdianzi site…

These composites were calculated using the NCEP/NCAR datasets…

[Figure]

…(a–b) surface wind (arrow) and v-wind (shading)…

…The green triangle in panel (a-b) illustrates the location of the Shangdianzi site…

Figure 4 in the old version were revised and is Figure 7 now.

[Figure]

Figure 7. The associated atmospheric circulation. (a) The correlation coefficients between the JJA mean OWI and surface air temperature (shading), wind (arrow) at 200 hPa and geopotential height at 500 hPa (contour) from 1979 to 2017. The black dots indicate that the CC with surface air temperature was above the 95% confidence level. The cross-section (110 °–125 °E mean) correlation coefficients between JJA mean OWI (a), EU pattern index (b) and relative humidity (shading), temperature (contour), wind (arrow, vertical speed multiplied by 100) from 1979 to 2017. The black dots indicate that the CC with relative humidity exceeded the 95% confidence level (t test). The data used here are ERA-Interim datasets.

**5.3 In the figure captions with the contours, it is not stated what are the contour intervals. In some of the supplemental figures the contours are labelled. Either label or define (e.g., is dashed for negative in Figure 4a?).**

*Reply:*

The labels of the contours were added in the revised manuscript.

Please see the above attached Figure 7 (i.e., Figure 4 in the former version).

The revised Figure S4 in the supplementary materials were also attached below.

[Figure]

**Figure S4.** The associated atmospheric circulation. (a) The correlation coefficients between the JJA mean OWI and surface air temperature (shading), wind (arrow) at 200 hPa and geopotential height at 500 hPa (contour) from 1979 to 2017. The black dots indicate that the CC with surface air temperature was above the 95% confidence level. The cross-section (110°–125°E mean) correlation coefficients between JJA mean OWI (a), EUTP index (b) and relative humidity (shading), temperature (contour), wind (arrow, vertical speed multiplied by 100) from 1979 to 2017. The black dots indicate the CC with relative humidity exceeding the 95% confidence level (t test). The data used here are NOAA datasets.

**5.4 Some of the figures are too small or the shading is too dark (saturated) or the presence of wind arrows makes it difficult to see the dotted significant areas (Figs 4, 5, 6c,d).**

*Reply:*

The related Figures were re-plotted. The wind arrows were enlarged, the color bar of shading was improved, i.e., **the saturation is modified**.

*Revision:*

The revisions for Figure 4 can be found in the reply to Comments 5.2 and 5.3.

The revised Figure 9 (i.e., Figure 5 in the former version) and Figure 10 (i.e., Figure 6 in the former version) in the supplementary materials were also attached below.

[Figure]

Figure 9. The associated meteorological conditions. (a) The correlation coefficients between the JJA mean OWI and v wind at 10 m (shading), surface wind (arrow), (b) relative humidity near the surface (shading), boundary layer height (contour), (c) precipitation (shading), downward UV radiation at the surface (contour), (d) downward solar radiation at the surface (shading), sum of low and medium cloud cover (contour) from 1979 to 2017. The black dots indicate that the CC with temperature was above the 95% confidence level. The contours plotted in panel (b–d) exceeded the 95% confidence level. The data used here are ERA-Interim datasets.

[Figure]

Figure 10. The role of the Arctic sea ice. (a) The correlation coefficients between the JJA mean OWI and May sea ice, (b) The variation of the May SI index (red bar, area-averaged sea ice of the green boxes in panel a), JJA mean EU pattern index (blue bar) and JJA mean observational SDZ MDA8 (black bar) from 2007 to 2017. (c) The correlation coefficients between the May SI index and surface air temperature (shading), geopotential height at 500 hPa (contour) from 1979 to 2017. The black dots indicate that the CC with surface air temperature was above the 95% confidence level. (d) The correlation coefficients between the May SI index and precipitation (shading), surface wind (arrow), (e) downward UV radiation at the surface (shading) and sum of low and medium cloud cover (contour) from 1979 to 2017. The black dots indicate that the shading CC with precipitation (d) and downward UV radiation (e) was above the 95% confidence level. The data used here are ERA-Interim datasets.

**5.5 As for the supplemental, I do not understand the blue lines in Figure S2, and the labelling of the xaxis (does it start June 2006 or June 2007, the tick makes no sense with the figure caption).**

*Reply:*

The data were JJA 2007, JJA 2008….Actually, they were not temporally continuous. Thus the blue lines were plotted to **separate the data in different years**.

The citation of Figure S1 was improved to include more explanation, as follows:

*Revision:*

Figure S1. The variation in the SDZ MDA8 from June to August during 2007–2017.**The blue lines were used to divide the data in different years. For example, the data on two sides of the first lines were belonged to JJA 2007 and JJA 2008, respectively.**

**5.6 I also do not know how to read figure S3 (maybe a table would be better?). It looks to me like the histogram has been cut off and values well above 90 should be shown.**

*Reply:*

According to the reviewer's advice, the Figure was **changed to Table S1**, which was much clearer to show the variation in the number of MOP and NOP.

*Revision:*

Table S1. The number of days with MOP and NOP events.

|  | 2007 | 2008 | 2009 | 2010 | 2011 | 2012 | 2013 | 2014 | 2015 | 2016 | 2017 | Mean |
|---|---|---|---|---|---|---|---|---|---|---|---|---|
| **MOP** | 22 | 8 | 8 | 24 | 14 | 9 | 10 | 11 | 26 | 11 | 12 | 14 |
| **NOP** | 4 | 12 | 12 | 8 | 4 | 17 | 22 | 9 | 4 | 18 | 16 | 11.5 |

6. **Throughout the manuscript, both ERA-Interim and NCEP/NCAR reanalysis data (referred to as NOAA data in the paper, but more commonly referred to as NCEP/NCAR reanalysis) are used, with ERA-Interim being used as the main result and NCEP/NCAR reanalysis shown in the supplemental.**

   **In a similar vein, it is said that Shangdianzi station (SDZ) is one of three regional background stations in China (Line 56); Is it possible to use the other two stations to test the OWI methods?**

*Reply:*

(1) The expression of NOAA data has been **changed to the NCEP/NCAR** data throughout the MS.

(2) The other two regional background stations in China are Longfengshan in Heilongjiang province and Lin'an to the southwest of Shanghai, which are quite far away from the North China and are **uncorrelated to this study.** Furthermore, the

data from the regional background stations were not public, and we did not gain the data in the other stations.

*Revision:*

The daily mean and monthly mean ERA-Interim data were directly downloaded from the ERA-Interim website in this study. Furthermore, the daily mean and monthly reanalysis datasets supported by the National Oceanic and Atmospheric Administration  were also employed and denoted as  NCEP/NCAR (National Center for Environmental Prediction and the National Center for Atmospheric Research) data. The 2.5°×2.5° geopotential height (Z), zonal and meridional wind, relative humidity, vertical velocity, air temperature at different pressure levels, SAT and wind, downward UV radiation, downward solar radiation, low and medium cloud cover were downloaded  (Kalnay et al. 1996). The BLH of NCEP/NCAR dataset was only available from 1979 to 2014 in  the website of the NOAA-CIRES 20th Century Reanalysis version 2c (Giese et al., 2016). The daily precipitation data was from the CPC global analysis of the daily precipitation dataset (Chen et al., 2008).

**7. In Section 3, the boxed regions that are used for the OWI calculations are shown in Figure 1. Make a better connection between these boxed regions in the text to the respective figure and also state that SDZ is located within these boxes. This is why you are doing the correlation coefficient between the SDZ ozone concentrations to the meteorological fields within these boxes, right? The boxes look to fit the maximum correlation for the shaded composite fields and therefore are different sizes. Can the authors discuss more as to the methods which lead to these boxed regions?**

*Reply:*

(1) The location of Shangdianzi station was also plotted in Figure 2, i.e., **the green triangle, to enhance the connection**.

(2) The process of calculate the meteorological index is added. The important point is the method to ensure the boxed region. Simply, **the averaging area for meteorological indexes were the regions with most significantly different elements** in the composites of MOP and NOP events. Because the box was chosen for each element, the boxes was a little different, but still near the North china.

*Revision:*

(1) …During calculating the correlation coefficients with the meteorological conditions, the averaging area for meteorological indexes were the regions with most significantly different elements in the composites of MOP and NOP events…

(2) …The green triangle in panel (a-b) illustrates the location of the Shangdianzi site…

8. **At the start of Section 4, the authors state "After 1979, the quality of the reanalysis data was improved to support studies of climate variability and change." This isn't quite true, the quality of reanalysis data improved for the period in the datasets after the assimilation of satellite data, which was made possible starting in 1979. The NCEP/NCAR reanalysis covers the period prior to the satellite era, therefore studies of climate variability and change must take into consideration the introduction of satellite data as well as subsequent changes in the observation system (introduction of new satellites and when satellites are no longer in operation). This needs to be properly addressed in the paper.**

*Reply:*

The original presentation was confusing and not accurate. According to the reviewer's suggestion, the error was revised.

*Revision:*

…After the assimilation of satellite data, possible in 1979, the quality of the reanalysis data was improved…

9. **In Section 5, only the month of May sea ice is discussed. Did the authors investigated other lag periods?**
   **Is there literature that describes the interaction between sea ice concentrations and large-scale atmospheric circulation that can be referenced in this manuscript?**

*Reply:*

(1) The other lag periods were also studied during our research. In the other months, i.e., **from December to April, the sea ice anomalies did not show closely connections**

with the JJA mean OWI (Figure R1). Related discussions were added in the revised manuscript.

[Figure]

Figure R1. The correlation coefficients between the JJA mean OWI and December (a), January (b), February (c), March (d), April (e) and May (f) sea ice concentration.

(2) The literature that describes the interaction between sea ice concentrations and large-scale atmospheric circulation were referenced in the revised manuscript.

*Revision:*
(1) The correlation between the sea ice and JJA OWI was monthly checked (Figure omitted), and we found the interannual variation of OWI was significantly correlated with May sea ice conditions to the north of Eurasia, especially near the Gakkel Ridge, the Canada Basin and the Beaufort Sea (Figure 10a).
(2) The EU pattern originated from the Arctic region. The preceding sea ice anomalies could stimulate atmospheric responses like EU pattern in summer (Wang and He, 2015) Thus, the role of Arctic sea ice on the OWI was also explored in this study.

**Minor Comments:**
**Line 26: smog is visible to humans and ozone is a key ingredient to smog.**
*Reply:*
According to the reviewer's advice, the discussion about the visibility was **deleted**.

*Revision:*

Due to drastic air pollution control in China since 2013, haze pollutions are being controlled in recent years (The environmental statistics unit of stat-centre in Peking University, 2018), appearing as sharp decreasing in fine particulate matter (PM$_{2.5}$). However, surface O$_3$ pollution, which always occurred on clear and sunny days (Wang et al., 2017), has not improved (Li et al., 2018).

**Line 27-28: Can you provide any references which have looked at ozone pollution in China linked to climate variability**

*Reply:*

A reference was cited.

*Revision:*

… but the impacts of climate variability on surface O3 pollution in China (Yang et al, 2014) have not been sufficiently studied…

**Line 28: 'benefitted' with two t's is the British spelling.**
**Line 28: Provide a reference and further details on the European 'rigorous air protection act' and what you mean by 'maintained good air quality' and in the same sentence 'ozone levels are increasing'.**

*Reply:*

(1) According to the major comment 2, the introduction of European clean air laws was deleted.

(2) Thus, the word "benefitted" was concomitantly deleted.

*Revision:*

**Line 30: is the ambient air quality standard set by China or the World Health Organization? Please define and reference.**

*Reply:*

The information has been added in the text. It is the standard in China.

*Revision:*

…the surface O₃ concentrations exceeded the ambient air quality standard of China (i.e., 100 µg/m³) by 100–200 % (Wang et al., 2017)…

**Line 31: Can you define North China, or indicate it on a map? It is confusing as on Line 93 it is written 'in the north of China, especially in North China".**

*Reply:*

(1) According to the reviewer's advice, the range of North China was plotted in Figure 1h and 5d.

(2) Line 93 was rewritten.

*Revision:*

…The distribution of correlation coefficients is similar to the MDA8 on Figure 1 (a, c, e, g). The SDZ MDA8 significantly covaried with the MDA8 in North China in summer…

[Figure]

The black box in panel h is the range of North China.

**Line 32: What do you mean by discharge?**

*Reply:*

It should be emission. According to the other reviewer's advice, this sentence was modified to "Surface $O_3$ is a secondary pollutant".

*Revision:*

Surface $O_3$ is a secondary pollutant…

**Line 33: Describe the suitable weather conditions here, versus later in Lines 38-39 describing in detail the unfavorable weather conditions for ozone formation.**

*Reply:*

The suitable and unfavorable weather were explained in the revised MS.

*Revision:*

…The precursors of $O_3$ (e.g. NOx and VOC) photochemically react to generate $O_3$ under suitable weather conditions, i.e., hot-day and sunny environments (An et al., 2009)…

…Focusing on the dataset in 2014, a significantly strong west Pacific subtropical high resulted in higher relative humidity, more clouds, more rainfall, less ultraviolet radiation and lower air temperatures, which were unfavourable for the formation of $O_3$ (Zhao and Wang, 2017)…

**Line 41: Is it more common in the literature to refer to this teleconnection pattern simply as EU? In some of the figures EU is used. Commit to either using EU throughout or EUTP throughout the manuscript and figures.**

*Reply:*

The abbreviation of Eurasia teleconnection pattern was **unified as EU** throughout the manuscript and figures.

*Revision:*

was constructed, which extends the study period to the historical period before 2007 and the projected future. Here, we show that in addition to anthropogenic emissions, the Eurasia teleconnection pattern (EUTPEU), a major globally significant atmospheric teleconnection pattern, influences surface $O_3$ pollution in North China on a time scale of climate. The local meteorological conditions associated with the EUTPEU pattern positive phase supported intense and efficient photochemical reactions to produce more surface $O_3$. The associated southerlies over North China transported surrounding $O_3$ precursors to superpose local emissions. Increased Ssolar radiation and high temperature during the positive EU phase dramatically enhanced $O_3$ photochemical reactionsproduction. Furthermore, due to the close connection between the preceding May Arctic sea ice and summer EUTPEU pattern, approximately 60% of the interannual variability of summer surface $O_3$ pollution was

**Line 45: Can you list any more recent studies?**

*Reply:*

More recent studies were listed, such as Zhu and Liao (2016) and Gaudel et al., (2018).

*Revision:*

Due to the close relationship between surface $O_3$ and meteorological conditions, the impacts of climate change on $O_3$ have been projected by various numerical models (Doherty et al., 2013; Melkonyan and Wagner, 2013; **Zhu and Liao, 2016; Gaudel et al., 2018**).

**Line 45: Why talk about eastern China when the paper is interested in North China. Are there references that look at future ozone in North China?**

*Reply:*

(1) North China was **a part of e**astern China.

(2) Related studies concentrated in North China were quite few, thus we talked about the findings in eastern China. Although these researches were done for the larger region, i.e., eastern China, some findings were be appropriate for the ozone pollution in North China. The cited papers actually provided clues to us.

**Line 47: Which "previous studies" are you referring to?**

*Reply:*

Related reference was cited here.

*Revision:*

…However, previous studies mainly focused on observational analyses of several synoptic processes (e.g., Zhao and Wang, 2017), rather than long-term climate diagnostics, because of the lack of long-term surface $O_3$ observations…

**Line 54-58: Is this ozone data publicly available?**

*Reply:*

The ozone data from 2014 to 2017 were publicly available in the website of the Ministry of Environmental Protection of China.

**Line 58: What is special about the MDA8 calculation of the Technical Regulation on Ambient Air Quality Index that it required referencing it?**

*Reply:*

The MDA8 was the maximum of the **running 8 h mean** $O_3$ concentrations during 24 hours in the day.

The explanation was supplemented in the revised manuscript.

*Revision:*

…The MDA8 was calculated as the maximum of the running 8 h mean $O_3$ concentrations during 24 hours in the day…

**Line 62: What is the native resolution of ERA-Interim? Did you download the data to this resolution or regrid it? Did you download it originally at the 6-hour resolution and then created daily and monthly datasets?**

*Reply:*

The resolution of ERA-Interim here is $1°{\times}1°$.

The daily mean and monthly mean datasets were directly downloaded and used in the manuscript.

*Revision:*

…The $1°{\times}1°$ ERA-Interim data used here included … The daily mean and monthly mean ERA-Interim data were directly downloaded from the ERA-Interim website in this study…

**Line 69-70: As stated above, this reanalysis is more commonly referred to as NCEP/NCAR reanalysis.**

*Reply:*

The expression of NOAA data has been changed to the NCEP/NCAR data.

*Revision:*

The daily mean and monthly mean ERA-Interim data were directly downloaded from the ERA-Interim websiteanalyzed in this study. Furthermore, the daily mean and monthly reanalysis datasets supported by the National Oceanic and Atmospheric Administration (NOAA) were also employed and denoted as NOAA NCEP/NCAR (National Center for Environmental Prediction and the National Center for Atmospheric Research) data. The 2.5°×2.5° geopotential height (Z), zonal and meridional wind, relative humidity, vertical velocity, air temperature at different pressure levels, SAT and wind, downward UV radiation, downward solar radiation, low and medium cloud cover were downloaded from the National Center for Environmental Prediction and the National Center for Atmospheric Research (Kalnay et al. 1996). The BLH of NCEP/NCAR dataset was only available from 1979 to 2014 in the NOAA data was derived fromthe website of the NOAA-CIRES 20th Century Reanalysis version 2c (Giese et al., 2016). The daily precipitation data was from the CPC global analysis of the daily precipitation dataset (Chen et al., 2008).

**Line 73: Correct Wang et al to Wang and He.**

*Reply:*

The errors were corrected.

*Revision:*

…the calculation procedure for the EU index here was consistent with that in Wang and He (2015)…

**Line 80: Is it possible to label on a map these three regions?**
*Reply:*
We tried to label these three regions on a map, but the Figure became unclear. To keep the meaning of the sentence and avoid confusion, we finally deleted these three regions.

*Revision:*
…During 2006–2014, O3 concentrations were only observed in the most developed regions in China…

**Line 82: What is meant by "which appeared to be bordered by the Yangtze River".**
*Reply:*
This sentence was confusing, and was improved as follows:
*Revision:*
…$O_3$ concentrations in the high-mid latitudes were higher than those in the lower latitudes, which appeared to be **separated** by the Yangtze River…

**Line 82: 'rather high' is subjective. Change to be more qualitative.**
*Reply:*
This sentence was confusing, and was improved as follows:

*Revision:*
…The $O_3$ concentrations in North China were already high in 2014;…

**Line 86: Who's threshold?**
*Reply:*
It is the threshold of the server surface $O_3$ pollution in China.

*Revision:*
…The observations, with maximum MDA8 higher than 265 μg/m³ (i.e., the threshold of the server surface $O_3$ pollution in China)…

**Line 89: State that SDZ is labelled on Figure S1a (though really should be in S1b logically since right hand panel compares to SDZ).**
*Reply:*
The location of **SDZ is labelled on panels** b, d, f, and h in Figure 1 now.

*Revision:*

The green triangle in panels b, d, f, and h illustrate the location of the SDZ station.

[Figure]

Figure 1. The distribution of the JJA mean MDA8 (a, c, e, g) and the correlation coefficients (b, d, f, h) between the daily MDA8 and SDZ MDA8 from 2014 to 2017. The black cross in panels a, c, e, and g indicate that the maximum daily MDA8 was larger than 265 µg/m³. The black cross in panels b, d, f, and

h indicate that the CC was above the 95% confidence level. The green triangle in panels b, d, f, and h illustrate the location of the SDZ station. The black box in panel h is the range of North China.

**Line 92: There are a few instances where the degree symbol is not superscript in the manuscript.**

*Reply:*

The similar errors were corrected throughout the manuscript.

**Line 94: How can there be a diurnal difference in a maximum daily average? This sentence makes no sense to me.**

*Reply:*

The errors were corrected. It should be **daily difference**.

*Revision:*

The daily difference in MDA8 was large, which contradicts the quasi-constant emission of ozone precursors.

**Line 97: Can you switch the order, introducing NOP before MOP.**

*Reply:*

According to the reviewer's advice, the orders were switched.

*Revision:*

…the thresholds of non-surface $O_3$ polluted level (NOP) and moderate surface $O_3$ polluted level (MOP) are 100 μg/m³ and 215 μg/m³, respectively…

**Line 99: The mean number of MOP days is not explicitly shown in FigureS3.**

*Reply:*

Replied in comment 5.6, the Figure was changed to Table S1 and the mean number of the MOP and NOP days were also listed.

*Revision:*

Table S1. The number of days with MOP and NOP events.

| | 2007 | 2008 | 2009 | 2010 | 2011 | 2012 | 2013 | 2014 | 2015 | 2016 | 2017 | Mean |
|---|---|---|---|---|---|---|---|---|---|---|---|---|
| **MOP** | 22 | 8 | 8 | 24 | 14 | 9 | 10 | 11 | 26 | 11 | 12 | 14 |
| **NOP** | 4 | 12 | 12 | 8 | 4 | 17 | 22 | 9 | 4 | 18 | 16 | 11.5 |

**Line 100: significant at what level? If this is from a figure in the supplement, consider moving that figure out of the supplement.**

*Reply:*

The interannual variation in MOP (NOP) days was significant at the 95% confidence level, which was verified by t-test.

*Revision:*

…The interannual variation in MOP (NOP) days was significant at the 95% confidence level, without an obvious long-term trend…

**Line 103: use SAT instead of spelling it out and introduce why cooler temperature in the high troposphere (T200) is favorable for surface ozone pollution (mentioned later in Line 117).**

*Reply:*

(1) the abbreviation, i.e., SAT, was used.

(2) According to the other reviewer's suggestion (attached below), the discussion about the reason, why cooler temperature in the high troposphere (T200) is favorable for surface ozone pollution, was deleted.

> Line 116-117: The temperature of the upper troposphere is much more dynamicallythan radiatively-influenced at synoptic timescales (i.e. through tropopause height variations). This sentence should be removed.

Because we cannot perfectly address it now, the question was leave as an open question in the "Conclusion and Discussion" section.

*Revision:*

…The processes how the weather conditions impacted the photochemical reaction were not deeply discussed here and have been analyzed in many previous studies by the atmospheric chemists. **However, the reason why the cooler high troposphere contributed to the surface ozone pollution was still an open question and needed further attention**…

**Line 121: what is downwash**

*Reply:*

The downwash was confusing and was changed to "entrainment…… into the boundary layer".

*Revision:*

…The **entrainment of atmospheric ozone from the upper air into the boundary layer** enlarged the surface O3 concentration (An et al., 2009)…

**Line 126: Why is Figure S5 included in the supplement**

*Reply:*

All of the results from the NCEP/NCAR datasets were included in the supplementary information. The identical results were Figure 2 c, d by the ERA-Interim data.

**Line 130: Any time MDA8 is used, is the reader expecting it to be the SDZ MDA8 unless otherwise stated? Make that clear earlier on in the text.**

*Reply:*

The negligence were corrected throughout the manuscript. The MDA8 in SDZ station was denoted as **SDZ MDA8**, comparing to the **MDA8 in the other sites**.

*Revision:*

2017). The correlation coefficients between SDZ MDA8 and the observed MDA8 at the other sites were calculated and are shown in Figure 1 (b, d, f, h). The distribution of correlation coefficients is similar to the MDA8 on Figure 1 (a, c, e, g). The SDZ MDA8 significantly covaried with the MDA8 in North China in summer. Along with the increasing of the surface $O_3$ pollution, the covariation and the representativeness of SDZ MDA8 to the MDA8 in North China was strengthened. However,

**Line 137: Reference Figure S7b-d. 'in North China (Figure S7b-d)'**

*Reply:*

The related Figure, i.e., Figure 5, was referred now.

*Revision:*

…The significantly positive correlations were distributed in North China (Figure 5 b-d)…

**Line 143-144: remind the reader that SDZ data began in 2006. This sentence is not clear to me.**

*Reply:*

The confusing expression was corrected as follows:

*Revision:*

…During 2007–2017, the constructed JJA (June-July-August) mean OWI varied similarly with the observed MDA8 and captured the extremes (Figure 6). **Although the range of the SDZ MDA8 was 2006–2017, in the above OWI construction processes, only the data from 2007 to 2017 were used.** Thus, the datasets in 2006 were independent samples, and could verify the performance of the OWI…

**Line 145: What do you mean by a staged minimum?**

*Reply:*

The presentation was not clear. Actually, the value in 2006 was the minimum.
This sentence was modified.

*Revision:*

…The JJA mean OWI in 2006 successfully reflected the variation in observed MDA8;
even the MDA8 in 2006 was the minimum…

**Line 148: Significant to what test and level?**

*Reply:*

This sentence was modified.

*Revision:*

…Before the mid-1990s, the OWI was below zero, with a slightly decreasing trend and
insignificant interannual variation. Since then, the OWI has increased…

**Line 150: provide a reference for the sentence ending '….due to the steady economic development in China"**

*Reply:*

A related reference was provided.

*Revision:*

…The emissions of $O_3$ precursors increased persistently and linearly due to the steady
economic development in China (Wang 2017)…

**Response to Reviewer #2**

**Summary and General Comments:**

**1. This study attempts to link surface O3 from a site in North China to May Arctic sea ice extent and the Eurasia Teleconnection Pattern (EUTP), which amounts to a total of three degrees of separation (May Sea Ice -> Eurasia Teleconnection Pattern -> Ozone Weather Index (OWI) -> MDA8 O3 at the SDZ site).**

**1.1 The main results of the paper seem a bit overstated because of this leap. The authors show some nice analyses linking each of these factors to one another, but the point of this paper is to show skill in predicting summer average MDA8 O3 from EUTP and May Arctic sea ice. I am not convinced the authors were so successful in that regard. For example, on Figure 6b, the association with the Sea Ice Index and MDA8 is weak at best (see 2009, 2014, 2015, and 2017), and only 11 years of JJA O3 data from a single site are used to make these claims. The authors state in the abstract that May sea ice extent explains 60% of the interannual variability in summer surface O3, but that is actually the relationship that they found between sea ice and the OWI (Lines 188-190), which has its own separate, imperfect, relationship with MDA8 O3 levels at one specific site.**

*Reply:*

(1) Possibly, our presentation was confusing. The goal of this study **was not prediction** of summer average O$_3$ MDA8. When we mentioned the predictions, we just emphasized the importance of our study to seasonal O$_3$ prediction. Furthermore, our studies, based on long-term meteorological data, **could support scientific basis and improve the potential of prediction**. This is the further meaning of our finding.

To avoid the confusing and the overstated problem, the words like "seasonal prediction" were modified or deleted.

The goal of this study was to reveal the **climatic connections** among Arctic sea ice, EU pattern, and surface ozone pollution. To enhance the theme, the title was also revised as "Arctic sea ice, Eurasia teleconnection pattern and summer surface ozone pollution in North China*: in terms of climate variability*".

(2) Another overstated expression, i.e., making OWI≈surface O3 pollution, was also corrected throughout the manuscript. In the Discussion, we also mentioned it as "**the OWI was still a substitution focusing on the impacts of the weather conditions**" and "Thus, the results in this study **concentrated on and emphasized the meteorological and climate factors**".

(3) Lines 188-190: the related Figure S11 was moved to the main text, i.e., Figure 11. This Figure 11 was to show the contribution of May sea ice, and **was not related to the seasonal prediction**. Although the generalized additive model could introduce linear and nonlinear relationships, the red line was fitted from historical May sea ice index and did not include any prediction.

[Figure]

Figure 11. The variation in the observational OWI (black) and the fitted OWI by the generalized additive model (red) from 1979 to 2017

*Revision:*

**The seasonal prediction was deleted, as follows:**

attributed to Arctic sea ice to the north of Eurasia. This finding will aids in understanding the interannual variation of O3 pollution, specially the related meteorological conditions.the seasonal prediction of O₃ pollution.

effective driver, were was also analyzedanalysed. The outcomes of our research, in term of climate variability, may provide a basis for understanding the interannual variation of O₃ pollution, specially the related meteorological conditions. and its seasonal to interannual prediction.

**The analysis focused on the O₃-related weather conditions, such as……**

…Furthermore, due to the close connection between the preceding May Arctic sea ice and summer EU pattern, approximately 60% of the interannual variability of **O₃-related weather conditions** was attributed to Arctic sea ice to the north of Eurasia…

…This finding will aids in understanding the interannual variation of $O_3$ pollution, **specially the related meteorological conditions**…

"the linear and nonlinear relationships were both introduced using the generalized additive model (Figure 11), and the contribution of May sea ice to the interannual variability of **OWI** was approximately 60%."

…In order to extend the time range of this study, the OWI was constructed in North

China. Although the feasibility of the construction approach was strictly examined, **the OWI was still a substitution focusing on the impacts of the weather conditions**. When discussing the impacts of atmospheric circulations, the linear trend was removed to weaken the signal of anthropogenic emissions. Thus, **the results in this study concentrated on and emphasized the meteorological and climate factors**. However, there is no doubt that the polluted emissions are the fundamental inducement of the surface O3 pollution…

**1.2 In general, I found the statistical analysis and OWI formulation in the first half of the paper to be more enlightening than the second half. Perhaps a more careful discussion on EUTP and the sea ice effect will lead to a more convincing paper - Sections 4 and 5 are barely two pages of double-spaced text.**

*Reply:*

(1) In the revised version, the section 4&5 lengthened about 50**%, i.e., from 60 lines to 90 lines.** The number of the Figures were **7** in the revised version, instead of 4 in the old version, in the main body.

(2) What's more important is the causality between May sea ice and OWI was verified by **a new numerical experiment by CAM5**. The proposed relationship and physical mechanisms were **reproduced by the well-designed experiment** (Figure 12). The details can be found the attached revised texts.

*Revision:*

The causality, i.e., the preceding May sea ice anomalies contributed to the subsequent JJA OWI in North China, was also confirmed by CAM5. During the control experiment (CTRL), the CAM5 model firstly integrated 20 years with climate mean initial and boundary conditions, and then, integrated 10 years with each 1st September of the last 5 years (i.e., five slightly different initial conditions). The JJA mean results of the last 6 years (i.e., 6 years $\times$ 5 groups = 30 ensembles) were employed as the output of the CTRL. On the basis of CTRL, the May sea ice concentration in the two boxes of Figure 10a was separately reduced by 10% (denoted as LowASI experiments), i.e., totally 30 sensitive runs. Similarly, the JJA mean results of 30 sensitive ensembles were employed as the output of the LowASI. The differences (LowASI minus CTRL) were the responses of atmospheric circulations and meteorological conditions to the declining May sea ice.

It was evident that an EU-like Rossby wave train was induced on the mid-troposphere

(Figure 12a), which propagated from the Taymyr Peninsula (–), Northeast China (+), to east of China and the west Pacific (+). Under such large-scale atmospheric anomalies, the anomalies of relative humidity were significantly positive and resulted in denser low and cloud cover in North China (Figure 12d). Furthermore, the cover of cloud efficiently prevented the solar radiation from reaching the land surface, meanwhile, cooled the air in the boundary layer (Figure 12b). Without hot-dry air and violent sunshine, the photochemical reaction was significantly decelerated and the generation of surface $O_3$ was rather weak. On the other side, sufficient moisture and clouds caused more rainfall (Figure 12c). The wet deposition effect was also significantly enhanced. Thus, corresponding to less Arctic sea ice in May, the photochemical process to generate $O_3$ was weakened, but the wet deposition effect to decrease $O_3$ was enhanced. That is, the positive relationship and associated physical mechanisms were causally verified.

[Figure]

Figure 12. Composite results of the LowASI experiments (LowASI minus Ctrl) by the CAM5 model: (a) geopotential height at 500 hPa, (b) preciptation, (c) net radiative flux at the top of the atmosphere (shading) and temperature at 925 hPa (contour), and (d) sum of low and medium cloud fraction (shading) and relative humidity at 925 hPa (contour). The black hatching denotes the differences with shading were above the 95% confidence level (t-test).

**2. There is a wealth of information buried in the Supplemental Figures, which is often frustrating to a lot of readers. Figures S1 and S6 in particular receive a lot of attention in the text, and therefore should be included as regular figures. Figure S11 is of great importance to the result stated in the second last sentence of the abstract (though I have separate issues with it as I mention above). Several of the Supplemental Figures should be moved to the main paper.**

*Reply:*

(1) The Figure S1, S6, S7, S8 and S11 were **moved to the main text** in the revised version. Now, there were 12 Figures in the main text and 6 Figures in the supplemental information.

(2) In the revised version, the main texts of this article lengthened about **35%, i.e., from 220 lines to 290 lines.**

(3) The number of the Figures were **12** in the revised version, instead of 6 in the old version, in the main body.

(4) Both of the description of the data and methods and the discussion of the results were **rewritten and were recognized.**

*Revision:*

The mentioned Figures were Figure 1, 3, 5, 8 and 11 in the revised manuscript.

**3. What is the motivation behind analyzing NCEP/NCAR Reanalyses in addition to ERA-Interim? This adds unnecessary supplementary figures and text. You can simply state in the text that a sensitivity test was performed with NCEP/NCAR reanalysis, which yielded very similar results (this should be expected).**

*Reply:*

(1) The motivation to include the results of two popular Reanalysis data is to show the diagnostic results were **independent of the kinds of data**. In addition, to use two kinds of data could, in some extent, **decreases the uncertainties**.

In the manuscript, we clarify the mentioned motivation, such as "the analyses and conclusions were independent of data sets".

 (2) Our research filed is a bit **interdisciplinary**, and sometime, we received the reviewer's comments that asked us to add contrastive results both from the ERA and NCEP/NCAR dataset. Thus, in the supplementary information of this manuscript, we directly submit them.

*Revision:*

... The above independent verifications proved that the performance of the summer OWI did not depend on the kinds of reanalysis data,…

…the impacts of the atmospheric circulations were confirmed by both the ERA-Interim and NCEP/NACAR data, i.e., the analyse and conclusions were independent of data sets….

**Recommendation:**

**4. In its current form, this paper is not yet suitable for publication in ACP. I am suggesting major revisions that include *(1)* reorganization of Figures, and substantial edits to the text, *(2)* mainly in the Introduction and *(3)* Sections 4 and 5. I think that the paper discussion should remain focused on the meteorological relationships found linking the teleconnection patterns and surface O3 (which are interesting and useful to quantify!), and *(4)* less on claims of seasonal predictability that may not be warranted by the current study.**

*Reply:*

The manuscript has been revised according **to the ACP format**. The title was also revised as "**Arctic sea ice, Eurasia teleconnection pattern and summer surface ozone pollution in North China: in terms of climate variability**"

(1) In the revised version, the main texts of this article lengthened about **35%, i.e., from 220 lines to 290 lines.** The number of the Figures were **12** in the revised version, instead of 6 in the old version, in the main body.

(2) The **introductions were entirely revised and rewritten** now. ①The introduction of European clean air laws was confusing, and now was deleted. Thus, the introduction of the ozone polluted features f**ocused on those in China.** ② In the second paragraph of the introduction, the ideas how the climate anomalies (e.g., NAO, jet stream, west Pacific subtropical high and East Asia summer monsoon) were **expanded on**. The details can be found in the following revisions attached. ③ Due to insufficient studies, related to how the climate anomalies impacted the ozone pollutions in China, some closely findings in North American were still introduced. Indeed, the findings, such as NAO-ozone in Europe and jet stream-ozone on North American, provided meaningful and substantial clues to our studies.

(3) In the revised version, the section 4&5 lengthened about 50**%, i.e., from 60 lines to 90 lines.** What's more important is the causality between May sea ice and OWI was verified by **a new numerical experiment by CAM5**. The proposed relationship and

physical mechanisms were **reproduced by the well-designed experiment** (Figure 12). The details can be found the attached revised texts.

(4) The goal of this study **was not prediction** of summer average MDA8 $O_3$. When we mentioned the predictions, we just emphasized the importance of our findings to seasonal $O_3$ prediction. Furthermore, our studies, based on long-term meteorological data, **could support scientific basis and improve the potential of prediction**. This is the further meaning of our finding. To avoid the confusing and the overstated problem, the words like "seasonal prediction" were modified or deleted.

(5) Throughout the manuscript, the writing was corrected to focus on the meteorological conditions related to $O_3$ production. In the Discussion, we also mentioned it as "**the OWI was still a substitution focusing on the impacts of the weather conditions**" and "Thus, the results in this study **concentrated on and emphasized the meteorological and climate factors**".

**Revision:**

(1) The revised manuscript without and with tracks were both uploaded for review.

(2) …For example, the prevailing positive phase of the North Atlantic Oscillation contributed to the increasing ozone concentration in western and northern Europe, through the anomalous atmospheric circulations to influence regional photochemical processes (Christoudias et al., 2012; Pausata et al., 2012)…

…The summer surface $O_3$ variability in North America is significantly modulated by the position of the jet stream (Lin et al., 2014). Barnes and Fiore (2013) pointed out jet position may dynamically modulate surface ozone variability in eastern North America and other northern mid-latitude regions…

…A strong positive correlation between the East Asian summer monsoon and summer mean ozone existed. The model simulation by Yang et al. (2014) illustrated that the changes in meteorological parameters, associated with East Asian summer monsoon, lead to 2–5% interannual variations of surface $O_3$ concentrations over central eastern China. Focusing on the dataset in 2014, a significantly strong west Pacific subtropical high resulted in higher relative humidity, more clouds, more rainfall, less ultraviolet radiation and lower air temperatures, which were unfavourable for the formation of $O_3$ (Zhao and Wang, 2017)…

(3) The revisions for section 4&5 can be found in the replies to Comment 1.2.

(4) The revisions for 4-(4) (5) can be found in the replies to Comment 1.1.

**Specific/Minor Comments:**

**Line 10: Surface ozone pollution is only increasing in certain parts of the world. Please be specific about China here.**

*Reply:*

The error has been corrected.

*Revision:*

Summer surface $O_3$ pollution has rapidly intensified **in China** recently, damaging human and ecosystem health.

Introduction: I find the discussion on "haze pollution" to be very confusing, especially because particulate pollution is not the focus in this paper. It would make the introduction much clearer to eliminate the use of "haze" and discuss only the O3 pollution issues in China.

*Reply:*

We mentioned the "haze pollution" to contrastively show that the haze is decreasing, however the ozone pollution is increasing and lack of research. The confusing writing was improved to **focus on the comparison and make it focus on the surface ozone pollution.**

*Revision:*

…Due to drastic air pollution control in China since 2013, **haze pollutions are being controlled in recent years** (The environmental statistics unit of stat-centre in Peking University, 2018), appearing as sharp decreasing in fine particulate matter ($PM_{2.5}$). **However, surface $O_3$ pollution**, which always occurred on clear and sunny days (Wang et al., 2017), **has not improved** (Li et al., 2018). The negative effects of surface $O_3$ pollution was not weaker than those of haze (Liu et al., 2018), but **the impacts of climate variability on surface $O_3$ pollution in China (Yang et al, 2014) have not been sufficiently studied.** In the major urban agglomerations in China…

**Lines 26-27: Surface O3 is a major component of photochemical smog, so it is actually very frequently visible to humans when found in high amounts.**

*Reply:*

According to the reviewer's advice, the discussion about the visibility was **deleted**.

*Revision:*

Due to drastic air pollution control in China since 2013, haze pollutions are being controlled in recent years (The environmental statistics unit of stat-centre in Peking University, 2018), appearing as sharp decreasing in fine particulate matter ($PM_{2.5}$). However, surface $O_3$ pollution, which always occurred on clear and sunny days (Wang et al., 2017), has not improved (Li et al., 2018).

**Lines 27-28: I disagree with this sentence. Surface O3 pollution in China is now a heavily studied topic. Rewrite this sentence to mainly highlight the novel aspect of your research into this topic: "The impacts of climate variability on surface O3 pollution in China have not been sufficiently studied."**

*Reply:*

The error was corrected, according to the reviewer's comment.

*Revision:*

…but **the impacts of climate variability on surface $O_3$ pollution in China** (Yang et al, 2014) have not been sufficiently studied…

**Lines 28-29: This sentence on European pollution controls is out of place. Find a local example of pollution controls in China to discuss or remove this sentence entirely.**

*Reply:*

The introduction of European clean air laws was confusing, and now was **deleted**. Thus, the introduction of the ozone polluted features f**ocused on those in China.**

*Revision:*

Line 30: Please quote the current Chinese air quality standards for surface O3.

*Reply:*

The air quality standards was added in the manuscript.

*Revision:*

In the major urban agglomerations in China, such as Beijing-Tianjin-Hebei, Yangtze River delta and the Pearl River delta, the surface $O_3$ concentrations exceeded the ambient air quality standard **of China (i.e., 100 µg/m³)** by 100–200 % (Wang et al., 2017).

**Lines 35-40: These examples of large-scale atmospheric circulation patterns affecting surface O3 are relevant, but need to be presented more clearly to allow the reader to understand why they are worth discussing.**

*Reply:*

In the second paragraph of the introduction, the ideas how the climate anomalies (e.g., NAO, jet stream, west Pacific subtropical high and East Asia summer monsoon) were **expanded on**. The details can be found in the following revisions attached.

*Revision:*

…For example, the prevailing positive phase of the North Atlantic Oscillation contributed to the increasing ozone concentration in western and northern Europe, through the anomalous atmospheric circulations to influence regional photochemical processes (Christoudias et al., 2012; Pausata et al., 2012)…

…The summer surface $O_3$ variability in North America is significantly modulated by the position of the jet stream (Lin et al., 2014). Barnes and Fiore (2013) pointed out jet position may dynamically modulate surface ozone variability in eastern North America and other northern mid-latitude regions…

…A strong positive correlation between the East Asian summer monsoon and summer mean ozone existed. The model simulation by Yang et al. (2014) illustrated that the changes in meteorological parameters, associated with East Asian summer monsoon, lead to 2–5% interannual variations of surface $O_3$ concentrations over central eastern China. Focusing on the dataset in 2014, a significantly strong west Pacific subtropical high resulted in higher relative humidity, more clouds, more rainfall, less ultraviolet radiation and lower air temperatures, which were unfavourable for the formation of $O_3$ (Zhao and Wang, 2017)…

**Lines 40-47: Similar comment as the one above. There are several studies discussed in one sentence each, but the organization makes this cumbersome to read. In general, the Introduction would benefit from a careful rewrite.**

Lines 47-52: This paragraph is a good example of how the rest of the Introduction should be written to motivate carrying out this study.

*Reply:*

Lines 40-47 were also **expanded on**. The details can be found in the following revisions attached.

*Revision:*

…The photochemical reaction was the main local sources of $O_3$. The hot and dry environments and the violent solar radiation could accelerate the chemical conversion from the precursor to $O_3$ (An et al., 2009; Tong et al., 2017). In 2013, a severe heat wave, with highest temperature 41.1 ℃, contributed to the high $O_3$ concentration in the Yangtze River Delta (Pu et al., 2017). The frequency of large-scale, extreme heat events is closely related to atmospheric patterns, such as the Eurasia teleconnection pattern (EU; Pu et al., 2017; Li and Sun, 2018) and aerosol effective radiative forcing (Liu and Liao, 2017). The winds from a polluted area also transport $O_3$ and its precursors downwind (Doherty et al., 2013). Due to the close relationship between surface $O_3$ and meteorological conditions, the impacts of climate change on $O_3$ have been projected by various numerical models (Doherty et al., 2013; Melkonyan and Wagner, 2013; Zhu and Liao, 2016; Gaudel et al., 2018). Over eastern China, the surface ozone concentration and possibility of severe ozone pollution may both increase in the future (Wang et al., 2013)…

**Line 79: The first figure introduced should not be supplemental figure. Figure S1 is discussed extensively in the text, so it would be useful to just make this Figure 1.**

**Figure S6: Much like Figure S1, this is a lot of text to dedicate to a figure that is buried in the Supplemental Information.**

*Reply:*

The Figure S1, S6, S7, S8 and S11 were **moved to the main text** in the revised version. Now, there were 12 Figures in the main text and 6 Figures in the supplemental information.

*Revision:*

The mentioned Figures were Figure 1, 3, 5, 8 and 11 in the revised manuscript.

**Figure S1 Caption: Green "triangle", not "triple"**

*Reply:*

The error is corrected.

*Revision:*

The green **triangle** in panels b, d, f, and h illustrate the location of the SDZ station. The black box in panel h is the range of North China.

**Line 98: Are these statistics only for June to August? Please be clear.**

*Reply:*

It is for June to August.

*Revision:*

During the years 2007–2017, there were 126 NOP days and 155 MOP days **in summer** at SDZ station.

**Line 101: Clarify that you are still referring to the O3 levels at SDZ.**

*Reply:*

This sentence was clarified.

Throughout the manuscript, the MDA8 in SDZ station was denoted as **SDZ MDA8**, comparing to the **MDA8 in the other sites**.

*Revision:*

…although the meteorological conditions were **composited for the MOP and NOP days in SDZ** (Figure 2), the results were also appropriate for those in North China…

2017). The correlation coefficients between SDZ MDA8 and the observed MDA8 at the other sites were calculated and are shown in Figure 1 (b, d, f, h). The distribution of correlation coefficients is similar to the MDA8 on Figure 1 (a, c, e, g). The SDZ MDA8 significantly covaried with the MDA8 in North China in summer. Along with the increasing of the surface O3 pollution, the covariation and the representativeness of SDZ MDA8 to the MDA8 in North China was strengthened. However,

**Figure 1, panels g and h: Clarify in the figure caption which temperature is contoured and which is shaded in color. Does geopotential height at 500 or 250 hPa**

**tell you more about surface O3 variability than 200 hPa temperature?**

*Reply:*

(1) The negligence was revived. **The SAT is with shading and the temperature at 200 hPa is contoured.**

(2) We also tried 500 hPa and the lower levels, which was positive and similar with the surface temperature. The temperature anomalies at 200 hPa was opposite with those below 300 hPa. 250 hPa was the transitional layer and the features was not as clear as 200 hPa. Therefore, **we chose the temperature difference between surface and 200 hPa.**

*Revision:*

Figure 2…(g–h) SAT (shading), and temperature at 200 hPa (contour)…

**Lines 105-106: Are the northerly winds associated with higher surface O3 to the south of SDZ?**

*Reply:*

(1) The ozone concentration in North China, most of where were located to the south of SDZ station. Due to the covariation (Figure 1), **the northerlies dispersed the $O_3$ precursors in North China, and the surface $O_3$ concentration was reduced**. As for the influenced area, it is depended on **the intensity of the northerlies**. In total, statistically, associated with the northerlies, the surface $O_3$ concentration reduced in North China. (2) We did not carefully examine the impacts of northerly wind on the ozone concentrations to the south of North China (i.e., far away from SDZ). Possibly, the influenced factors were different. According to the new publication by **Li et al (2018), the sensitive meteorological factors related to the ozone pollution in Yangtze River Delta was relative humidity, zonal and meridional wind, which was different with those in North China.**

Li, K., Jacob, D. J., Liao, H., Shen, L., Zhang, Q., Bates, K. H.: Anthropogenic drivers of 2013–2017 trends in summer surface ozone in China, P NATL ACAD SCI USA., https://doi.org/10.1073/pnas.1812168116, 2018

Line 108: Here and a couple of other locations, please fix the degree symbols.

*Reply:*

The similar errors were corrected throughout the manuscript.

*Revision:*

was reduced (Figure 2b). The correlation coefficient between the SDZ $O_3$ concentration and the area-averaged meridional wind at 10 m (35–50°N, 110–122.5°E, denoted as V10mI) was 0.39, exceeding the 99% confidence level. The cloudy skies and precipitation weakened the photochemical reaction by influencing exposure to ultraviolet rays. In addition, precipitation was also an important indicator of the wet removal efficiency (Figure 2f). In summer, a day without rain represents efficient solar radiation, in favor of the occurrence of surface $O_3$ pollution (Figure 2e). The correlation coefficient between the area-averaged precipitation (37.5–42.5°N, 112–127.5°E, denoted as PI) and the SDZ $O_3$ concentration was –0.35 (above the 99% confidence level), indicating that precipitation was connected with more NOP days.

In contrast, high SAT enhanced the photochemical reactions and resulted in higher surface $O_3$ concentrations (Figure 2 g). The correlation coefficient between the area-averaged difference in the temperature at the surface and 200 hPa (SAT minus temperature at 200 hPa, 37.5–47.5°N, 110–122.5°E, denoted as DTI) and the SDZ $O_3$ concentration was 0.49. Furthermore,

**Lines 108-112: Please rewrite these sentences to more simply state that clouds and precipitation are unfavorable for surface O3 production, leading to the meteorological composite in Figure 1f.**

*Reply:*

This sentence was revised.

*Revision:*

**The cloudy skies and precipitation weakened the photochemical reaction by influencing exposure to ultraviolet rays**. In addition, precipitation was also an important indicator of the wet removal efficiency (Figure 2f).

**Line 116-117: The temperature of the upper troposphere is much more dynamicallythan radiatively-influenced at synoptic timescales (i.e. through tropopause height variations). This sentence should be removed.**

*Reply:*

According to the reviewer's suggestion, the discussion about the reason, why cooler temperature in the high troposphere (T200) is favorable for surface ozone pollution, was **deleted**.

Because we cannot perfectly address it now, the question was leave as an open question in the "Conclusion and Discussion" section.

*Revision:*

…The processes how the weather conditions impacted the photochemical reaction were not deeply discussed here and have been analyzed in many previous studies by the atmospheric chemists. **However, the reason why the cooler high troposphere contributed to the surface ozone pollution was still an open question and needed further attention**…

**Line 136: Were there any sites that had a larger correlation between MDA8 and OWI than the SDZ site?**

*Reply:*

Although the correlation coefficients between the OWI and observed MDA8 at the other sites were significantly positive in North China (Figure 5), the CC, which were **was larger than that in SDZ station, were few (Figure R1).**

[Figure]

Figure R1. The correlation coefficients between the daily MDA8 and OWI from 2014 to 2017. **The black crosses indicate that the CC was larger than that in SDZ station.** The black box in panel d is the range of North China.

**Lines 140-152 and Figure 3: I think analysis of OWI before 2006/2007 is not necessary, and introduces the uncertainty of a changing observing system (i.e. the satellite era beginning in 1979, ATOVS in 1998, etc.). These discussions are certainly outside of the scope of this paper, so I recommend not extending the OWI to periods when there are no O3 observations to support your analyses. Limit your discussion to 2007 to the present.**

*Reply:*

(1) Most of the ozone observations were from 2014 to 2017 (Figure 1). Even at the SDZ station with measurements from 2006 to 2017, **the range of data was insufficient for the climate research**. Generally, the climate research required at least data for **30 years**.

(2) In this study, in addition to reveal the related meteorological conditions, we try to connect the ozone concentration with the climate anomalies. Thus, we calculated the daily OWI and then gained the **monthly mean OWI from 1979 to 2017**, which was treated as a substitution focusing on the impacts of the weather conditions.

(3) Once, we constructed the long-term OWI, discussion about the relationship with Arctic sea ice and large-scale atmospheric circulations became possible. This is the basis for the Section 4 & 5 in the manuscript, because **we cannot link the MDA8 with sea ice via short time series**.

(4) Although the goal of this study was not seasonal prediction, the findings potentially improve the possibility of seasonal prediction. Actually, the emissions linearly increased in recent year, **the annual incremental method could deduce the signal of emission and emphasized the climatic component and make the seasonal prediction feasible.**

**The authors had successful experience in the seasonal predictions of haze pollution (Yin and Wang 2016, 2017).**

Yin Z. C. , Wang H. J., 2016, Seasonal Prediction of Winter Haze Days in the North-

Central North China Plain,Atmos. Chem. Phys.,60(15):1395~1400

Yin Z. C. , Wang H J. 2017. Statistical Prediction of Winter Haze Days in the North China Plain Using the Generalized Additive Model. Journal of Applied Meteorology and Climatology. 56:2411–2419

**Figure 3: All of the other figures that show SDZ data begin in 2007. Why does this include 2006, and why is the line a different color green here? Why does the**

**NCEP/NCAR data end in 2014? Please clarify.**

*Reply:*

(1) The data form 2017-2017 were the **training set**, while the data in 2006 was th**e test set**. "Although the range of the SDZ MDA8 was 2006–2017, only the data from 2007 to 2017 were used in the above OWI construction processes. Thus, the datasets in 2006 were **independent samples**, and could verify the performance of the OWI."

(2) In the section 2, "The BLH of NCEP/NCAR dataset was only available from 1979 to 2014 in the website of the NOAA-CIRES 20th Century Reanalysis version 2c (Giese et al., 2016)". **Due to the absence of BLH after 2014, thus, the OWI from NCEP/NCAR data was limited from 1979-2014.** It's important to note that the other variables used in this study was from 1979 to 2017.

Detailed explanation was added in the revised manuscript.

*Revision:*

…The monthly OWI was computed as the monthly mean of the daily OWI. During 2007–2017, the constructed JJA (June-July-August) mean OWI varied similarly with the observed MDA8 and captured the extremes (Figure 6). Although the range of the SDZ MDA8 was 2006–2017, only the data from 2007 to 2017 were used in the above OWI construction processes. Thus, the datasets in 2006 were independent samples, and could verify the performance of the OWI. The JJA mean OWI in 2006 successfully reflected the variation in observed MDA8; even the MDA8 in 2006 was the minimum, confirming the robustness of the OWI…

**Figures 4 and 5: I thought the OWI was constructed by using some of these same meteorological variables. What is the value in calculating correlations between these variables and OWI?**

*Reply:*

To answer this question, the **daily weather process** and **climate anomalies** must be distinguished. (1) The construction of the daily OWI was based on the **weather conditions**. (2) The daily OWI from 1979 to 2017 were calculated to obtain the climatic time series, i.e., the JJA mean OWI from 1979 to 2017. (3) The correlation coefficient between the JJA mean OWI and meteorological conditions were calculated from 1979 to 2017 to reveal the climatic connection.

**Although there were similarity between the related weather conditions and the climatic anomalies, their meanings were substantially different.**

(4) Based on the climatic connection, the contribution of May sea ice was studied, which cannot be supported by the weather analysis.

To enhance the theme, the title was also revised as "Arctic sea ice, Eurasia teleconnection pattern and summer surface ozone pollution in North China*: in terms of climate variability*".

**Lines 170-177: This discussion is essentially a repeat of Lines 110-126. What is new here?**

*Reply:*

The response can referred the above reply of comment related to **Figures 4 and 5.**

**Although there were similarity between the related weather conditions and the climatic anomalies, their meanings were substantially different.**

**Lines 185-186: What sea ice variable is used here? Is this sea ice extent (generally referred to as ocean areas with at least 15% ice coverage)?**

*Reply:*

The variable is the sea ice **area**.

*Revision:*

The averaged (green boxes in Figure 10a) **sea ice area** in May was calculated as the SI index, whose linear correlation coefficient with JJA OWI was 0.67 (after detrending) from 1979 to 2017.

Line 188: Was the MDA8 anomaly also detrended? Is the MDA8 anomaly an anomaly from the mean of all JJA 2007-2017 MDA8 values?

*Reply:*

The MDA8 anomalies was the original value **subtracted the mean** of all JJA 2007-2017 MDA8 values

**Lines 205-207: As I understand your analysis, this statement is not correct (same with the abstract). The May sea ice extent contributes to about 60% of the interannual variability of OWI, not surface O3 pollution. There is a separate**

relationship between OWI and MDA8 at SDZ to be considered. The May Arctic sea ice anomaly does not even have the same sign as the MDA8 anomaly about 30% of the time (Line 187).

*Reply:*

Another overstated expression, i.e., making OWI≈surface O3 pollution, was also corrected throughout the manuscript. In the Discussion, we also mentioned it as "**the OWI was still a substitution focusing on the impacts of the weather conditions**" and "Thus, the results in this study **concentrated on and emphasized the meteorological and climate factors**".

*Revision:*

The analysis focused on the $O_3$-related weather conditions, such as……

…Furthermore, due to the close connection between the preceding May Arctic sea ice and summer EU pattern, approximately 60% of the interannual variability of **$O_3$-related weather conditions** was attributed to Arctic sea ice to the north of Eurasia…

…This finding will aids in understanding the interannual variation of $O_3$ pollution, **specially the related meteorological conditions**…

"the linear and nonlinear relationships were both introduced using the generalized additive model (Figure 11), and the contribution of May sea ice to the interannual variability of **OWI** was approximately 60%."

…In order to extend the time range of this study, the OWI was constructed in North China. Although the feasibility of the construction approach was strictly examined, **the OWI was still a substitution focusing on the impacts of the weather conditions**. When discussing the impacts of atmospheric circulations, the linear trend was removed to weaken the signal of anthropogenic emissions. Thus, **the results in this study concentrated on and emphasized the meteorological and climate factors**. However, there is no doubt that the polluted emissions are the fundamental inducement of the surface O3 pollution…

**Technical Corrections:**

**Line 15-16: Clarify by rewriting as "Increased solar radiation and high temperatures during the EUTP positive phase dramatically enhanced O3 production."**

*Reply:*

According to the reviewer's comment, the sentence was **rewritten**.

*Revision:*

Increased solar radiation and high temperature during the positive EU phase dramatically enhanced $O_3$ production.

**Lines 24-25: See comments on the use of "haze" in my Specific Comments. This sentence should be removed or at least rewritten.**

*Reply:*

We mentioned the "haze pollution" to contrastively show that the haze is decreasing, however the ozone pollution is increasing and lack of research. The confusing writing was **improved to focus on the comparison and make it focus on the surface ozone pollution.**

*Revision:*

…Due to drastic air pollution control in China since 2013, **haze pollutions are being controlled in recent years** (The environmental statistics unit of stat-centre in Peking University, 2018), appearing as sharp decreasing in fine particulate matter ($PM_{2.5}$). **However, surface $O_3$ pollution**, which always occurred on clear and sunny days (Wang et al., 2017), **has not improved** (Li et al., 2018). The negative effects of surface $O_3$ pollution was not weaker than those of haze (Liu et al., 2018), but **the impacts of climate variability on surface $O_3$ pollution in China (Yang et al, 2014) have not been sufficiently studied.** In the major urban agglomerations in China…

**Line 25: Peking University?**

*Reply:*

According to the reviewer's comment, the error was corrected.

*Revision:*

Due to drastic air pollution control in China since 2013, haze pollutions are being controlled in recent years (The environmental statistics unit of stat-centre in **Peking** University, 2018), appearing as sharp decreasing in fine particulate matter ($PM_{2.5}$).

**Line 32: Rewrite this sentence "Surface O3 is a secondary pollutant."**

*Reply:*

According to the reviewer's comment, the sentence was **rewritten**.

*Revision:*

Surface $O_3$ is a secondary pollutant.

**Line 33: Change "Sedimentation" to "Surface deposition" and "attenuation" to "dispersion"**

*Reply:*

According to the reviewer's comment, the sentence was rewritten.

*Revision:*

Surface deposition, dynamic transport and dispersion of $O_3$ are also closely related to atmospheric circulations.

Line 50: Change "were" to "was"

*Reply:*

According to the reviewer's comment, the sentence was rewritten.

*Revision:*

The role of May Arctic sea ice, as a preceding and effective driver, **was** also analysed.

Line 64: Do you mean the NCEP/NCAR Reanalysis? Please refer to these products as the NCEP/NCAR Reanalysis, instead of NOAA.

*Reply:*

The expression of NOAA data has been changed to the **NCEP/NCAR data**.

*Revision:*

The daily mean and monthly mean ERA-Interim data were directly downloaded from the ERA-Interim websiteanalyzed in this study. Furthermore, the daily mean and monthly reanalysis datasets supported by the National Oceanic and Atmospheric Administration (NOAA) were also employed and denoted as NOAA NCEP/NCAR (National Center for Environmental Prediction and the National Center for Atmospheric Research) data. The 2.5°×2.5° geopotential height (Z), zonal and meridional wind, relative humidity, vertical velocity, air temperature at different pressure levels, SAT and wind, downward UV radiation, downward solar radiation, low and medium cloud cover were downloaded from the National Center for Environmental Prediction and the National Center for Atmospheric Research (Kalnay et al. 1996). The BLH of NCEP/NCAR dataset was only available from 1979 to 2014 in the NOAA data was derived fromthe website of the NOAA-CIRES 20th Century Reanalysis version 2c (Giese et al., 2016). The daily precipitation data was from the CPC global analysis of the daily precipitation dataset (Chen et al., 2008).

Line 82: "appeared to be delineated by the Yangtze River."

*Reply:*

According to the reviewer's comment, the sentence was **rewritten**.

*Revision:*

…$O_3$ concentrations in the high-mid latitudes were higher than those in the lower latitudes, which appeared to be **separated** by the Yangtze River…

Line 84: Delete "approximately yearly."

*Reply:*

**Deleted**

*Revision:*

that time, the $O_3$ polluted region has expanded approximately yearly. ]

**Line 86: Replace "almost higher than" with "nearly." Also, what is the threshold of severe surface O3 pollution? Is this a definition set by the Chinese government? Are these O3 data publicly available?**

*Reply:*

(1) According to the reviewer's comment, the sentence was **rewritten**.

(2) The ozone data from 2014 to 2017 were **publicly available in the website** of the Ministry of Environmental Protection of China.

*Revision:*

higher than 120 μg/m³.   existed in the south of Hebei Province and the north of Shandong Province (Figure 1a). Since that time, the O₃ polluted region has expanded . In 2017, the areas with summer mean MDA8 > 120 μg/m³ were visibly enlarged. In North China, the summer mean MDA8 observations were larger than 150 μg/m³, and the maximum MDA8 was nearly 265 μg/m³ . South of the

**Line 92: Rewrite this sentence, suggestion: "The distribution of correlation coefficients is similar to the MDA8 on Figure S1 panels a, c, e, and g."**

*Reply:*

According to the reviewer's comment, the sentence was **rewritten**.

*Revision:*

The distribution of correlation coefficients is similar to the MDA8 on Figure 1 (a, c, e, g).

**Line 93: There is an extra period after China.**

*Reply:*

It is in **summer**.

*Revision:*

The SDZ MDA8 significantly covaried with the MDA8 in North China in summer.

**Line 94: Change "diurnal" to "daily"**

*Reply:*

The error was corrected.

*Revision:*

The **daily** difference in MDA8 was large, which contradicts the quasi-constant emission of ozone precursors.

**Line 96 and in the references: Should be "Ministry of Environmental Protection"**

*Reply:*

The error was corrected.

*Revision:*

(The Ministry of Environmental **Protection** of China, 2012)

**Line 97: What is "nonsurface?" Please rename this to something like "non-polluted surface O3 levels".**

*Reply:*

The "**nonsurface**" was renamed as non-surface $O_3$ polluted (NOP) level.

*Revision:*

The thresholds of non-surface $O_3$ polluted (NOP) level and moderate surface $O_3$ polluted (MOP) level are 100 $\mu g/m^3$ and 215 $\mu g/m^3$, respectively.

Line 114: Delete "sufficient"

*Reply:*

The "sufficient" was **deleted**.

*Revision:*

…indicating that precipitation was connected with more NOP days….

Lines 115-116: Rewrite: "In contrast, higher SAT enhances the photochemical reactions and resulted in higher surface O3 concentrations (Figure 1g)."

*Reply:*

According to the reviewer's comment, the sentence was **rewritten**.

*Revision:*

In contrast, high SAT enhanced the photochemical reactions and resulted in higher surface $O_3$ concentrations (Figure 2 g).

**Line 121: Is this referring to the entrainment of O3 into the boundary layer? Please clarify and eliminate the word "downwash"**

*Reply:*

According to the reviewer's comment, the sentence was **rewritten**.

*Revision:*

The entrainment of atmospheric ozone from the upper air into the boundary layer enlarged the surface O$_3$ concentration

Line 126: Use NCEP/NCAR reanalysis.

*Reply:*

According to the reviewer's comment, the sentence was **corrected**.

*Revision:*

…the above composite analysis was repeated with NCEP/NCAR reanalysis data, and identical results were obtained…

**Line 132: Delete "In contrast"**

*Reply:*

The "**In contrast**" was **deleted**.

*Revision:*

measured MDA8 during 2007–2017 (i.e., 92 days × 11 years). The correlation coefficient between the observed MDA8 and daily OWI was also 0.61 for the 11 year period. Thus, the OWI was easily constructed by accumulating the

**Figure S5: Either add a second panel with the ERA-Interim data for comparison or remove this figure. Consider also my comments about the inclusion of NCEP/NCAR Reanalysis in this paper in general.**

*Reply:*

**Add a second panel** with the ERA-Interim data for comparison.

*Revision:*

[Figure]

**Figure S3.** Differences in the boundary layer height between the MOP and NOP events during 2007–2014, basing on the ERA-Interim (a) and NCEP/NCAR datasets (b). The black dots denote that the composite passed the 95% confidence level. The boxes represent the area to calculate the daily OWI. These composites were calculated using the NOAA datasets.

Line 141: NCEP/NCAR reanalysis.

*Reply:*

According to the reviewer's comment, the sentence was **corrected**.

*Revision:*

Here, the daily OWI was calculated with both ERA-Interim and NCEP/NCAR reanalysis data from 1979.

Figure S11 Caption: "1979 to 2017", correct?

*Reply:*

Yes, it is 1979 to **2017**.

*Revision:*

Figure 11. The variation in the observational OWI (black) and the fitted OWI by the generalized additive model (red) from 1979 to 2017

---

## Referee Report (RR1)

Review of "Arctic sea ice, Eurasia teleconnection pattern and summer surface ozone pollution in North China:  in terms of climate variability"

The authors have greatly improved the manuscript responding to the three reviewers' comments.  I believe this paper is suitable for publication after minor revisions.  My two main concerns are that the NOP and MOP need to be clearly defined as a range of O3 concentrations.  This is not clear to me and thus how Figure 2 is calculated is not clear to me either.   I am troubled by two statements in the conclusion section:

If "the processes how the weather conditions impact the photochemical reaction were not deeply discussed here" then I do not understand what was shown in this paper.

"However, the reason why the cooler high troposphere contributed to the surface ozone pollution was still an open question and needed further attention".  If this is true, then why did the two temperature fields get compared in the first place? What meteorological understanding would lead this to be shown in Figure 2?

I have inserted these specific comments at the appropriate line number in my Minor and Technical Comments below.

Two additional references that have recently been published may be of interest to the authors to include in their introduction:

*Int J Climatol.*
Inter-annual variation of the spring haze pollution over the North China Plain: Roles of atmospheric circulation and sea surface temperature
By Chen S, Guo J, Song L, Li J, Liu L, Cohen JB.
https://rmets.onlinelibrary.wiley.com/doi/10.1002/joc.5842

ACP
Impacts of meteorology and emissions on summertime surface ozone increases over central eastern China between 2003 and 2015
by Lei Sun, Likun Xue, Yuhang Wang, Longlei Li, Jintai Lin, Ruijing Ni, Yingying Yan, Lulu Chen, Juan Li, Qingzhu Zhang, and Wenxing Wang
https://www.atmos-chem-phys.net/19/1455/2019/

Minor and Technical Comments:
Line 10: define how many years is meant by "recently"
Line 12: Make sure O3 has a subscript 3 everywhere
Line 14: To have both "major" and "significant" is too much.  I recommend removing "significant"
Line 24:  This opening sentence does not make sense to me.  Has social and economic development been serious in China?
Line 25: Explain "haze" pollution.  Is this from particulate matter or from a mix of other pollutants as well?
Line 28: Change decreasing to decrease.

Line 28: Why?  Are the controls specific to winter or only targeting what might impact haze?  I could imagine some controls might have also helped the O3 precursors.

Line 29:  Be specific, are "the negative effects of surface O3 pollution" regarding human health, agriculture, economy, etc?

Line 32:  remove "i.e."

Lines 33-34:  Why is this important?  No explanation of ozone precursors, authors should consider explaining what those are before this statement.

Line 35: Why are both O3 and the precursors large?

Line 39:  Define NOx and VOC.  I would also suggest adding "with sunlight" after "react"

Line 47:  Add references for sentence ending in "mean ozone existed".

Line 51:  The sentence "The photochemical reaction was the man local sources of O3" is in reference to which paper?  And with NOx or VOCs?

Line 61:  Do the authors mean "previous studies of O3 pollution in China mainly focused on ….."?  Consider adding "of O3 pollution in China" or some other description of these previous studies.

Line 62:  Is there a "lack of long-term surface O3 observations" at all or publicly available?

Line 65:  change "was" to "is"

Line 66:  The authors may want to stress here that this may have possible seasonal forecast application.

Line 67: Can I suggest adding subsections 2.1 Observation-based data (Line 68) and 2.2 Model-based data (Line 78) or something like? And maybe a section 2.3 Teleconnection before Line 89

Lines 68, 69 and 72:  I am confused by the terminology of "observation duration of the surface O3 concentration was much shorter than the meteorological measurements".  Is the "much shorter" in relation to "the hourly O3 concentration data from 2014 to 2017, " or the Shangdianzi station "observed from 2006 to 2017"?  Eleven years of data versus four years of data is definitely a different story for short term vs long-term.

Line 72: make sure to add a space between "to2017".

Line 76:  The sea ice data should not be in the same paragraph as the ozone data.  Can the authors make it a new paragraph and add more details about this dataset such as the time period it covers, is this a widely used data set at that resolution?

Line 78:  Give more detail about the ERA-Interim dataset.  It is available from 1979-present.  Besides the horizontal resolution, can the authors provide the vertical resolution of the data used.

Line 83: Give more detail about the NCEP/NCAR dataset.  It is available from when to when, vertical resolution of the dataset, etc?

Line 86:  The BLH is not from the NCEP/NCAR dataset but from a different reanalysis altogether.  Describe the NOAA-20CR reanalysis (horizontal and vertical resolution, period of coverage, how it is different to NCEP/NCAR).

Line 88: Consider moving daily precip data to the 2.1 Observation section if it is an observation data set and not a model dataset.

Line 91:  I do not care for the use of "vice versa" used throughout the paper instead of writing it out what the alternative conditions may be.  Perhaps this is a technical edit to discuss with ACP.

Line 93:  Consider adding "for summertime" following "the calculation procedure for the EU index here"

Line 95: Give all equations numbers. This would then be Equation 1 and the authors can reference it in the paper as such.

Line 100: What is the top of the atmosphere for this model?

Line 107: I think change "The observations, with" to "Observations with"

Line 108: Change "server" to "severe"

Line 109: I do not agree fully with the statement "the ozone polluted region has expanded" as over time the measurement coverage has increased spatially so it is possible the other areas were already polluted but just not observed. Can the authors defend this statement?

Line 111: In the figure the crosses to indicate the ozone is above the severe threshold of 265 make the circles actually look like a darker shade of red. Is there a reason the authors chose not to use colors based on the thresholds?

Line 115: Can SDZ still be considered one of the "background monitoring stations" as the authors argue it represents urban centers?

Line 121: I do not understand why the large daily difference in MDA8 contradicts the quasi-constant emission of ozone precursors.

Line 125: Are NOP levels anything above 100 micrograms per meter cubed or less than 100? or are NOP levels between 100 to 215 and then MOP is anything above 215? The definition of NOP and MOP are not clear to me and this is VERY IMPORTANT for Figure 2.

Line 129: Why is NOP in brackets?

Line 129: Do the authors have enough data for a trend analysis?

Lines 131-133: This sentence does not make sense. Maybe remove "although" and add "and" after "(Figure 2),"

Line 132: If the composites are for MOP and NOP conditions, then what are they differenced from to get positive and negative values? It is very important to make sure the NOP and MOP definitions are made abundantly clear. If NOP is anything up to 100 and MOP is anything above 215, then are the authors using the conditions between the two (100 to 215) as the control?

Line 133: significant by which test?

Line 135: Insert references to which Figure 2 panel the reader should look at. For example, "The anomalous southerlies (Fig. 2a), higher BLH (Fig. 2c), less rainfall (Fig. 2e), warmer surface air temperature (Fig. 2g), and cooler temperature in the high troposphere (Fig. 2g) favored surface O3 pollution (i.e., MOP conditions)." Remove the vice versa.

Line 139: change was to were and add "in North China" after "surface O3 concentration".

Line 141: I'd like to suggest that the OWI calculation in line 159 becomes Equation 2 and then wherever the authors "denote" a variable for this equation then Equation 2 is referenced. Such as here would become "denoted as V10mI, Eqn. 2".

Line 141: I would also suggest saying the region given in the brackets here is the box in Fig 2a.

Line 141: The authors switched gears and started describing the NOP with the sentence "The cloudy skies…." I would suggest consider keeping the MOP description together separate from the NOP.

Line 145, 149, 153: state boxed regions are from which panel in Fig 2 and add the Eqn. 2 reference.

Line 148: Explain why one would look for this "difference in the temperature at the surface and 200 hPa". Is it a good coincidence or what are the links in the meteorology for cold air aloft to

be expected with clear conditions instead of cloudy conditions?  What led the authors to make this plot?

Line 151:  The authors could look at ozone from ERA-Interim, as this dataset has assimilated satellite-retrieved ozone.  Otherwise, the authors should state clearly that this is "not shown" in the paper and then reference An et al., 2009.  There are other papers which also show this such as Ott et al., 2016, JGR.

Line 158:  Is "long-term" meaning a 10 year period here?

Line 159: As stated before, make OWI as a separate Equation from the text.

Line 168:  "the historical period before 2007 and the projected future" assumes the O3 emissions are the same as for the 11 year period from 2007 to 2017?

Line 170:  This first sentence should be a general one so I would suggest removing the "the" before reanalysis and the "was" before improved.

Line 175:  Is it possible to add 2006 to Table S1?

Line 180: "Before the mid-1990s" but looking at the figure is it also after 1983?

Line 181 and 183: Does "insignificant" in line 181 contradict the "strong" in line 183?

Line 182:  Over what time period do the "emissions of O3 precursors increase persistently and linearly"?

Line 189:  Can the authors be more definitive than "appeared to be the positive phase of the EU pattern".  Can the authors provide a reference if they can't be certain.

Line 196:  What does "After 2007, the EU index and the observational SDZ MDA8 synchronously changed" mean?  During period with data, one can see good agreement between the JJA EU and O3?

Line 198: Consider changing "intensified" to "large".  Also, "homodromous" is an odd choice of word.

Line 202: Add (Fig. 7c) after "300 hPa"

Line 210: Add "likely" before facilitated

Line 218: Check spelling of NCAR

Line 227:  Insert "(SI)" after "sea ice" as SI is used later in that sentence.

Line 232:  "could induce EU-like pattern"….so are they the EU or are they something else?  Be clear what we see in Fig 10c.  Line 236-237 the authors sound very confident that it is the EU pattern.

Line 246:  Is ASI for Arctic Sea Ice?

Line 253: Should the reference be Fig 12c not 12b?

Line 253-254: I think the statement "the photochemical reaction was significantly decelerated and the generation of surface O3 was rather weak" is speculative since this was not specifically shown.  It should be stated as such.

Line 255:  I also think the statement "The wet deposition effect was also significantly enhanced" is speculative since not shown specifically.

Line 256: Change "but" to "and"

Line 257: At the end of this paragraph I am left wondering, What scientific question did this section answer.  Make this clear.

Line 260: Change "Spatially" to "In general"

Line 261:  Is the EU predictable on seasonal time-scales?

Line 262: Does "long-term" mean 11 years or is it reference to the reanalysis datasets?

Line 262: Add "and ozone" before "observations"

Line 264: Add "which may help for seasonal forecasting" after "climatic driver".

Line 267: This final sentence in this paragraph is what I was looking for at the end of Section 5 (line 257).

Line 270: Can the authors pick one "induce" or "enhance" or can the accumulated sea ice in May do both?

Line 273: "was accelerated" sounds too bold to me. Maybe "supported" instead?

Line 274: Can the authors provide a reference for "were barotropic and could persist for a long time"

Line 274: "fairly" in fairly high seems weak for something that is over an air quality standard. Can the authors be less vague and more quantitative, possibly linking back to the thresholds used in MOP.

Line 275: Perhaps the authors want to add "and O3" before "measurements"

Line 276: "casually verified". I don't think the authors want to describe their work as casual.

Line 277: Describe How the OWI extended the time range of this study.

Line 282: "lucubrated" is a bit of an odd word

Line 283: if "the processes how the weather conditions impact the photochemical reaction were not deeply discussed here" then what did the authors show in this paper?

Line 284: Add references for "many previous studies"

Figure 2: Can the authors add MOP and NOP headers above the left and right panels, respectively? Also what are the units for each variable?

Line 474: Remove "s" at the end of "ERA-Interim datasets"

Figure 7: Shading temp in (b) and (c) might have matched shading in (a) better?

Figure 8: What are the units for the Y-Axis?

---

## Author Response (AR2)

**Response to Reviewer #1**

Overall I am happy with how the authors have addressed my comments. Some of the overstatement about "seasonal prediction" and the assumption that the Ozone Weather Index (OWI) ≈ surface ozone pollution have been removed. The authors have also dedicated more discussion to the meteorological links among the sea ice, Eurasia teleconnection pattern (EU), and the OWI. The addition of CAM5 numerical simulation experiments (and Figure 12) also adds confidence in the authors' findings on the links among sea ice, the EU pattern, and ozone-related weather conditions. Sections 4 and 5 have been extended and are much clearer. I am also pleased to see several of the supplementary figures moved to the main paper. This makes the paper easier to follow and makes more organizational sense.

Recommendation:

The authors have addressed my major concerns with the previous version. Aside from a couple specific comments and minor technical corrections, **I find this paper acceptable for publication in ACP.** The technical corrections were numerous so the English in the paper should be carefully reviewed once more by the authors and copy editor before publication.

**I'll suggest a slight title change because the current one is a bit awkward, but I think the authors have the right idea: "Climate variability links among Arctic sea ice, Eurasia teleconnection pattern and summer surface ozone pollution in North China"**

*Reply:*

   **Referring to the reviewer's comment, the title was revised to "*Links of Climate variability among Arctic sea ice, Eurasia teleconnection pattern and summer surface ozone pollution in North China*".**

*Revisions:*

Links of Climate Variability among Arctic sea ice, Eurasia teleconnection pattern and

summer surface ozone pollution in North China

**Specific/Minor Comments:**

**Line 243: What is "each September 1st?" Meteorological data from September 1st on each of the last 5 years? What 5 years? Please be clear here.**

*Reply:*

Related presentations were modified.

*Revisions:*

……During the control experiment (CTRL), the CAM5 model firstly integrated 20 years with climate mean initial and boundary conditions. Next, **the data in 1st September of the last 5 years (i.e., 16–20 years) were designated as five slightly different initial conditions. With each initial condition, the CAM5 model integrated 10 years**……

**Line 260: Could you explain your "generalized additive model" more in the text? How do you construct the model that gives you the red line in Figure 11?**

*Reply:*

The "generalized additive model" is introduced in Section 2 and a necessary reference is added.

*Revisions:*

……The generalized additive model, a data-driven method, is particularly effective at handling the complex nonlinear and non-monotonous relationships between the dependent variable and the independent variables (Hastieand Tibshirani, 1990). This approach used a smoothing function, determined by the independent variables themselves, to transform the expressions, and addressed the dependent variable with different probability distributions by the link function……

Hastie, T. J. and Tibshirani, R. J.: Generalized Additive Models, Chapman & Hall, London, UK, 1990.

**Technical Corrections:**

**Line 12: "Based on…"**

*Reply:*

According to the reviewer's comment, the sentence was revised.

*Revisions:*

……Based on the close relationships……

**Line 12: Be sure to subscript the O3 in all instances.**

*Reply:*

Throughout the manuscript, the neglects were corrected.

*Revision:*

……Basing on the close relationships between the $O_3$ concentration and the meteorological conditions……

**Line 21: Change "specially" to "specifically"**

*Reply:*

According to the reviewer's comment, the sentence was revised.

*Revisions:*

……variation of $O_3$ pollution, specifically    the related meteorological conditions……

**Line 24: I preferred the original "increasing" to "serious." This sentence could be clarified by writing: "Over the past several decades, along with social and economic development, air pollution has been increasing in China."**

*Reply:*

According to the reviewer's comment, the sentence was revised.

*Revisions:*

……Over the past several decades, due to fast economic development, air pollution has been increasing in China……

**Line 26-28: "haze pollution is being controlled…" "appearing as a sharp decrease in fine particulate matter."**

*Reply:*

According to the reviewer's comment, the sentence was revised.

*Revisions:*

……haze pollution is being controlled in recent years (The environmental statistics unit of stat-centre in Peking University, 2018), appearing as a sharp decrease in fine particulate matter……

**Line 28: Change "always occurred on" to "always occurs on"**

*Reply:*

According to the reviewer's comment, the sentence was revised.

*Revisions:*

……which always occurs on clear and sunny days……

**Line 29-30: It is unclear what you mean by "The negative effects of surface O3 pollution was not weaker than those of haze…" It is commonly known that particulate matter pollution is generally more harmful to human health than ozone pollution.**

*Reply:*

According to the reviewer's comment, the sentence was revised.

*Revisions:*

……The negative effects of surface $O_3$ pollution, such as corroding human's lungs and destroying agricultural crops and forest vegetation, were not weaker than those of haze……

**Line 42: delete the comma on this line.**

*Reply:*

According to the reviewer's comment, the sentence was revised.

*Revisions:*

…….northern Europe through the anomalous atmospheric circulations……

**Line 43: Change "to" to "that"**

*Reply:*

According to the reviewer's comment, the sentence was revised.

*Revisions:*

……anomalous atmospheric circulations that influence regional photochemical processes……

**Line 46-47: Suggested rewrite "A strong positive correlation also exists between the East Asian summer monsoon and summer mean ozone in China."**

*Reply:*

According to the reviewer's comment, the sentence was revised.

*Revisions:*

……A strong positive correlation between the East Asian summer monsoon and summer mean ozone were found by model simulations (Yang et al., 2014), illustrating that the changes in meteorological parameters, associated with East Asian summer monsoon, lead to 2–5% interannual variations of surface $O_3$ concentrations over central eastern China……

**Line 51: Suggested rewrite: "Local photochemical production was the main source of O3."**

*Reply:*

According to the reviewer's comment, the sentence was revised.

*Revisions:*

……The photochemical reaction was the main local sources of $O_3$ (Sun et al., 2019)……

**Line 52: Change "violent" to "intense"**

*Reply:*

According to the reviewer's comment, the sentence was revised.

*Revisions:*

……the intense solar radiation……

**Line 66: Change "specially" to "specifically"**

*Reply:*

According to the reviewer's comment, the sentence was revised.

*Revisions:*

……specifically the related meteorological conditions……

**Line 68-69: I think this sentence is unnecessary, or could be moved elsewhere to a more appropriate location in the text.**

*Reply:*

According to the reviewer's comment, the sentence was moved to the ending of this paragraph.

*Revisions:*

 The hourly $O_3$ concentration data from 2014 to 2017 in China were provided by the Ministry of Environmental Protection of China. As one of the three regional background air-monitoring stations in China, the hourly $O_3$ concentration data at the Shangdianzi station (SDZ: located at 40°39'N, 117°07'E and 293.3 m high) was continuously observed from 2006 to 2017, and were controlled by the National Meteorological Information Center, China Meteorological Administration. According to the Technical Regulation on Ambient Air Quality Index of China (the Ministry of Environmental Protection of China, 2012), the maximum daily average 8 h concentration of ozone (MDA8) was used to represent the daily $O_3$ conditions. The MDA8 was calculated as the maximum of the running 8 h mean $O_3$ concentrations during 24 hours in the day. However, the systematic observation duration of the surface $O_3$ concentration was much shorter than the meteorological measurements and could not support the climate analysis.

**Line 71: Change "high" to "amsl" for above mean sea level.**

*Reply:*

According to the reviewer's comment, the sentence was revised.

*Revisions:*

……(SDZ: located at 40°39'N, 117°07'E and 293.3 m amsl)……

**Line 72: Need a space between "to2017"**

*Reply:*

According to the reviewer's advice, the sentence was revised.

*Revision:*

……from 2006 to 2017……

**Line 76: In your response to my review you indicated that you were using sea ice area, but here you cite sea ice "concentrations." Please be clear about which data set you are using.**

*Reply:*

The downloaded data from Met Office Hadley Centre was sea ice concentration. When calculating the sea ice index, we first computed the sea ice area and then calculate the mean value.

**Line 88: "Hadley Centre"**

*Reply:*

According to the reviewer's advice, the sentence was revised.

*Revisions:*

……Met Office Hadley Centre……

**Line 91: "and vice versa"**

*Reply:*

According to the reviewer's advice, the sentence was revised.

*Revisions:*

……and *vice versa*……

**Line 101: "Hadley Centre"**

*Reply:*

According to the reviewer's advice, the sentence was revised.

*Revisions:*

……Hadley Centre……

**Line 108: "severe"**

*Reply:*

According to the reviewer's advice, the sentence was revised.

*Revisions:*

……the threshold of the severe surface $O_3$ pollution in China……

**Line 119: Suggested rewrite: "…, the covariation of SDZ MDA8 and MDA8 in North China strengthens the representativeness of SDZ for North China."**

*Reply:*

According to the reviewer's advice, the sentence was revised.

*Revisions:*

……the covariation of SDZ MDA8 and MDA8 in North China strengthens the representativeness of SDZ for North China……

**Line 124: I still find "non-surface" to be confusing. Perhaps make this "non-polluted surface O3" (NPO) and "moderately-polluted surface O3" (MPO)?**

*Reply:*

We revised the definition into "non-$O_3$ polluted (NOP) level" at surface and "moderate-$O_3$ polluted (MOP) level".

*Revisions:*

……we defined non-$O_3$ polluted (NOP) level at surface as the $O_3$ concentration < 100 μg/m³ and moderate-$O_3$ polluted (MOP) level with $O_3$ concentration > 215 μg/m³, respectively……

**Line 127: Replace "were not the NOP days" with "exceeded the NOP threshold," Also see my comment about this acronym.**

*Reply:*

According to the reviewer's advice, the sentence was revised.

*Revisions:*

……75% of summer days exceeded the NOP threshold even at the regional background……

**Line 133: Replace "During calculating" with "During the calculation of". The rest of this sentence is unclear and needs to be rewritten.**

*Reply:*

According to the reviewer's advice, the sentence was revised.

*Revisions:*

……The regional average meteorological elements were calculated as meteorological indexes, and here the selected regions determined on the most significantly different areas in the composites of MOP and NOP events in Figure 2……

**Line 139: Replace "was" with "were"**

*Reply:*

According to the reviewer's advice, the sentence was revised.

*Revisions:*

……the $O_3$ precursors in North China were dispersed……

**Line 142: Replace "the photochemical reaction" with "photochemistry"**

*Reply:*

According to the reviewer's advice, the sentence was revised.

*Revisions:*

……weakened the photochemistry by influencing exposure……

**Line 151: Replace "enlarged" with "enhanced"**

*Reply:*

According to the reviewer's advice, the sentence was revised.

*Revisions:*

……into the boundary layer enhanced the surface $O_3$ concentration……

**Line 156: "(Figures S2 and S3)"**

*Reply:*

According to the reviewer's advice, the sentence was revised.

*Revisions:*

……(Figures S2, S3)……

**Line 177: Delete ";even the MDA8 was the minimum"**

*Reply:*

According to the reviewer's advice, the sentence was revised.

*Revisions:*

and could verify the performance of the OWI. The JJA mean OWI in 2006 successfully reflected the variation in observed MDA8; even the MDA8 in 2006 was the minimum, confirming the robustness of the OWI. Derived from two different

**Line 179: Replace "kinds of reanalysis data." with "specific reanalysis data set."**

*Reply:*

According to the reviewer's advice, the sentence was revised.

*Revisions:*

……did not depend on the specific   reanalysis data……

**Line 213: Delete "a harsh and" and replace "reaction" with "production"**

*Reply:*

According to the reviewer's advice, the sentence was revised.

*Revisions:*

……supported efficient photochemical production of $O_3$……

**Line 214: Replace "by" with "with"**

*Reply:*

According to the reviewer's advice, the sentence was revised.

*Revisions:*

……with the NCEP/NCAR data……

**Line 215: "The correspondence between the large-scale…"**

*Reply:*

According to the reviewer's advice, the sentence was revised.

*Revisions:*

……The correspondence between large-scale EU teleconnection,and anti-cyclonic circulations were clear……

**Line 223: "responses like the EU pattern…"**

*Reply:*

According to the reviewer's advice, the sentence was revised.

*Revisions:*

……responses like the EU pattern……

**Line 224: Replace "monthly checked" with "evaluated each month."**

*Reply:*

According to the reviewer's advice, the sentence was revised.

*Revisions:*

……The correlation between the sea ice and JJA OWI was evaluated each month (Figure omitted)……

Line 228: Replace "anomalies may be" with "anomalies are"

*Reply:*

According to the reviewer's advice, the sentence was revised.

*Revisions:*

……May SI anomalies are followed by……

**Line 241: Replace "contributed" with "contributing"**

*Reply:*

According to the reviewer's advice, the sentence was revised.

*Revisions:*

……the preceding May sea ice anomalies contributing to the subsequent……

**Line 242: Replace "firstly integrated 20 years" with "was first integrated for 20 years"**

*Reply:*

According to the reviewer's advice, the sentence was revised.

*Revisions:*

……the CAM5 model was first integrated for 20 years with……

**Line 246: Replace "i.e., totally 30 sensitive runs." With "i.e. a total of 30 sensitivity runs."**

*Reply:*

According to the reviewer's advice, the sentence was revised.

*Revisions:*

……i.e., a total of 30 sensitivity runs……

**Line 247: Replace "of 30 sensitive ensembles" with "of the 30 sensitivity runs"**

*Reply:*

According to the reviewer's advice, the sentence was revised.

*Revisions:*

……of the 30 sensitivity runs……

**Line 248: Replace "were the" with "represent the"**

*Reply:*

According to the reviewer's advice, the sentence was revised.

*Revisions:*

……The differences (LowASI minus CTRL) represent the responses of……

**Line 253: Replace "violent" with "intense"**

*Reply:*

According to the reviewer's advice, the sentence was revised.

*Revisions:*

……intense sunshine……

**Line 253-254: Replace "the photochemical reaction" with "photochemical production"**

*Reply:*

According to the reviewer's advice, the sentence was revised.

*Revisions:*

……the photochemical production was significantly decelerated……

**Line 254: Replace "On the other side," with "Additionally,"**

*Reply:*

According to the reviewer's advice, the sentence was revised.

*Revisions:*

……Additionally, sufficient moisture……

**Line 256: Replace "but" with "and"**

*Reply:*

According to the reviewer's advice, the sentence was revised.

*Revisions:*

……was weakened, and the wet deposition effect……

**Line 263: Replace "NOAA" with "NCEP/NCAR"**

*Reply:*

According to the reviewer's advice, the sentence was revised.

*Revisions:*

……the ERA-Interim and NCEP/NCAR reanalysis datasets……

**Line 268: Replace "seasonal look" with "understanding of seasonal variability"**

*Reply:*

According to the reviewer's advice, the sentence was revised.

*Revisions:*

……to the understanding of seasonal variability of O3 pollution……

**Line 272: "less low- and medium-altitude cloud cover"**

*Reply:*

According to the reviewer's advice, the sentence was revised.

*Revisions:*

……less low- and medium-altitude cloud cover……

**Line 273: Replace "reaction" with "reactions"**

*Reply:*

According to the reviewer's advice, the sentence was revised.

*Revisions:*

……photochemical reactions……

**Line 276: Replace "casually" with "causally"**

*Reply:*

According to the reviewer's advice, the sentence was revised.

*Revisions:*

……and was causally verified by……

**Line 283: Replace "the photochemical reaction" with "photochemistry"**

**Line 284: Replace "the atmospheric" with "atmospheric"**

*Reply:*

According to another reviewer's comment, this sentence was removed.

**Figure S1 Caption: Delete "were" in "were belonged"**

*Reply:*

According to the reviewer's advice, the sentence was revised.

*Revisions:*

……the first lines belonged to JJA 2007 and JJA 2008……

**Figure S2 Caption: Replace "NOAA" with "NCEP/NCAR"**

*Reply:*

According to the reviewer's advice, the sentence was revised.

*Revisions:*

……calculated using the NCEP/NCAR datasets……

**Figure S3 Caption: Replace "basing" with "based" and replace "NOAA" with "NCEP/NCAR"**

*Reply:*

According to the reviewer's advice, the sentence was revised.

*Revisions:*

**Figure S3.** Differences in the boundary layer height between the MOP and NOP events during 2007–2014, based on the ERA-Interim (a) and NCEP/NCAR datasets (b). The black dots denote that the composite passed the 95% confidence level. The boxes represent the area to calculate the daily OWI. These composites were calculated using the NCEP/NCAR datasets.

Figure S4-S6 Caption: Replace "NOAA" with "NCEP/NCAR

*Reply:*

The neglects were corrected.

**Response to Reviewer #2**

The authors have greatly improved the manuscript responding to the three reviewers' comments. **I believe this paper is suitable for publication after minor revisions.**

**1. My two main concerns are that (1) the NOP and MOP need to be clearly defined as a range of O3 concentrations. (2) This is not clear to me and thus how Figure 2 is calculated is not clear to me either.**

*Reply:*

(1) The NOP and MOP was defined as a range of $O_3$ concentrations now, that is, defining non-surface $O_3$ polluted (NOP) level as the $O_3$ concentration $< 100$ µg/m³ and moderate surface $O_3$ polluted (MOP) level with $O_3$ concentration $> 215$ µg/m³.

(2) The composite results were calculated as the differences between MOP or NOP events with the rest events (i.e., all events but excluded MOP and NOP events). The calculations were illustrated in the caption of Figure 2 in the revised version.

*Revision:*

......we defined non-$O_3$ polluted (NOP) level at surface as the $O_3$ concentration $< 100$ µg/m³ and moderate-$O_3$ polluted (MOP) level with $O_3$ concentration $> 215$ µg/m³, respectively......

Caption of Figure 2 .......The composite results were calculated as the differences between MOP or NOP events with the rest events (i.e., all events but excluded MOP and NOP events)......

**I am troubled by two statements in the conclusion section:**
**(1) If "the processes how the weather conditions impact the photochemical reaction were not deeply discussed here" then I do not understand what was shown in this paper. (2) "However, the reason why the cooler high troposphere contributed to the surface ozone pollution was still an open question and needed further attention". If this is true, then why did the two temperature fields get**

**compared in the first place? What meteorological understanding would lead this to be shown in Figure 2?**

*Reply:*

According to the reviewer's comment, the related sentences were removed to avoid unwanted confusions.

*Revision:*

The joint effects of the climate anomalies and the historical emissions should be lucubrated using the numerical models in the future.  The EU pattern was a well-known continental Rossby wave train and could link the mid-high latitude climate with the change of the

**Two additional references that have recently been published may be of interest to the authors to include in their introduction:**

*Int J Climatol.*

Inter-annual variation of the spring haze pollution over the North China Plain: Roles of atmospheric circulation and sea surface temperature By Chen S, Guo J, Song L, Li J, Liu L, Cohen JB. https://rmets.onlinelibrary.wiley.com/doi/10.1002/joc.5842

ACP

Impacts of meteorology and emissions on summertime surface ozone increases over central eastern China between 2003 and 2015 by Lei Sun, Likun Xue, Yuhang Wang, Longlei Li, Jintai Lin, Ruijing Ni, Yingying Yan, Lulu Chen, Juan Li, Qingzhu Zhang, and Wenxing Wang https://www.atmos-chem-phys.net/19/1455/2019/

*Reply:*

The recommended publication related to $O_3$ pollution was cited in this manuscript.

The other paper, discussing the haze pollution, was also interesting and was cited in another manuscript of the authors, whose topic was the haze pollution.

**Minor and Technical Comments:**

**Line 10: define how many years is meant by "recently"**

*Reply:*

According to the reviewer's advice, the sentence was revised.

*Revision:*

……Summer surface $O_3$ pollution has rapidly intensified in China in the recent decade, damaging……

**Line 12: Make sure O3 has a subscript 3 everywhere**

*Reply:*

Throughout the manuscript, the neglects were corrected.

*Revision:*

……Basing on the close relationships between the $O_3$ concentration and the meteorological conditions……

**Line 14: To have both "major" and "significant" is too much. I recommend removing "significant"**

*Reply:*

According to the reviewer's advice, the sentence was revised.

*Revision:*

……a major globally atmospheric teleconnection pattern……

**Line 24: This opening sentence does not make sense to me. Has social and economic development been serious in China?**

*Reply:*

According to the reviewer's advice, the sentence was revised.

*Revision:*

……Due to fast economic development, air pollution has been serious in China……

**Line 25: Explain "haze" pollution. Is this from particulate matter or from a mix of other pollutants as well?**

*Reply:*

According to the reviewer's advice, the sentence was revised.

*Revision:*

……The major air pollution types in China are haze pollution (i.e., high-level fine particulate matter) in winter……

**Line 28: Change decreasing to decrease.**

*Reply:*

According to the reviewer's advice, the sentence was revised.

*Revision:*

……appearing as sharp decrease in……

**Line 28: Why? Are the controls specific to winter or only targeting what might impact haze? I could imagine some controls might have also helped the O3 precursors.**

*Reply:*

In the new publication (Li et al., 2018), Li and the co-authors pointed out "The most important cause of the **increasing ozone in NCP appears to be the decrease in PM$_{2.5}$**, slowing down the sink of **hydroperoxy radicals** and thus speeding up ozone production". Thus, "Decreasing ozone in the future will require a combination of NO$_x$ and VOC emission controls to overcome the effect of decreasing PM$_{2.5}$."

Ref: Li, K., Jacob, D. J., Liao, H., Shen, L., Zhang, Q., Bates, K. H.: Anthropogenic drivers of 2013–2017 trends in summer surface ozone in China, P NATL ACAD SCI USA., https://doi.org/10.1073/pnas.1812168116, 2018

**Line 29: Be specific, are "the negative effects of surface O3 pollution" regarding human health, agriculture, economy, etc?**

*Reply:*

According to the reviewer's advice, the sentence was revised.

*Revision:*

……The negative effects of surface $O_3$ pollution, such as corroding human's lungs and destroying agricultural crops and forest vegetation, were not weaker than those of haze……

**Line 32: remove "i.e."**

*Reply:*

According to the reviewer's advice, the sentence was revised.

*Revision:*

……the surface $O_3$ concentrations exceeded the ambient air quality standard of China (100 μg/m³) by 100–200 %......

**Lines 33-34: Why is this important? No explanation of ozone precursors, authors should consider explaining what those are before this statement.**

*Reply:*

According to the reviewer's advice, the explanation on the ozone precursors was forward.

*Revision:*

……Furthermore, the concentration of $O_3$ and its precursors, e.g. nitrogen oxides ($NO_x$) and volatile organic compounds (VOCs), in Beijing-Tianjin-Hebei……

**Line 35: Why are both O3 and the precursors large?**

*Reply:*

In the cited references, the authors only stated the observed variations from the measurements, but did not talk about the reasons for both $O_3$ and the precursors became large.

In our opinion, due to the increasing of precursors emissions, the $O_3$ production enlarged by consuming the precursors. However, the consumption cannot counteract the fast emission by human activities.

**Line 39: Define NOx and VOC. I would also suggest adding "with sunlight" after "react"**

*Reply:*

According to the reviewer's advice, the explanation on the ozone precursors was forward, and the sentence was revised.

*Revision:*

……Furthermore, the concentration of $O_3$ and its precursors, e.g. nitrogen oxides ($NO_x$) and volatile organic compounds (VOCs), in Beijing-Tianjin-Hebei……

……photochemically react with sunlight to generate $O_3$ under suitable weather conditions……

**Line 47: Add references for sentence ending in "mean ozone existed".**

*Reply:*

According to the reviewer's advice, the sentence was revised.

*Revision:*

……A strong positive correlation between the East Asian summer monsoon and summer mean ozone were found by model simulations (Yang et al., 2014), illustrating that the changes in meteorological parameters, associated with East Asian summer monsoon, lead to 2–5% interannual variations of surface $O_3$ concentrations over central eastern China……

**Line 51: The sentence "The photochemical reaction was the man local sources of O3" is in reference to which paper? And with NOx or VOCs?**

*Reply:*

According to the reviewer's advice, the sentence was revised.

*Revision:*

……The photochemical reaction was the main local sources of $O_3$ (Sun et al., 2019)……

**Line 61: Do the authors mean "previous studies of O3 pollution in China mainly focused on….."? Consider adding "of O3 pollution in China" or some other description of these previous studies.**

*Reply:*

According to the reviewer's advice, the sentence was revised.

*Revision:*

……previous studies of $O_3$ pollution in China mainly focused on observational analyses of several synoptic processes (e.g., Zhao and Wang, 2017),……

**Line 62: Is there a "lack of long-term surface O3 observations" at all or publicly available?**

*Reply:*

At all. The network was only established in recent years.

**Line 65: change "was" to "is"**

*Reply:*

According to the reviewer's advice, the sentence was revised.

*Revision:*

……as a preceding and effective driver, is also analysed……

**Line 66: The authors may want to stress here that this may have possible seasonal forecast application.**

*Reply:*

In the revised version, we toned down the contents about the seasonal prediction.

**Line 67: Can I suggest adding subsections 2.1 Observation-based data (Line 68) and 2.2 Modelbased data (Line 78) or something like? And maybe a section 2.3 Teleconnection before Line 89**

*Reply:*

Secondary headings are infrequent in the ACP format, thus the structure maintains itself.

**Lines 68, 69 and 72: I am confused by the terminology of "observation duration of the surface O3 concentration was much shorter than the meteorological measurements". Is the "much shorter" in relation to "the hourly O3 concentration data from 2014 to 2017, " or the Shangdianzi station "observed from 2006 to 2017"? Eleven years of data versus four years of data is definitely a different story for short term vs long-term.**

*Reply:*

What we wanted to present is that the network of ozone observation, not that in the single Shngdianzi site. The sentence was revised.

*Revision:*

……The **systematic** observation duration of the surface $O_3$ concentration was much shorter than the meteorological measurements……

**Line 72: make sure to add a space between "to2017".**

*Reply:*

According to the reviewer's advice, the sentence was revised.

*Revision:*

……from 2006 to 2017……

**Line 76: The sea ice data should not be in the same paragraph as the ozone data. Can the authors make it a new paragraph and add more details about this dataset such as the time period it covers, is this a widely used data set at that resolution?**

*Reply:*

According to the reviewer's advice, the sea ice data was carefully introduced.

*Revision:*

……The monthly sea ice concentrations (1°×1°) were downloaded from the Met Office Hadley Center (Rayner et al. 2003), which are widely used in sea ice-related analysis. The sea ice fields are made more homogeneous by compensating satellite microwave-based sea ice concentrations for the impact of surface melt effects on

retrievals in the Arctic, and by making the historical in situ concentrations consistent with the satellite data. The gridded sea ice data was available from 1870 to date, and those during 1979 to 2018 were extracted here……

**Line 78: Give more detail about the ERA-Interim dataset. It is available from 1979-present. Besides the horizontal resolution, can the authors provide the vertical resolution of the data used.**

*Reply:*

According to the reviewer's advice, the sentence was revised.

*Revision:*

……air temperature from 1000 hPa to 100hPa……

……The daily mean and monthly mean ERA-Interim data from 1979 to present were directly downloaded from the ERA-Interim website……

**Line 83: Give more detail about the NCEP/NCAR dataset. It is available from when to when, vertical resolution of the dataset, etc?**

*Reply:*

According to the reviewer's advice, the sentence was revised.

*Revision:*

……air temperature at from 1000 hPa to 100hPa……

……were downloaded, which was available from 1948 to present…….

**Line 86: The BLH is not from the NCEP/NCAR dataset but from a different reanalysis altogether. Describe the NOAA-20CR reanalysis (horizontal and vertical resolution, period of coverage, how it is different to NCEP/NCAR).**

*Reply:*

According to the reviewer's advice, the sentence was revised.

*Revision:*

……The BLH dataset was only available from 1979 to 2014 in the website of the NOAA-CIRES 20th Century Reanalysis version 2c……

**Line 88: Consider moving daily precip data to the 2.1 Observation section if it is an observation data set and not a model dataset.**

*Reply:*

Secondary headings are infrequent in the ACP format, thus the structure maintains itself.

**Line 91: I do not care for the use of "vice versa" used throughout the paper instead of writing it out what the alternative conditions may be. Perhaps this is a technical edit to discuss with ACP.**

*Reply:*

After checking some published paper in ACP, the "*vice versa*" was maintained.

**Line 93: Consider adding "for summertime" following "the calculation procedure for the EU index here"**

*Reply:*

According to the reviewer's advice, the sentence was revised.

*Revision:*

……the calculation procedure for the EU index for summertime was consistent with……

**Line 95: Give all equations numbers. This would then be Equation 1 and the authors can reference it in the paper as such.**

*Reply:*

According to the reviewer's advice, it was revised.

*Revision:*

$$\text{EU index} = [-1 \times \overline{\text{H500}}_{(70-80°N,60-90°E)} + 2 \times \overline{\text{H500}}_{(45-55°N,90-110°E)}$$

$$-1 \times \overline{\text{H500}}_{(35-45°N,120-140°E)}]/4 \qquad (1)$$

**Line 100: What is the top of the atmosphere for this model?**

*Reply:*

……CAM5.3 uses vertical hybrid δ-pressure coordinates including 26 layers with the top located at about 3.5 hPa……

*Revision:*

……CAM5.3 uses vertical hybrid δ-pressure coordinates including 26 layers with the top located at about 3.5 hPa……

**Line 107: I think change "The observations, with" to "Observations with"**

*Reply:*

According to the reviewer's advice, the sentence was revised.

*Revision:*

……Observations, with maximum MDA8……

**Line 108: Change "server" to "severe"**

*Reply:*

According to the reviewer's advice, the sentence was revised.

*Revision:*

……threshold of the severe surface……

**Line 109: I do not agree fully with the statement "the ozone polluted region has expanded" as over time the measurement coverage has increased spatially so it is possible the other areas were already polluted but just not observed. Can the authors defend this statement?**

*Reply:*

(1) In Figure 1, the number of sites was almost the same in 2015, 2016 and 2015. (2) Some overlapping sites were not polluted in 2015, but became polluted in 2016 and 2017.

**Line 111: In the figure the crosses to indicate the ozone is above the severe threshold of 265 make the circles actually look like a darker shade of red. Is there a reason the authors chose not to use colors based on the thresholds?**

*Reply:*

The colors were used to show the mean MDA8. We want to include more information without a new Figure, thus the crosses was selected.

**Line 115: Can SDZ still be considered one of the "background monitoring stations" as the authors argue it represents urban centers?**

*Reply:*

The SDZ station was the background monitoring station. The reason for using the data in SDZ is the long time rang of observation.

**Line 121: I do not understand why the large daily difference in MDA8 contradicts the quasiconstant emission of ozone precursors.**

*Reply:*

To avoid the confusion, the sentence was removed.

**Line 125: Are NOP levels anything above 100 micrograms per meter cubed or less than 100? Or are NOP levels between 100 to 215 and then MOP is anything above 215? The definition of NOP and MOP are not clear to me and this is VERY IMPORTANT for Figure 2.**

*Reply:*

According to the reviewer's advice, the sentence was revised.

*Revision:*

……we defined non-$O_3$ polluted (NOP) level at surface as the $O_3$ concentration < 100 μg/m³ and moderate-$O_3$ polluted (MOP) level with $O_3$ concentration > 215 μg/m³, respectively……

**Line 129: Why is NOP in brackets?**

*Reply:*

According to the reviewer's advice, the sentence was revised.

*Revision:*

……Both of the interannual variation in MOP and NOP days was significant at the 95% confidence level……

**Line 129: Do the authors have enough data for a trend analysis?**

*Reply:*

According to the reviewer's advice, the sentence was revised.

*Revision:*

……without an obvious trend……

**Lines 131-133: This sentence does not make sense. Maybe remove "although" and add "and" after "(Figure 2),"**

*Reply:*

According to the reviewer's advice, the sentence was revised.

*Revision:*

……Due to the significant covariation between the SDZ MDA8 to the MDA8 in North China, the meteorological conditions were composited for the MOP and NOP days in SDZ (Figure 2) and the results were also appropriate for those in North China……

**Line 132: If the composites are for MOP and NOP conditions, then what are they differenced from to get positive and negative values? It is very important to make sure the NOP and MOP definitions are made abundantly clear. If NOP is anything up to 100 and MOP is anything above 215, then are the authors using the conditions between the two (100 to 215) as the control?**

*Reply:*

(1) The NOP and MOP was defined as a range of $O_3$ concentrations now, that is,

defining non-surface $O_3$ polluted (NOP) level as the $O_3$ concentration $< 100$ µg/m³ and moderate surface $O_3$ polluted (MOP) level with $O_3$ concentration $> 215$ µg/m³.

(2) The composite results were calculated as the differences between MOP or NOP events with the rest events (i.e., all events but excluded MOP and NOP events).

*Revision:*

……we defined non-surface $O_3$ polluted (NOP) level as the $O_3$ concentration $< 100$ µg/m³ and moderate surface $O_3$ polluted (MOP) level with $O_3$ concentration $> 215$ µg/m³, respectively……

Caption of Figure 2 …….The composite results were calculated as the differences between MOP or NOP events with the rest events (i.e., all events but excluded MOP and NOP events)……

**Line 133: significant by which test?**

*Reply:*

According to the reviewer's advice, the sentence was revised.

*Revision:*

……The local and surrounding weather conditions were significantly different (t-test)……

**Line 135: Insert references to which Figure 2 panel the reader should look at. For example, "The anomalous southerlies (Fig. 2a), higher BLH (Fig. 2c), less rainfall (Fig. 2e), warmer surface air temperature (Fig. 2g), and cooler temperature in the high troposphere (Fig. 2g) favored surface O3 pollution (i.e., MOP conditions)." Remove the vice versa.**

*Reply:*

According to the reviewer's advice, the sentence was revised.

*Revision:*

……The anomalous southerlies (Figure 2a), higher BLH (Figure 2c), less rainfall (Figure 2e), warmer surface air temperature, and cooler temperature in the high

troposphere (Figure 2g) favored surface $O_3$ pollution……

**Line 139: change was to were and add "in North China" after "surface O3 concentration".**

*Reply:*

According to the reviewer's advice, the sentence was revised.

*Revision:*

……the $O_3$ precursors in North China were dispersed, and the surface $O_3$ concentration in North China was reduced……

**Line 141: I'd like to suggest that the OWI calculation in line 159 becomes Equation 2 and then wherever the authors "denote" a variable for this equation then Equation 2 is referenced. Such as here would become "denoted as V10mI, Eqn. 2".**

*Reply:*

We tried to modify the paragraph like the comment, but it became disordered. Thus it is maintained.

**Line 141: I would also suggest saying the region given in the brackets here is the box in Fig 2a.**

*Reply:*

According to the reviewer's advice, the sentence was revised.

*Revision:*

……(35–50 ᵒN, 110–122.5ºE, black box in Figure 2a, denoted as V10mI)……

**Line 141: The authors switched gears and started describing the NOP with the sentence "The cloudy skies…." I would suggest consider keeping the MOP description together separate from the NOP.**

*Reply:*

The physical mechanisms for MOP and NOP were the same, just opposite. To be concise, they were analyzed together, instead of separate discussion.

**Line 145, 149, 153: state boxed regions are from which panel in Fig 2 and add the Eqn. 2 reference.**

*Reply:*

According to the reviewer's advice, the sentence was revised.

*Revision:*

……(37.5–42.5 °N, 112–127.5 °E, black box in Figure 2e, denoted as PI)……

……(37.5–47.5 °N, 110–122.5 °E, black box in Figure 2g, denoted as DTI)……

……(37.5–47.5 °N, 112.5–120 °E, black box in Figure 2c, denoted as BI)……

**Line 148: Explain why one would look for this "difference in the temperature at the surface and 200 hPa". Is it a good coincidence or what are the links in the meteorology for cold air aloft to be expected with clear conditions instead of cloudy conditions? What led the authors to make this plot?**

*Reply:*

In sunny days, no more long-wave radiations were reflected by the cloud, this might be a possible reason.

**Line 151: The authors could look at ozone from ERA-Interim, as this dataset has assimilated satellite-retrieved ozone. Otherwise, the authors should state clearly that this is "not shown" in the paper and then reference An et al., 2009. There are other papers which also show this such as Ott et al., 2016, JGR.**

*Reply:*

It is really new work for us to analyze the assimilated satellite-retrieved ozone, but indeed a meaningful suggestion. In our next job, we will try to read the data and analyzed them.

**Line 158: Is "long-term" meaning a 10 year period here?**

*Reply:*

It means more than 30 years.

**Line 159: As stated before, make OWI as a separate Equation from the text.**

*Reply:*

Reply to **Line 141:** We tried to modify the paragraph like the comment, but it became disordered. Thus it is maintained.

**Line 168: "the historical period before 2007 and the projected future" assumes the O3 emissions are the same as for the 11 year period from 2007 to 2017?**

*Reply:*

According to the reviewer's advice, the sentence was revised.

*Revision:*

……Thus, it is reasonable to analyse the variation in **surface O$_3$-related atmospheric circulations** in North China using the OWI……

**Line 170: This first sentence should be a general one so I would suggest removing the "the" before reanalysis and the "was" before improved.**

*Reply:*

According to the reviewer's advice, the sentence was revised.

*Revision:*

……the quality of reanalysis data improved……

**Line 175: Is it possible to add 2006 to Table S1?**

*Reply:*

The data in 2006 were independent samples, thus we did not analyzed it.

**Line 180: "Before the mid-1990s" but looking at the figure is it also after 1983?**

*Reply:*

According to the reviewer's advice, the sentence was revised.

*Revision:*

……During mid-1980s to the mid-1990s……

**Line 181 and 183: Does "insignificant" in line 181 contradict the "strong" in line 183?**

***Reply:***

According to the reviewer's advice, the sentence was revised.

***Revision:***

……The strong interannual variation in the OWI after mid-1990s,……

**Line 182: Over what time period do the "emissions of O3 precursors increase persistently and linearly"?**

***Reply:***

According to the reviewer's advice, the sentence was revised.

***Revision:***

……increased persistently and linearly due to the steady economic development after 1978 in China……

**Line 189: Can the authors be more definitive than "appeared to be the positive phase of the EU pattern". Can the authors provide a reference if they can't be certain.**

***Reply:***

According to the reviewer's advice, the sentence was revised.

***Revision:***

……appeared to be the positive phase of EU pattern (Wang and He 2015)……

**Line 196: What does "After 2007, the EU index and the observational SDZ MDA8 synchronously changed" mean? During period with data, one can see good agreement between the JJA EU and O3?**

***Reply:***

According to the reviewer's advice, the sentence was revised.

***Revision:***

……the EU index and the observational SDZ MDA8 showed good agreement……

**Line 198: Consider changing "intensified" to "large". Also, "homodromous" is an odd choice of word.**

*Reply:*

According to the reviewer's advice, the sentence was revised.

*Revision:*

……the large EU pattern anomalies (i.e., the |EU pattern index| > 0.8 ×its standard deviation) always induced in-phase surface ozone pollution……

**Line 202: Add (Fig. 7c) after "300 hPa"**

*Reply:*

According to the reviewer's advice, the sentence was revised.

*Revision:*

……below 300 hPa (Figure 7c)……

**Line 210: Add "likely" before facilitated**

*Reply:*

According to the reviewer's advice, the sentence was revised.

*Revision:*

……likely facilitated……

**Line 218: Check spelling of NCAR**

*Reply:*

According to the reviewer's advice, the sentence was revised.

*Revision:*

……NCEP/NCAR data……

**Line 227: Insert "(SI)" after "sea ice" as SI is used later in that sentence.**

*Reply:*

According to the reviewer's advice, the sentence was revised.

*Revision:*

……sea ice (SI) area in May……

**Line 232: "could induce EU-like pattern"….so are they the EU or are they something else? Be clear what we see in Fig 10c. Line 236-237 the authors sound very confident that it is the EU pattern.**

*Reply:*

According to the reviewer's advice, the sentence was revised.

*Revision:*

……These positive sea ice anomalies could induce EU pattern responses in the subsequent summer……

**Line 246: Is ASI for Arctic Sea Ice?**

*Reply:*

In section 1, the abbreviation ASI was pointed to Arctic sea ice.

*Revision:*

……The role of May Arctic sea ice (ASI), as a preceding and effective driver……

**Line 253: Should the reference be Fig 12c not 12b?**

*Reply:*

According to the reviewer's advice, the sentence was revised.

*Revision:*

……cooled the air in the boundary layer (Figure 12c)……

**Line 253-254: I think the statement "the photochemical reaction was significantly decelerated and the generation of surface O3 was rather weak" is speculative since this was not specifically shown. It should be stated as such.**

*Reply:*

This statement was linked with Section3.

**Line 255: I also think the statement "The wet deposition effect was also significantly enhanced" is speculative since not shown specifically.**

*Reply:*

According to the reviewer's advice, the sentence was revised.

[Figure]

*Revision:*

……The wet deposition effect might be enhanced……

**Line 256: Change "but" to "and"**

*Reply:*

According to the reviewer's advice, the sentence was revised.

*Revision:*

……and the wet deposition effect……

**Line 257: At the end of this paragraph I am left wondering, What scientific question did this section answer. Make this clear.**

*Reply:*

According to the reviewer's advice, the sentence was revised.

*Revision:*

……That is, the positive relationship and associated physical mechanisms (i.e., climate links among ASI, EU pattern and summer surface ozone pollution in North China) were causally verified ……

**Line 260: Change "Spatially" to "In general"**

*Reply:*

According to the reviewer's advice, the sentence was revised.

*Revision:*

……In general, the $O_3$ concentrations……

**Line 261: Is the EU predictable on seasonal time-scales?**

*Reply:*

According to the reviewer's advice, the sentence was revised.

*Revision:*

……To reveal the climatic driver of summer surface $O_3$ pollution in North China……

**Line 262: Does "long-term" mean 11 years or is it reference to the reanalysis datasets?**

*Reply:*

It is reference to the reanalysis datasets.

**Line 262: Add "and ozone" before "observations"**

*Reply:*

According to the reviewer's advice, the sentence was revised.

*Revision:*

……was constructed based on meteorological and ozone observations……

**Line 264: Add "which may help for seasonal forecasting" after "climatic driver".**

*Reply:*

According to the reviewer's advice, the sentence was revised.

*Revision:*

……a preceding and efficient climatic driver, which may help for seasonal forecasting.……

**Line 267: This final sentence in this paragraph is what I was looking for at the end of Section 5 (line 257).**

*Reply:*

Line 257 was modified.

*Revision:*

Line 257: ……That is, the positive relationship and associated physical mechanisms (i.e., climate links among ASI, EU pattern and summer surface ozone pollution in North China) were causally verified ……

**Line 270: Can the authors pick one "induce" or "enhance" or can the accumulated sea ice in May do both?**

*Reply:*

According to the reviewer's advice, the sentence was revised.

*Revision:*

 ……The accumulated sea ice in May could induce the positive EU phase ……

**Line 273: "was accelerated" sounds too bold to me. Maybe "supported" instead?**

*Reply:*

 According to the reviewer's advice, the sentence was revised.

*Revision:*

 ……photochemical reaction to produce surface O3 was supported……

**Line 274: "fairly" in fairly high seems weak for something that is over an air quality standard. Can the authors be less vague and more quantitative, possibly linking back to the thresholds used in MOP.**

*Reply:*

 According to the reviewer's advice, the sentence was revised.

*Revision:*

 ……were continuous to achieve a high concentration……

**Line 275: Perhaps the authors want to add "and O3" before "measurements"**

*Reply:*

 Here, the sentence was maintained.

**Line 276: "casually verified". I don't think the authors want to describe their work as casual.**

*Reply:*

 According to the reviewer's advice, the sentence was revised.

*Revision:*

 ……was causally verified……

**Line 277: Describe How the OWI extended the time range of this study.**

*Reply:*

It is needless to expand how the OWI extended the time range, which was detailedly explained in the main text.

**Line 282: "lucubrated" is a bit of an odd word**

*Reply:*

According to the reviewer's advice, the sentence was revised.

*Revision:*

…should be studied using…

**Line 283: if "the processes how the weather conditions impact the photochemical reaction were not deeply discussed here" then what did the authors show in this paper?**

**Line 284: Add references for "many previous studies"**

*Reply:*

According to the reviewer's comment, the related sentences were removed to avoid unwanted confusions.

*Revision:*

The joint effects of the climate anomalies and the historical emissions should be lucubrated using the numerical models in the future.  The EU pattern was a well-known continental Rossby wave train and could link the mid-high latitude climate with the change of the

**Figure 2: Can the authors add MOP and NOP headers above the left and right panels, respectively? Also what are the units for each variable?**

**Line 474: Remove "s" at the end of "ERA-Interim datasets"**

*Reply:*

(1) the MOP and NOP headers were already indicated by >215 and <100. (2) The units were added in the captions.

Figure 2. Composite of the meteorological conditions associated with different $O_3$ events during 2007–2017. Results for MOP (a, c, e, g) and NOP (b, d, f, h) events included (a–b) surface wind (m/s, arrow) and v-wind (m/s, shading), (c–d) BLH (m), (e–f) precipitation (mm), (g–h) SAT (℃, shading), and temperature at 200 hPa (℃, contour). The black dots denote the composite results passed the 95% confidence level. The boxes represent the area used to calculate OWI. These composites were calculated using the ERA-Interim dataset. The green triangle in panel (a-b) illustrates the location of the Shangdianzi site. The composite results were calculated as the differences between MOP or NOP events with the rest events (i.e., all events but excluded MOP and NOP events).

**Figure 7: Shading temp in (b) and (c) might have matched shading in (a) better?**

*Reply:*

  Primitively, we plotted the Figures like this comment, but the distributions of relative humidity were not clear in contours. Thus, we changed the relative humidity into shading.

**Figure 8: What are the units for the Y-Axis?**

*Reply:*

  According to the reviewer's advice, the caption of Figure 8 was revised.

*Revision:*

[revised manuscript text omitted]

---

## Author Response (AR3)

**Response to Editor's Report**

Thank you for addressing the reviewers' comments. The paper is now much stronger and better communicates your research. However, there are two minor issues that should be corrected before publication.

First, the discussion around the definition of the O3 weather index (OWI) is confusing (page 47 in the author's responses, line 183). I support one reviewer's comment to add an Equation 2 with a more clear description of the terms. The OWI is an important part of the paper, but it's definition and definitions of its variables are buried within several paragraphs (e.g., lines, 183, 168, 172, and 176). Also what does the "I" represent in the variables (e.g., V10mI)? Addressing this reviewer's comment will improve readability of the paper.

**Reply:**

The definition of the ozone weather index was described in a separate paragraph and an Equation 2 was added in the revised manuscript.

**Revisions:**

To assess the interannual variation of surface  $O_3$  pollution and its relationship with climate variability (Cai et al., 2017), we fitted an  $O_3$  weather index (OWI) based on long-term meteorological observations. Firstly, the regional average meteorological elements were calculated as meteorological indexes (I), and here the selected regions determined on the most significantly different areas in the composites of MOP and NOP events in Figure 2. Then, we defined the OWI as Eqn. 2.

OWI=normalized V10mI+normalized BI–normalized PI+normalized DTI (2)

Where the V10mI is area-averaged meridional wind at 10 m (35–50 N, 110–122.5°E, black box in Figure 2a), and its correlation coefficient with the SDZ O3 concentration was 0.39. BI indicated area-averaged BLH (37.5–47.5 N, 112.5–120 E, black box in Figure 2c), and correlation coefficient with the SDZ O3 was 0.40. The PI defined as area-averaged precipitation (37.5–42.5 N, 112–127.5 E, black box in Figure 2e),

whose correlation coefficient with the SDZ  $O_3$  concentration was -0.35 (above the 99% confidence level). DTI represents the area-averaged difference in the temperature at the surface and 200 hPa (SAT minus temperature at 200 hPa, 37.5–47.5 N, 110–122.5 E, black box in Figure 2g), and the correlation coefficient with SDZ  $O_3$  concentration was 0.49.

Second, please justify better and more clearly why you are showing the difference in temperature between the surface and 200 hPa. See the original reviewer comment and your response on p. 31 of your response to reviews. For a paper focused on meteorology, you need a reference to back up why you made this figure. You didn't show SLP maps or upper-level wind patterns to show what the large-scale flow patterns might be. It is hard to tell from your now Figure 2a, but it may be for the polluted conditions, the winds are northward through the box, between cyclonic winds to the west and anticylonic flow to the east...hence warmer surface temps as in the warm sector/under a high pressure. Possibly the upper-level temperatures have to do with the location of the upper-level ridge/trough patterns but it is hard to tell from the composite of T200. If you can't explain why you plotted T at 200 hPa, it may not be worth mentioning or plotting at all. *Reply:*

According to your advice, the patterns of SLP, and geopotential height and wind at 200 hPa were supplemented as Figure S2.

Consistent with your reminder, for the polluted conditions, the winds are northward in North China, due to cyclonic anomalies to the west and anticyclonic flow to the east (Figure S2a). On the upper level, significant anticyclonic anomalies (Figure S2c) resulted in sunny days in summer. The temperature at 200 hPa above North China was significantly negative (Figure 2 g), dynamically associated with the upper-level anticyclone. The cold high pressure on the upper level also resulted in descending motion.